# STIM1-containing contact sites promote direct calcium flux from the endoplasmic reticulum to mitochondria

Yolanda Orantos-Aguilera[1,2,9], Irene Sanchez-Lopez[1,2,9], Carlos Pascual-Caro[1,2,3], Patricia Gómez-Suaga[4,5,6], Estela Area-Gomez [7], Eulalia Pozo-Guisado[2,8], Jorge Montesinos [7✉] & Francisco Javier Martin-Romero [1,2✉]

## Abstract

STIM1 is a transmembrane protein localized in the endoplasmic reticulum (ER), where it acts as a calcium ion sensor, activating store-operated Ca[2+] entry upon ER Ca[2+] depletion. Via cellular calcium influx, STIM1 is thought to indirectly affect mitochondrial calcium content. Here we show that STIM1 also interacts with mitochondrial proteins such as PTPIP51 and GRP75, suggesting its presence in mitochondria-associated ER membranes (MAMs), which are specialized ER regions that facilitate ER-mitochondria communication. Lowering STIM1 expression disrupts ER-to-mitochondria Ca[2+] transfer, reduces basal mitochondrial Ca[2+] levels, impairs maximal mitochondrial respiration, and reduces ATP production. The STIM1-GRP75 interaction depends on STIM1's Ca[2+]-sensing ability. ER Ca[2+] depletion or the constitutive-open R429C mutation both reduce STIM1 binding to GRP75, suggesting that conformational changes in STIM1 play a role in this interaction. Deletion analysis revealed that the STIM1 (551-611) segment is crucial for GRP75 binding, as the peptide STIM1(551-611) binds GRP75, while STIM1(Δ551-611) shows reduced binding. These findings reveal a previously unrecognized role of STIM1 in direct inter-organelle communication.

**Keywords** Calcium; GRP75; MAM; Mitochondria; STIM1
**Subject Categories** Membranes & Trafficking; Organelles

## Introduction

Mitochondria-associated ER membranes (MAMs) are specialized regions of the ER that establish transient physical and biochemical connections with mitochondria (Csordás et al, 2006; Giacomello and Pellegrini, 2016). These domains exhibit lipid raft-like properties, being enriched in cholesterol and sphingolipids that interact with each other and with lipid-binding proteins (Epand, 2004). In HeLa cells, approximately 5–20% of the mitochondrial network surface is in close proximity to the ER, highlighting the significance of these contact sites in cellular organization and signaling (Rizzuto et al, 1998).

An essential characteristic of MAMs, as membrane contact sites (MCS), is the presence of tethering proteins that facilitate the transient close apposition of the ER and mitochondrial membranes, enabling inter-organellar communication without membrane fusion (Scorrano et al, 2019). In mammals, numerous protein pairs at the MAM interface have been identified as potential tethers linking the ER to mitochondria. A notable example is the interaction between inositol 1,4,5-trisphosphate receptor (IP3R) in the ER and voltage-dependent anion-selective channel 1 (VDAC1) in the mitochondria, bridged by the glucose-regulated protein 75 (GRP75 or HSPA9). This complex, recognized as the principal pathway for ER-mitochondria Ca[2+] transfer, has been proposed as a tethering force (De Stefani et al, 2012; Szabadkai et al, 2006). However, its role appears to be primarily functional, as the deletion of all three IP3R isoforms does not disrupt the physical association between organelle membranes (Csordás et al, 2006). One of the earliest proteins proposed to facilitate ER-mitochondria tethering is mitofusin-2 (MFN2), a GTPase involved in mitochondrial fusion and found in both the ER and mitochondria. This role was postulated based on observations that the absence of MFN2 reduced ER-mitochondria colocalization and impaired key MAM-associated functions, such as mitochondrial Ca[2+] uptake following

[1]Department of Biochemistry and Molecular Biology, School of Life Sciences, Universidad de Extremadura, Badajoz, Spain. [2]Institute of Molecular Pathology Biomarkers, Universidad de Extremadura, Badajoz, Spain. [3]Paris Brain Institute (ICM), Sorbonne University, Inserm, CNRS, APHP, Hôpital de la Pitié Salpêtrière, Paris, France. [4]Department of Biochemistry and Molecular Biology, School of Nursing and Occupational Therapy, Universidad de Extremadura, Cáceres, Spain. [5]Network Center for Biomedical Research in Neurodegenerative Diseases, Instituto de Salud Carlos III (CIBER-CIBERNED-ISCIII), Madrid, Spain. [6]Instituto Universitario de Investigación Biosanitaria de Extremadura (INUBE), Cáceres, Spain. [7]Department of Biomedicine, Centro de Investigaciones Biológicas Margarita Salas, CSIC, Madrid, Spain. [8]Department of Cell Biology, School of Medicine, Universidad de Extremadura, Badajoz, Spain. [9]These authors contributed equally: Yolanda Orantos-Aguilera, Irene Sanchez-Lopez. ✉E-mail: jorge.montesinos@cib.csic.es; fjmartin@unex.es

IP3R stimulation (De Brito and Scorrano, 2008). In the same study, it was further suggested that ER-associated MFN2 mediates these interactions through homotypic or heterotypic binding to mitochondrial MFN2 or MFN1, respectively, with the latter being exclusively found in mitochondria (De Brito and Scorrano, 2008). Several studies have reported findings consistent with this model, reinforcing the idea of MFN2 as a key tethering protein in MAMs (Area-Gomez et al, 2012; Casellas-Díaz et al, 2021; Han et al, 2021; Naon et al, 2016). However, other investigations have challenged this perspective, showing that MFN2 ablation increases MAM contacts and enhances $Ca^{2+}$ transfer (Cosson et al, 2012; Filadi et al, 2015; Leal et al, 2016; Wang et al, 2015). Despite these contradictions, it is acknowledged that specific splicing variants of MFN2, named ERMIN2 and ERMIT2, are essential to shape the ER and to promote ER-mitochondria contacts (Naón et al, 2023), though its precise contribution remains a topic of debate. Among the best candidates for ER-mitochondria tethering are the ER-resident protein VAPB (vesicle-associated membrane protein-associated protein B) and the mitochondrial protein PTPIP51 (protein tyrosine phosphatase-interacting protein 51), with the loss of either protein consistently correlating with disruptions in both the physical structure and functional properties of MAMs (De Vos et al, 2012; Gomez-Suaga et al, 2017; Stoica et al, 2016).

The primary functions of MAMs focus on the transfer of $Ca^{2+}$ from the ER to the mitochondria, as well as the synthesis of phospholipids. These activities serve as a foundation for various MAM-associated processes, such as bioenergetics, lipid metabolism, mitochondrial dynamics, autophagy, apoptosis, ER stress responses, and inflammation (Krols et al, 2016; Martín-Guerrero et al, 2022; Perrone et al, 2020). Under resting conditions, mitochondrial free $Ca^{2+}$ concentration $[Ca^{2+}]_{mit}$ is similar to cytosolic levels (~100-200 nM). However, specific stimuli can trigger a 10- to 20-fold increase in $[Ca^{2+}]_{mit}$ (Modesti et al, 2021). Due to the relatively low $Ca^{2+}$ affinity of the mitochondrial calcium uniporter complex (mtCU)—the primary pathway for $Ca^{2+}$ entry into the mitochondrial matrix—efficient mitochondrial uptake relies on localized high-$Ca^{2+}$ microdomains. These microdomains are typically generated at the ER surface during $Ca^{2+}$ release (Giacomello et al, 2010). The driving force behind mitochondrial $Ca^{2+}$ accumulation is the highly negative membrane potential of the inner mitochondrial membrane (IMM) ($\Delta\Psi_m$, approximately -180 mV inside), which is established by the proton gradient generated through active proton pumping into the intermembrane space by mitochondrial complexes (Bernardi, 1999; Marchi et al, 2018). In this regard, MAMs play a critical role in facilitating efficient $Ca^{2+}$ transfer from the ER to the mitochondria, a process essential for maintaining $Ca^{2+}$ homeostasis and mitochondrial function. This transfer is primarily mediated by the IP3R-GRP75-VDAC complex, which physically bridges the ER and mitochondria (Ahumada-Castro et al, 2021; Csordás et al, 2018). Notably, all three isoforms of IP3R localize to MAM and actively participate in $Ca^{2+}$ transfer to the mitochondria.

VDAC are transmembrane proteins located in the outer mitochondrial membrane (OMM). These channels were traditionally considered passive conduits, freely permeable to ions and small molecules. However, emerging evidence suggests that VDAC act as an initial regulatory barrier for mitochondrial $Ca^{2+}$ entry. VDAC gating is voltage-dependent, shifting from a high-conductance "open" state to various low-conductance "closed" states at membrane potentials exceeding 30 mV (Rosencrans et al, 2021; Rostovtseva et al, 2005). Notably, while the closed states exhibit reduced permeability to metabolites, they display increased selectivity for $Ca^{2+}$. This finding highlights the active role of VDAC in regulating mitochondrial $Ca^{2+}$ homeostasis (Tan and Colombini, 2007).

The association between IP3R and VDAC is supported by a network of accessory proteins, with GRP75 serving as a central adaptor that directly bridges these channels, enabling efficient $Ca^{2+}$ transfer (Szabadkai et al, 2006). Among the key supporting interactors, the protein deglycase DJ-1 (or PARK7) plays a significant role in stabilizing the complex through its interactions with IP3R, GRP75, and VDAC1 (Liu et al, 2019). Additionally, the mitochondrial import receptor subunit TOM70 enhances this process by associating with IP3R and promoting its localization at MAMs (Filadi et al, 2018). Other notable contributors include transglutaminase 2 (TG2 or TGM2), which binds GRP75 to stabilize its interaction with IP3R (D'Eletto et al, 2018), and the sigma-1 receptor (Sig-1R). Sig-1R resides at MAMs in complex with the ER chaperone glucose-regulated protein 78 (GRP78) under resting conditions. However, upon ER $Ca^{2+}$ depletion, Sig-1R dissociates from GRP78 and binds IP3R, further facilitating $Ca^{2+}$ transfer (Hayashi and Su, 2007).

On the other hand, large-scale proteomic analyses of MAMs have revealed the presence of the ER $Ca^{2+}$ sensor STIM1 in these regions (Cho et al, 2020). STIM1 is a transmembrane protein predominantly located in the ER and is well-known for activating plasma membrane $Ca^{2+}$ channels during store-operated $Ca^{2+}$ entry (SOCE) upon ER $Ca^{2+}$ depletion. STIM1 functions as a $Ca^{2+}$ sensor within the lumen of the ER (Liou et al, 2005). When a decrease in free $Ca^{2+}$ concentration occurs, the dissociation of $Ca^{2+}$ from the EF-hand domain of STIM1 induces a conformational change that leads STIM1 to adopt an open conformation of its C-terminal domain (Stathopulos et al, 2008). This is followed by protein oligomerization and subsequent relocalization to ER–plasma membrane junctions (Liou et al, 2007), where STIM1 activates the plasma membrane $Ca^{2+}$ channel ORAI1 to activate SOCE (Luik et al, 2006). This regulatory activity of STIM1 on ORAI1 depends on a series of coiled-coil domains within the cytoplasmic segment of STIM1. While coiled-coil domains 2 and 3 (CC2/CC3) contain the minimal regions essential for ORAI1 coupling, CC1 modulates activation by maintaining an inhibitory interaction with CC2/CC3 until $Ca^{2+}$ store depletion triggers the conformational change that exposes CC2/CC3 (Maus et al, 2015).

Recently, proteomic analyses of STIM1 in HEK293 cells have identified potential interactions with mitochondrial proteins (Sanchez-Lopez et al, 2024), suggesting its localization at MAMs, consistent with previous findings (Cho et al, 2020). Despite this body of evidence, subsequent research has neither confirmed the presence of STIM1 in MAMs nor clarified its function in these regions. For this reason, our study focused on investigating the potential role of STIM1 in MAMs and its impact on inter-organelle communication. Specifically, we validated the interaction of STIM1 with mitochondrial proteins, evaluated the effects of STIM1 deficiency on the biochemical properties of MAMs, and examined the molecular mechanism by which STIM1 influences MAM function in ER-to-mitochondria $Ca^{2+}$ shuttling.

# Results

## STIM1 locates at ER-mitochondria contact sites

Although large-scale proteomic analyses suggest the presence of the ER Ca$^{2+}$ sensor STIM1 at MAMs (Cho et al, 2020), robust evidence typically requires low-throughput methods that focus on single proteins as preys to identify bait proteins or molecular interactors. A recent study has identified potential interactions between STIM1 and mitochondrial proteins, indicating that STIM1 may indeed localize at MAMs. Among these interactors are GRP75 (also known as HSPA9), VDAC3, SLC25A3, SLC25A6, and SLC25A5 (Sanchez-Lopez et al, 2024). Given that STIM1 is an ER Ca$^{2+}$ sensor, its interaction with GRP75 is particularly noteworthy, as this protein plays a role in ER-mitochondria Ca$^{2+}$ transfer. Furthermore, since VDAC channels mediate Ca$^{2+}$ import into mitochondria, we also explored their interaction with STIM1. However, VDAC3 was the only isoform identified in the interactome and has not yet been linked to Ca$^{2+}$-related functions.

To validate these interactions in vitro, we conducted a co-immunoprecipitation assay using STIM1-KO HEK293 cells stably transfected to inducibly express STIM1-GFP, or GFP (empty GFP vector) as a control. Cells were treated with doxycycline to induce gene expression, and whole cell lysates (WCL) were prepared. A GFP immunoprecipitation assay confirmed the interaction of STIM1 with GRP75 but not with any VDAC family member (Fig. 1A,B). Additionally, specific STIM1-GRP75 co-immunoprecipitation was observed at endogenous levels in HEK293 cell lysates (Fig. 1C), supporting the localization of STIM1 at MAMs. To further explore this, we performed subcellular fractionation in HEK293 cells to separate bulk ER and MAM fractions. The sequential protocol yielded fractions including total homogenate (TH), crude mitochondria (CM, containing mitochondria and MAMs), cytosol (Cyt), and ER-attached mitochondria (MER), facilitating protein distribution analysis across compartments (Fig. 1D). All fractions were subjected to immunoblotting (Fig. 1E), and the quality of the extractions was assessed using ACSL4 and ERLIN2 (MAM markers), TOM20 (mitochondria marker), p38MAPK (cytosolic marker), and IP3R1/2/3 (an ER protein known to be enriched in MAM). As a negative control, fractions from STIM1-KO HEK293 cells were analyzed. Results revealed an endogenous pool of STIM1 in the MAM fraction, which was enriched 5- to 6-fold compared to the bulk ER (Figs. 1E and EV1A).

To validate this finding, we analyzed the interaction between STIM1 and PTPIP51, a mitochondrial protein that regulates ER-mitochondria contacts by interacting with the ER-resident protein VAPB (De Vos et al, 2012; Stoica et al, 2014). For this assay, STIM1-GFP (or GFP as a control) was expressed in STIM1-KO HEK293 cells. STIM1-GFP was immunoprecipitated from cell lysates, and the co-precipitation of PTPIP51 was analyzed by immunoblotting. This approach confirmed PTPIP51 as a STIM1-interacting protein (Fig. 1F,G). Furthermore, the STIM1-PTPIP51 interaction was validated at endogenous levels using PLA, with rabbit anti-PTPIP51 and sheep anti-STIM1 primary antibodies (Fig. 1H).

An alternative strategy to assess the presence of STIM1 at ER-mitochondria contact sites was the use of dimerization-dependent fluorescent proteins (ddFP). Based on the constructs reported in (Miner et al, 2024), we generated cell lines for the inducible

expression of STIM1 fused to monomer B (STIM1-ddFP-B or STIM1-B, Fig. 1I,J). These cells were transfected with Mito-GA, that is, the monomeric GA protein (Ding et al, 2015) fused to a mitochondrial targeting sequence. After 24 h, reconstitution of the GA-B heterodimer was analyzed by fluorescence microscopy (Fig. 1J). This reconstitution marked the contact sites between STIM1 and mitochondria. Quantification of this fluorescence relative to the total mitochondrial mass, revealed by immunolocalization with anti-TOM20, indicated that approximately 37% of mitochondria are in contact with regions where STIM1 is present (Fig. 1K).

Taken together, these results demonstrate the presence of a pool of STIM1 localized at MAMs, where it interacts with mitochondrial proteins such as GRP75 and PTPIP51. This finding suggests that STIM1 may have a previously unrecognized function at ER-mitochondria contact sites.

## STIM1 deficiency increases the number of ER-mitochondria contact sites

To evaluate the role of STIM1 in MAMs, we investigated how its absence affects ER-mitochondria contacts. Specifically, we quantified the interactions between VAPB and PTPIP51 using PLA (Arjona et al, 2023; Gomez-Suaga et al, 2022; Stoica et al, 2014) in WT and STIM1-KO cells. As a control, we first confirmed that total levels of VAPB and PTPIP51 proteins were comparable in WT and STIM1-KO cells by immunoblotting (Fig. EV1B–E). Using rabbit anti-PTPIP51 and mouse anti-VAPB antibodies, we found that STIM1-deficient cells exhibited an increase in ER-mitochondria contacts compared to WT cells (Fig. 2A–C). Importantly, expression of ectopic STIM1 in STIM1-KO cells restored contact levels to those observed in WT cells in either U2OS or HEK293 cells (Fig. 2A–C), without altering PTPIP51 or VAPB protein levels (Fig. EV1B–E). Additionally, the increase in ER-mitochondria contacts was specific to the absence of STIM1, as STIM2-KO U2OS cells displayed contact levels comparable to WT cells (Fig. 2B), with no significant changes in the total levels of VAPB and PTPIP51 (Fig. EV1C–E). The characterization of the STIM2-KO cell line is shown in Fig. EV1F,G. These findings demonstrate that the absence of STIM1 leads to an increase in ER-mitochondria contact sites, a phenotype that can be reversed by restoring STIM1 expression.

An alternative approach to determine ER-mitochondria contacts is based on the use of ddFPs, as indicated above. In this case, U2OS cells were transiently transfected for the expression of Mito-GA and ER-B, as described in Miner et al (2024). To analyze the mitochondrial mass, an immunolocalization of TOM20 was performed. The analysis of both fluorescence signals allowed us to identify these contact sites (Fig. 2D), which were significantly increased in STIM1-deficient cells (Fig. 2E), a result consistent with that observed with the PLA between VAPB and PTPIP51 (Fig. 2A–C).

## STIM1 deficiency impairs ER-to-mitochondria Ca$^{2+}$ transfer

To further explore the functional consequences of this increased physical interaction between the organelles, we investigated its impacts on a specific biochemical pathway occurring at MAMs. A well-characterized function of MAM is to serve as a platform for

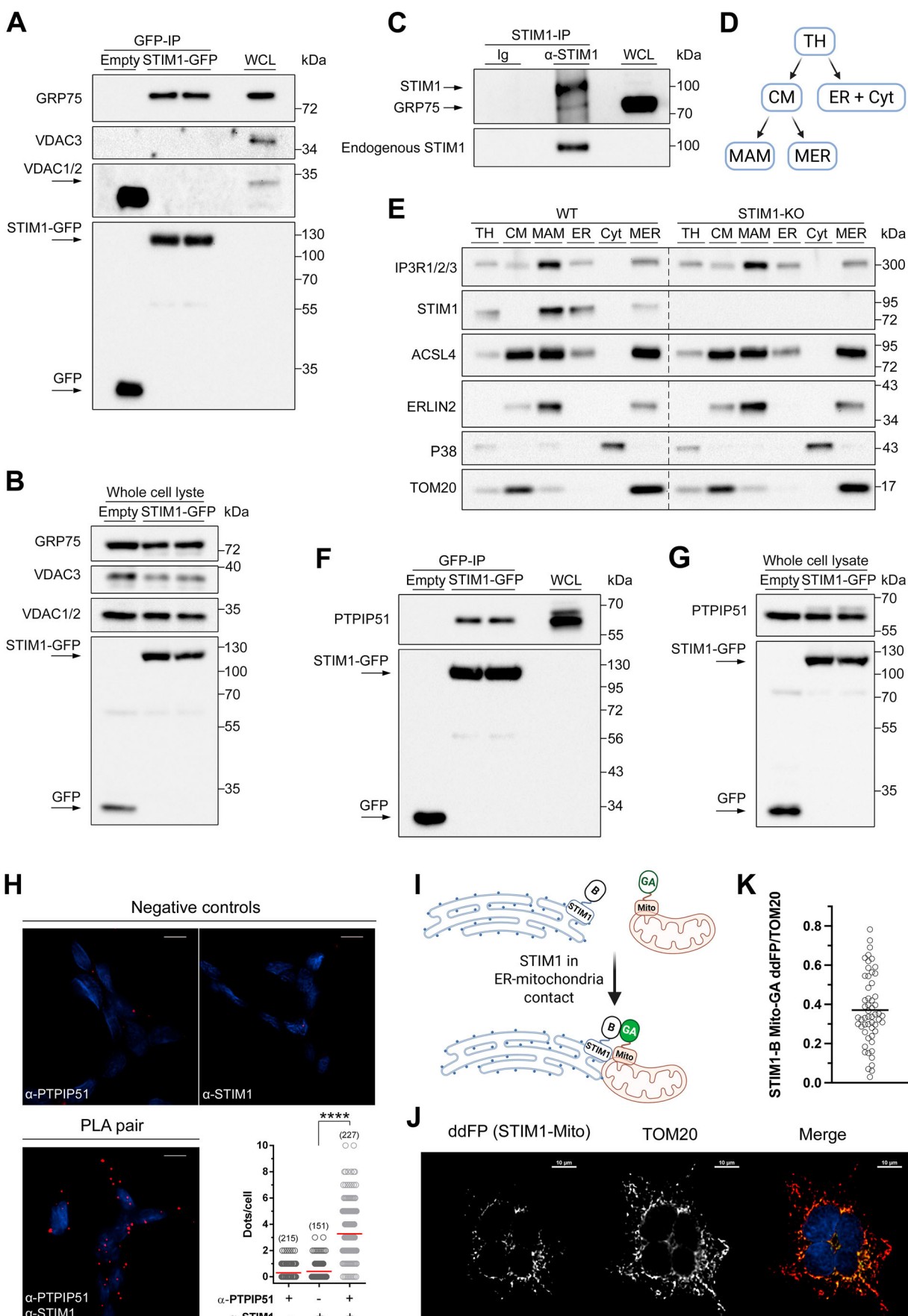

◄ **Figure 1.  A pool of STIM1 locates at ER-mitochondria contact sites.**

(A) STIM1-GFP or GFP (empty GFP vector) expression was induced with 1 µg/ml doxycycline for 22 h in STIM1-KO HEK293 cells. Immunoprecipitation of GFP-tagged proteins was performed using 1 mg of whole cell lysate (WCL), and co-precipitated proteins were analyzed by immunoblotting. For detecting VDAC1 and VDAC3, two different specific antibodies were tested, yielding the same result for both proteins. The antibodies used for VDAC1 were sc-390996 (Santa Cruz Biotechnology) and 10866-1-AP (Proteintech), while those for VDAC3 were PA5-51156 (ThermoFisher Scientific) and 55260-1-AP (Proteintech). As a positive control, 3 µg WCL was loaded (WCL lane). Blots are representative of 3 biological replicates. Levels of immunoprecipitated GFP or STIM1-GFP were analyzed as a loading control. (B) Total levels of GRP75 and VDAC proteins in WCL were evaluated by immunoblotting (30 µg protein/lane). (C) WCL (2.5 mg) was incubated with the anti-STIM1 antibody followed by Dynabeads™. Co-precipitation of GRP75 with endogenous STIM1 was evaluated by immunoblotting with a specific antibody (upper panel), which also detected STIM1 non-specifically due to the high amount of immunoprecipitated protein. As a positive control, 1 µg WCL was loaded. As a negative control, normal rabbit IgG was used instead of anti-STIM1 antibody (Ig lane). Blots are representative of 3 technical replicates from 2 biological replicates. Immunoprecipitated STIM1 levels were assessed as a loading control (lower panel). (D) Representative scheme of the subcellular fractionation procedure. The following fractions were isolated: total homogenate (TH), crude mitochondria (CM), mitochondria-associated ER membranes (MAM), ER-attached mitochondria (MER), bulk ER (ER), and cytosol (Cyt). (E) Total levels of STIM1 were analyzed in MAM fraction of HEK293 cells. All fractions were analyzed by immunoblotting (5 µg protein/lane). IP3R1/2/3 were used as an example of ER proteins enriched in MAM, ACSL4 and ERLIN2 were used as MAM markers, p38MAPK as a cytosolic marker and TOM20 as a mitochondrial marker. Fractions from STIM1-KO HEK293 cells were evaluated in separated gels (indicated by the dotted line) as negative control. (F) STIM1-KO HEK293 cells stably transfected for the inducible expression of STIM1-GFP (or GFP-empty as a control) were treated with 1 µg/ml doxycycline for 22 h. Immunoprecipitation of GFP-tagged proteins was performed from 1 mg WCL and the co-precipitation of PTPIP51 was assessed by immunoblotting. WCL (3 µg) was loaded as a positive control. Blots are representative of 3 biological replicates. Levels of immunoprecipitated GFP were analyzed as a loading control. (G) Total levels of PTPIP51 in WCL were evaluated by immunoblotting (30 µg protein/lane). (H) HEK293 cells cultured on collagen-coated coverslips were methanol-fixed and incubated with the specified primary antibodies (rabbit anti-PTPIP51 and/or sheep anti-STIM1) along with the DNA probes rabbit-PLUS and sheep-MINUS. As negative controls, cells were incubated with single primary antibodies. PLA signal (red dots) was analyzed under fluorescence microscopy. The panel shows representative images for all conditions. Scale bar = 10 µm. The graph shows the quantification of interactions (number of red dots) detected by PLA, with the number of cells evaluated in parentheses and the mean of the data represented by the red line. Statistical analysis with unpaired t-test, $p < 0.0001$. (I) Experimental design scheme for fluorescence reconstitution using ddFP. This diagram was created with BioRender.com. (J) STIM1-KO HEK293 cells engineered for the expression of inducible STIM1-ddFP-B were transfected for the transient expression of Mito-GA. Reconstitution of the GA-B heterodimer green fluorescence was monitored together with the immunolocalization of TOM20 (AlexaFluor-594). (K) Ratio of green (STIM1-mitochondria contacts) and red (mitochondria) fluorescence from 55 cells and 3 independent experiments. Statistical analysis with unpaired t-test, $p = 0.0071$. Source data are available online for this figure.

$Ca^{2+}$ transfer from the ER to the mitochondria (Csordás et al, 2010; Szabadkai et al, 2006). We examined ER-mitochondria $Ca^{2+}$ transfer by assessing mitochondrial $Ca^{2+}$ uptake following stimulation of $Ca^{2+}$ efflux from the ER via IP3R receptors. As a control, we evaluated the ability of the ER to release $Ca^{2+}$ in response to ATP +carbachol (CCh) in WT and STIM1-KO cells by measuring the free $Ca^{2+}$ concentration in the ER ($[Ca^{2+}]_{ER}$) before and after stimulation with agonists (Fig. 3A–C). Data analysis revealed that STIM1-KO cells had significantly lower basal $[Ca^{2+}]_{ER}$ compared to WT cells (Fig. 3B). However, both cell lines exhibited relatively similar release of $Ca^{2+}$ in response to stimulation, although STIM1-KO cells maintained a significant difference in $[Ca^{2+}]_{ER}$ after the stimulus compared to WT cells (Fig. 3C). While the release of $Ca^{2+}$ from the ER appeared comparable, the observed differences in $[Ca^{2+}]_{ER}$ were statistically significant (Fig. 3D). It is important to highlight that the calibration method of the signal from the $Ca^{2+}$ probe ER-GCaMP6-210 followed what it was described previously (de Juan-Sanz et al, 2017), i.e., with the addition of 5 µM ionomycin + 5 mM EGTA (to monitor Fmin) followed by the addition of 5 µM ionomycin + 10 mM $Ca^{2+}$ (to monitor Fmax). While this approach provides a consistent internal basis for comparison, we acknowledge that the absolute values may be either overestimated or underestimated due to the lack of high accuracy in the determination of Fmin and Fmax.

To further address these differences, we assessed ER $Ca^{2+}$ release by measuring cytosolic free $Ca^{2+}$ concentration ($[Ca^{2+}]_i$) following stimulation with ATP+CCh (Fig. 3E,F). Fura-2-loaded cells were stimulated, and analysis of $Ca^{2+}$ kinetics revealed no significant differences in cytosolic $[Ca^{2+}]_i$ between cell lines after IP3R stimulation (Fig. 3F).

We measured the concomitant mitochondrial $Ca^{2+}$ uptake to determine whether the increased ER-mitochondria contacts observed in STIM1-KO cells affected ER-to-mitochondria $Ca^{2+}$ transfer. HEK293 cells stably expressing the $Ca^{2+}$ sensor mito$^{4x}$-GCaMP6f were treated with doxycycline to induce construct expression (Fig. 3G). The fluorescence intensity of the $Ca^{2+}$ sensor was then measured following ATP+CCh stimulation. Interestingly, STIM1-KO mitochondria exhibited almost no $Ca^{2+}$ uptake compared to control cells (Fig. 3H,I). To investigate whether this deficiency in mitochondrial $Ca^{2+}$ uptake impacted steady-state $[Ca^{2+}]_{mit}$ we recorded the fluorescence from cells transiently transfected to express the mitochondrial $Ca^{2+}$ sensor 4mtD3cpv. Our results showed that STIM1-KO cells had approximately half the steady-state $[Ca^{2+}]_{mit}$ levels of WT cells (Fig. 3J).

One possible cause of the reduced $Ca^{2+}$ transfer to mitochondria in STIM1-KO cells could be an alteration in the expression profile of key proteins involved in this process. To investigate this, we examined whether the absence of STIM1 affected the expression of any IP3R isoforms, GRP75, VDAC1, or MCU by immunoblotting. No differences in the expression of these proteins were observed between WT and STIM1-KO whole cell lysates (Fig. EV2A–G). Therefore, we assessed their levels in MAMs relative to total ER and found no differences in the distribution of the three IP3Rs (Fig. 4A–D). The expression of IP3R1 was assessed using an antibody against all IP3R isoforms, as a specific anti-IP3R1 antibody showed significant background signal in MAM fractions. Using a pan-anti-IP3R antibody proved to be a valid approach, as no changes were observed in IP3R3 and IP3R2 levels when analyzed with isoform-specific antibodies. Regarding GRP75 and VDAC1, these proteins were more enriched in the CM fraction than in MAMs, as expected (Fig. 4A). For this reason, we analyzed them on a separate gel, loading only the MAM fraction to obtain more accurate quantification. This approach revealed reduced GRP75 and VDAC1 levels in STIM1-KO MAMs (Fig. 4E–G). This reduction in GRP75 and VDAC is not attributable to changes in total mitochondrial mass, as no differences were observed between

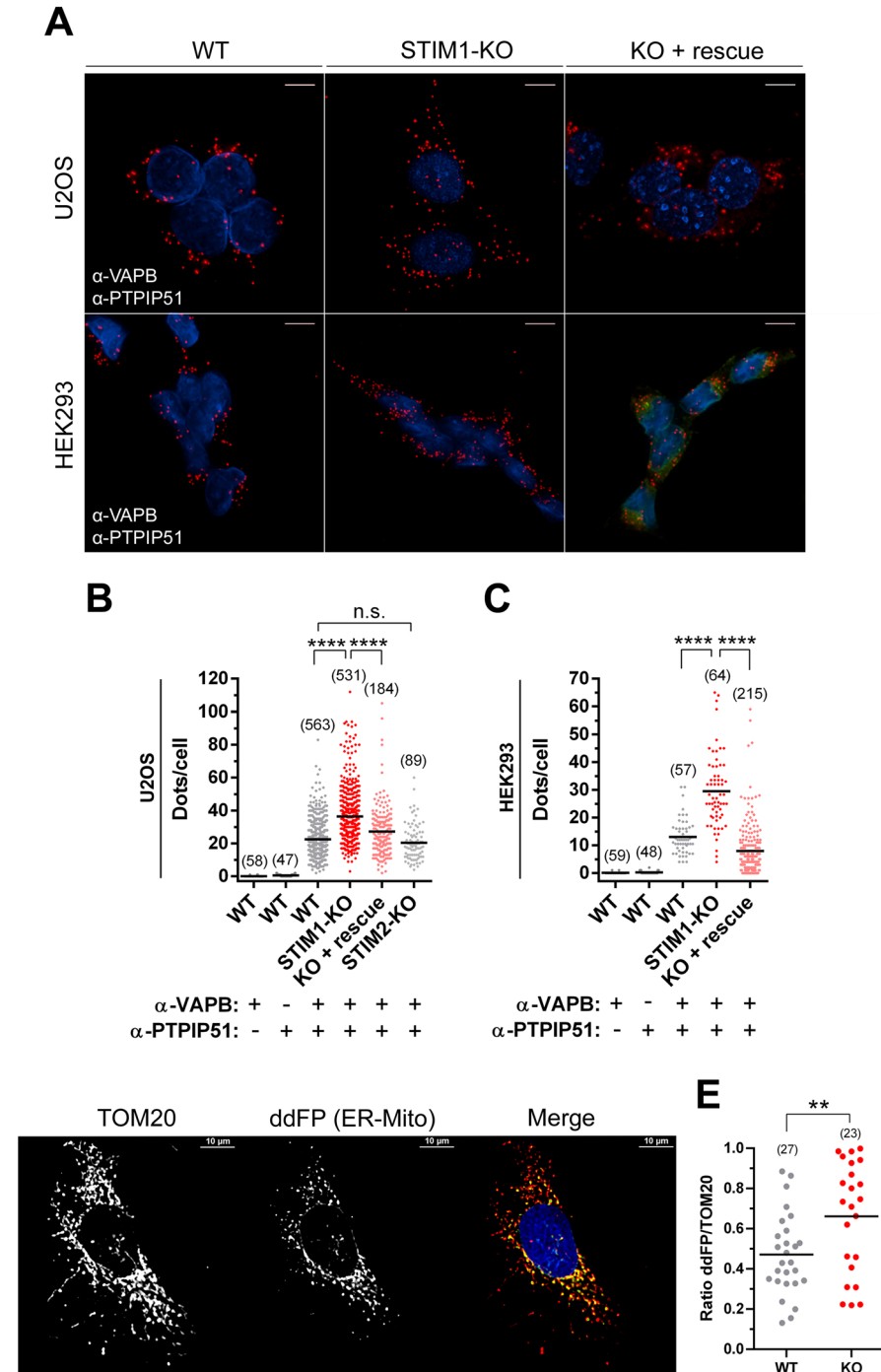

**Figure 2. STIM1 deficiency increases the number of ER-mitochondria contacts.**

(A) The number of ER-mitochondria contacts was assessed by analyzing the interactions between VAPB (ER) and PTPIP51 (mitochondria) in HEK293 and U2OS cells. Wild-type U2OS, STIM1-KO U2OS, and STIM1-KO U2OS cells stably expressing STIM1-mCherry (KO + rescue) (upper micrographs) were cultured on collagen-coated coverslips and fixed with methanol. Cells were then incubated with the indicated primary antibodies (mouse anti-VAPB and/or rabbit anti-PTPIP51), as well as with the DNA probes rabbit-PLUS and mouse-MINUS. The same assay was performed using HEK293 cells (bottom micrographs), where the rescue condition was represented by the STIM1-KO HEK293 cell line stably transfected for inducible expression of STIM1-GFP. As negative controls, cells were incubated with single primary antibodies. The panel shows representative images for all conditions described, except for negative controls. Scale bar = 10 μm. (B, C) Quantification of the interactions (number of red dots) between VAPB and PTPIP51 detected by PLA in U2OS (B) and HEK293 (C) cells. For the U2OS cell line, STIM2-KO cells were also evaluated. The number of cells evaluated is given in parentheses, and the mean of the data represented by the black line. Statistical analysis with unpaired t-test in all cases. P values are ****$p < 0.0001$ and $p = 0.1027$ (n.s.). (D) WT or STIM1-KO U2OS cells were transfected for the transient expression of Mito-GA and ER-B. GA–B heterodimer green fluorescence was monitored together with the immunolocalization of TOM20 (AlexaFluor-594). Scale bar = 10 μm. (E) Ratio of green (ER-contacts) and red (mitochondria) fluorescence from 27 WT cells or 23 KO cells, and two independent experiments. Statistical analysis with unpaired t-test, $p = 0.0071$. Source data are available online for this figure.

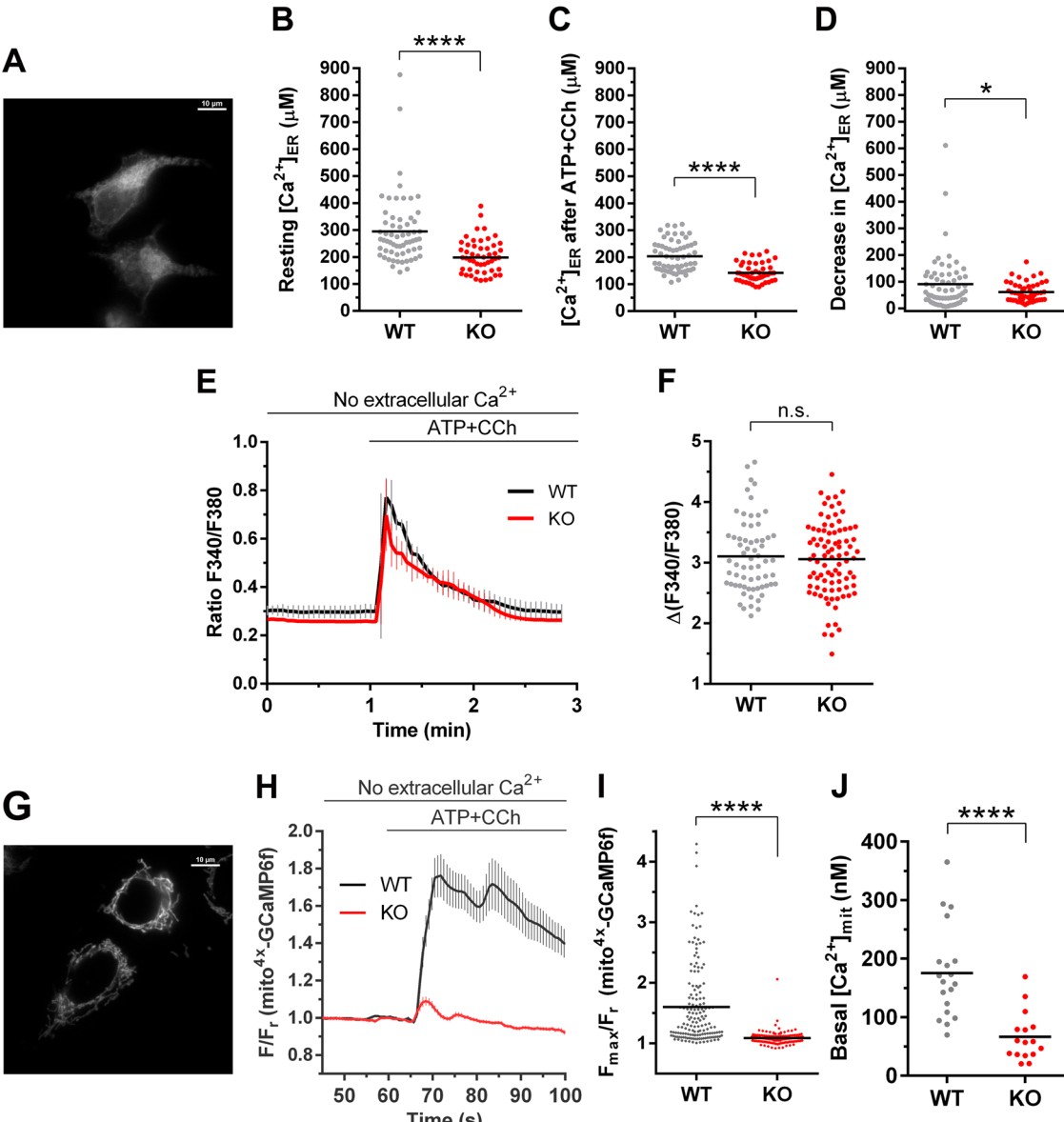

**Figure 3. STIM1 deficiency impacts ER-to-mitochondria Ca²⁺ shuttling.**

(**A**) HEK293 cells were transfected to express the Ca²⁺ sensor ER-GCaMP6-210. Scale bar = 10 μm. (**B**) Resting $[Ca^{2+}]_{ER}$ values were defined as the steady-state fluorescence value after the medium conditioning step and before stimulation with 100 μM ATP + 100 μM CCh. Statistical analysis with unpaired t-test, $p < 0.0001$. (**C**) Values of $[Ca^{2+}]_{ER}$ after Ca²⁺ release stimulated by ATP+CCh. Statistical analysis with unpaired t-test, $p < 0.0001$. (**D**) Decrease in $[Ca^{2+}]_{ER}$ after ATP+CCh stimulus. This decrease was calculated by subtracting the $[Ca^{2+}]_{ER}$ after stimulation with ATP+CCh from the resting $[Ca^{2+}]_{ER}$. Statistical analysis with unpaired t-test and $p = 0.0471$. In (**B–D**), the $[Ca^{2+}]_{ER}$ values were monitored in different ROIs from 4 independent experiments. Total number of ROIs analyzed: $n = 62$ WT and $n = 50$ KO ROIs. The black line represents the mean of the data. (**E**) Measurement of $[Ca^{2+}]_i$ after Ca²⁺ release from the ER in WT and STIM1-KO cells. Representative fluorescence profile of fura-2-loaded cells after stimulation with 100 μM ATP + 100 μM CCh in Ca²⁺-free Hank's balanced salt solution (min = 1). The graph shows the mean F340/F380 ratio ± S.D. over time from 3 independent experiments for WT HEK293 (black line) and 4 experiments for STIM1-KO cells (red line). Total number of cells analyzed: $n = 65$ WT and $n = 89$ KO cells. (**F**) Analysis of the maximum increase in the F340/F380 ratio after ATP+CCh addition in Ca²⁺-free assay medium. The maximum values of the F340/F380 ratio of each cell were normalized relative to the resting ratio value to calculate the magnitude of the increase in $[Ca^{2+}]_i$ after ER Ca²⁺ release. The mean of the data is represented by the black line. Statistical analysis with unpaired t-test. (**G**) Representative fluorescence image and (**H**) fluorescence profile of mito⁴ˣ-GCaMP6f after ATP+CCh stimulus. Cells stably transfected for the expression of the mitochondrial Ca²⁺ sensor mito⁴ˣ-GCaMP6f were cultured on collagen-coated coverslips and treated with 1 μg/ml doxycycline to induce expression of the Ca²⁺ sensor. ER Ca²⁺ release was triggered with ATP+CCh in Ca²⁺-free medium, as in (**E**) and (**F**). The ratios of fluorescence at each time point relative to resting fluorescence (F/Fᵣ) were calculated in different ROIs. The graph shows the mean F/Fᵣ ± S.E.M. from 4 independent experiments for WT HEK293 ($n = 39$ WT ROIs, black line) and STIM1-KO cells ($n = 40$ KO ROIs, red line). (**I**) Analysis of the maximum fluorescence increase ($F_{max}/F_r$) after ATP+CCh. The $F_{max}/F_r$ of all ROIs was analyzed from 15 independent experiments for WT cells and 17 experiments for KO cells ($n = 160$ WT and $n = 169$ KO ROIs). The mean of the data is represented by the black line. Statistical analysis with unpaired t-test, $p < 0.0001$. (**J**) Cells were transiently transfected to express the 4mtD3cpv sensor to measure basal $[Ca^{2+}]_{mit}$. Data from $n = 19$ WT cells and $n = 16$ STIM1-KO cells from 4 independent experiments. The black line represents the mean of data. Statistical analysis with unpaired t-test, $p < 0.0001$. Source data are available online for this figure.

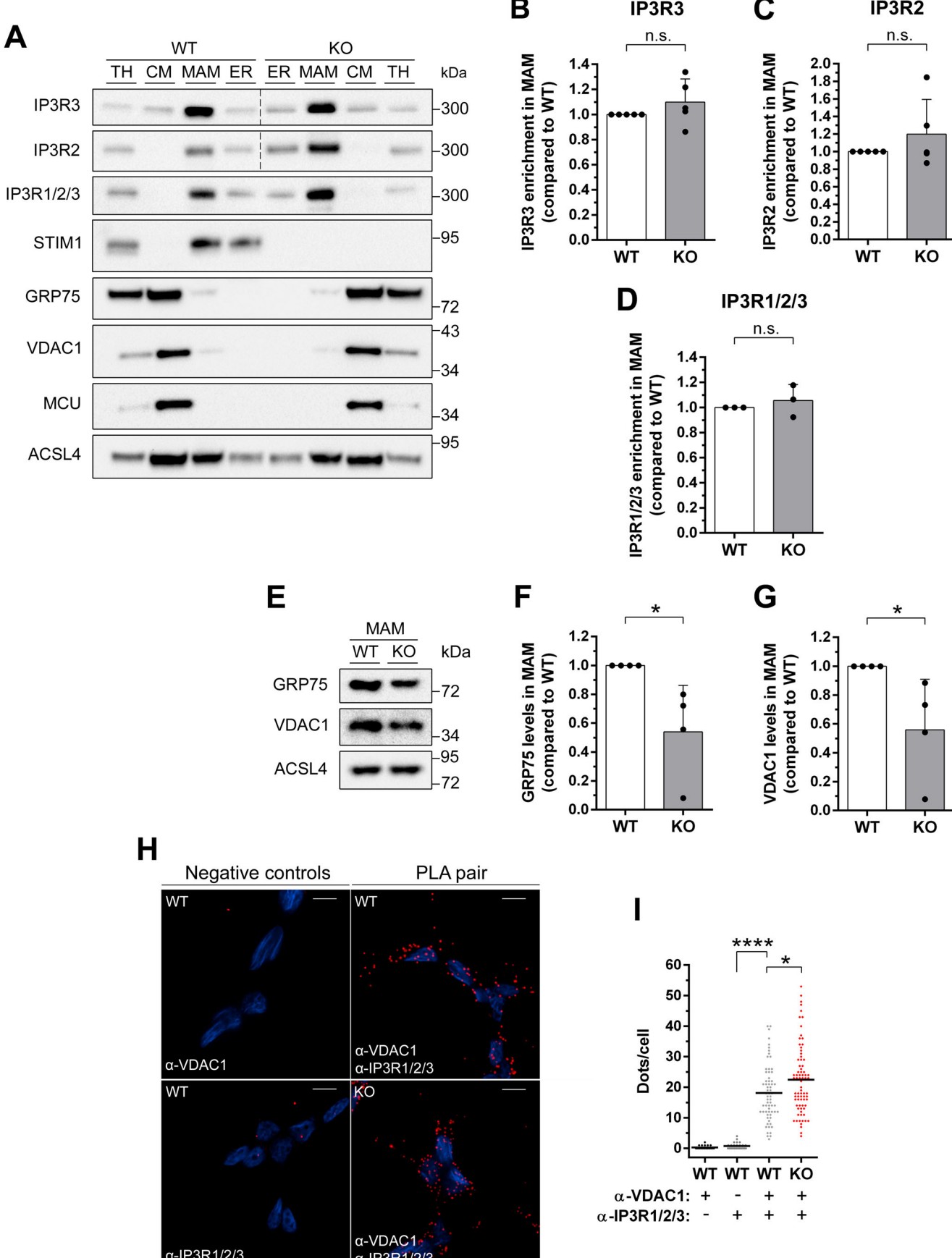

**Figure 4.  STIM1-deficient cells show decreased levels of GRP75 in MAMs.**

(A) Wild-type HEK293 and STIM1-KO cells were subjected to MAM fractionation, and TH, CM, MAM, and ER fractions were analyzed by immunoblotting (5 µg protein/lane). ACSL4 was used as a loading control for MAM fraction. (B–D) Enrichment of IP3Rs in the MAM fraction was quantified relative to total ER levels (MAM + ER signals). Data were normalized to the WT condition and plotted as mean ± S.D. from $n = 5$ biological replicates for IP3R3 and IP3R2, and $n = 3$ biological replicates for IP3R1/2/3. Statistical analysis with unpaired t-test in all cases. (E) GRP75 and VDAC1 proteins levels were analyzed in the MAM fractions from WT HEK293 and STIM1-KO HEK293 cells. ACSL4 was used as a loading control. (F, G) Data were normalized to the WT condition and plotted as mean ± S.D. from $n = 4$ biological replicates. Statistical analysis with unpaired t-test, $p = 0.0288$ in (F), $p = 0.0452$ in (G). (H) Methanol-fixed HEK293 cells were incubated with the indicated primary antibodies (rabbit anti-VDAC1 and/or mouse anti-IP3R1/2/3) as well as the DNA probes rabbit-PLUS and mouse-MINUS. As negative controls, cells were incubated with individual primary antibodies. Representative images for all the conditions are shown. Scale bar = 10 µm. (I) Quantification of the assay from (H), with the number of cells analyzed indicated in parentheses. The black line represents the mean of the data. Statistical analysis with unpaired t-test, ****$p < 0.0001$, *$p = 0.0139$. Source data are available online for this figure.

WT and KO cells. Mitochondrial mass was assessed by quantitative PCR of the mitochondrial gene *MT-ND2* (or *ND2*) as well as by flow cytometry using MitoTracker Green or MitoTracker Red staining (Fig. EV2H,I).

Since the reduction in VDAC1 and GRP75 protein levels could affect the interaction between IP3R and VDAC1, potentially underlying the deficiency in mitochondrial $Ca^{2+}$ uptake, we assessed IP3R-VDAC1 interactions by PLA. Interestingly, the number of contacts between these proteins was slightly higher in STIM1-KO cells than in WT cells (Fig. 4H,I), suggesting that the impairment in $Ca^{2+}$ transfer is not due to insufficient formation of IP3R-VDAC1 complexes.

Taken together, these results show that the absence of STIM1 leads to a significant decrease in mitochondrial $Ca^{2+}$ uptake following IP3R-mediated $Ca^{2+}$ release from the ER, which, in turn, affects basal $[Ca^{2+}]_{mit}$. This effect is not due to reduced $Ca^{2+}$ release from the ER or a decrease in IP3R-VDAC1 interactions. However, the lower levels of VDAC1 and GRP75 in the MAM fraction of STIM1-deficient cells suggest a dysfunction in the IP3R-GRP75-VDAC1 axis and an increase in the number of ER-contacts as a plausible compensatory mechanism to achieve a minimal $Ca^{2+}$ uptake by mitochondria.

## Impact of STIM1 deficiency on mitochondrial bioenergetics

Since mitochondrial matrix $[Ca^{2+}]$ regulates the activity of TCA cycle dehydrogenases, we next analyzed mitochondrial bioenergetics in WT and STIM1-KO cells using Agilent Seahorse technology. The oxygen consumption rate (OCR) was measured during the sequential addition of respiration modulators (oligomycin, FCCP, antimycin A/rotenone) (Figs. 5A–D and EV3). STIM1-KO cells exhibited a significant decrease in maximal respiration (Fig. 5B), ATP production (Fig. 5C), and spare respiratory capacity (Fig. 5D) compared to WT cells, indicating a slight but significant disruption in mitochondrial function. The spare respiratory capacity is particularly interesting, as STIM1-KO cells not only produce lower ATP levels in the resting state but also exhibited a limited ability to respond to increased energy demand. Other parameters, such as non-mitochondrial oxygen consumption, basal respiration, or proton leak, showed no changes between WT and KO cells (Fig. EV3). Because ATP production ultimately depends on the electron transport chain (ETC), we analyzed the levels of key mitochondrial ETC proteins by immunoblot (Fig. 5E). STIM1-KO cells showed reduced levels of NDUFB8, an accessory subunit of the NADH dehydrogenase complex (complex I),

compared to WT cells (Fig. 5F), while no significant differences were observed for proteins in the remaining complexes (II–V) (Fig. 5G–J). Given that NDUFB8 is essential for the stability and activity of complex I (Piekutowska-Abramczuk et al, 2018), and that complex I plays a central role in ATP generation, both the deficiency of NDUFB8 and the reduction in $[Ca^{2+}]_{mit}$ likely contribute to the bioenergetic impairment observed in STIM1-KO cells.

Mitochondrial oxidative phosphorylation is a major source of reactive oxygen species (ROS), particularly through complex I (Koopman et al, 2010). It is well-established that decreased cellular levels and activity of complex I lead to increased ROS production (Koopman et al, 2007; Verkaart et al, 2007), which may reflect an imbalance between energy production and demand (Okoye et al, 2023). Given the reduced complex I expression in STIM1-KO cells, we hypothesized that these cells would exhibit elevated ROS levels. To test this, we assessed oxidative stress in STIM1-deficient cells using the CellROX™ sensor. STIM1-KO cells displayed a significant increase in ROS levels compared to WT cells (Fig. 5K), likely due to the reduction in complex I expression.

Because steady-state mitochondrial matrix $[Ca^{2+}]$ levels regulate the TCA cycle, we monitored the phosphorylation status of one of its control points, the pyruvate dehydrogenase complex (Patel et al, 2014). This complex is activated by dephosphorylation mediated by a $Ca^{2+}$-dependent phosphatase activity and inhibited by phosphorylation at several residues, including Ser293 (Patel et al, 2014). We assessed the levels of phospho-Ser293 pyruvate dehydrogenase α1 (PDH) and found increased phosphorylation in STIM1-KO cells (Fig. 5L,M), suggesting a partial slowdown of the complex and, consequently, of the TCA cycle in the absence of STIM1. As a negative control, HEK293 cells were treated in parallel with 20 mM dichloroacetate for 24 h to inhibit PDH kinase 1, thereby establishing the basal phosphorylation level of the complex. As a consequence of the slowed TCA cycle, STIM1-KO cells exhibited a lower mitochondrial membrane potential (Fig. 5N,O), which may contribute to the observed reduction in ATP production.

Taken together, these findings suggest that the increased number of ER-mitochondria contacts observed in STIM1-KO cells does not correlate with an improvement in mitochondrial calcium handling MAM function. Instead, the data point to a compensatory effect underlying the increased ER-mitochondria contacts in the absence of STIM1. This compensatory response is likely driven by the need to maintain a minimal $Ca^{2+}$ flux to sustain TCA cycle activity. Collectively, these results highlight the essential role of STIM1 in maintaining proper MAM function.

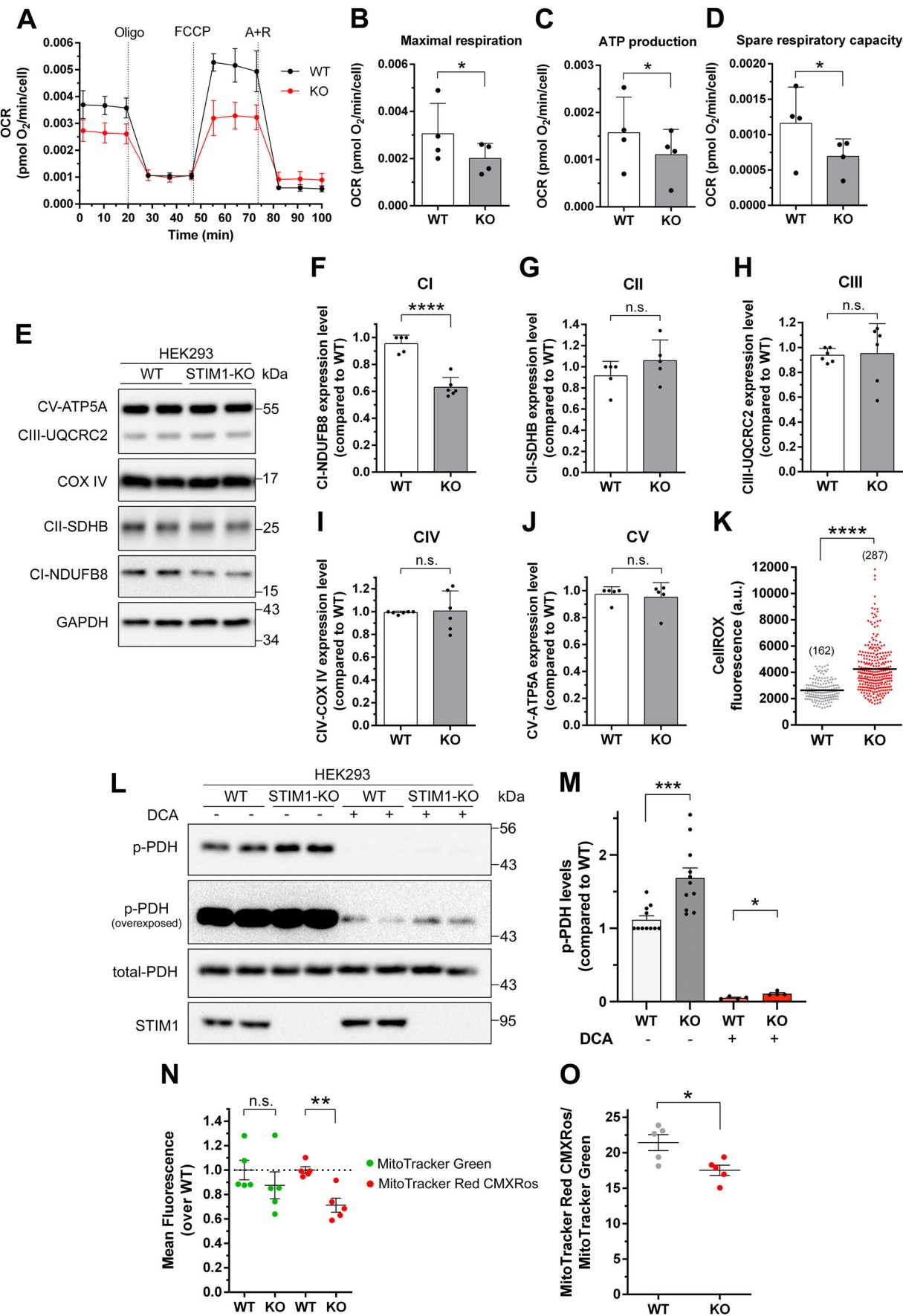

**Figure 5.  Mitochondrial fitness deficiency in the absence of STIM1.**

(A) Representative whole-cell Seahorse plot of the oxygen consumption rate (OCR, pmol $O_2$/min/cell). Wild-type and STIM1-KO U2OS cells were cultured on a Seahorse microplate for 24 h (20,000 cells/well). OCR was monitored under basal conditions and after the sequential addition of oligomycin, FCCP, and antimycin A + rotenone (A + R). After the assay, cells were counted in each well, and OCR values were normalized accordingly. Data are presented as mean ± S.D. from a single experiment, with $n = 3$ WT (black line) and $n = 5$ KO (red line) technical replicates. (B–D) Quantification of mitochondrial parameters derived from Seahorse assay analysis. The average values from at least 3 technical replicates across 4 independent experiments were evaluated. Statistical analysis was performed using a paired t-test ratio to assess the differences between pairs. (B) Maximal respiration: the maximum rate measurement after FCCP injection minus the non-mitochondrial respiration. $*p = 0.0356$. Data are plotted as mean ± S.D. (C) ATP production: the last rate measurement before oligomycin injection minus the minimum rate measurement after oligomycin injection. $*p = 0.0472$. Data are plotted as mean ± S.D. (D) Spare respiratory capacity: the maximal respiration rate minus the basal respiration. $*p = 0.0446$. Data are plotted as mean ± S.D. (E) Whole cell lysates from WT HEK293 and STIM1-KO HEK293 cells were obtained, and samples containing 20 µg protein were assessed by immunoblotting. Expression levels of proteins from all mitochondrial complexes were analyzed. NDUFB8 (CI), SDHB (CII), UQCRC2 (CIII), and ATP5A (CV) proteins were evaluated using total OXPHOS commercial antibody cocktail, while a specific anti-COX IV antibody was used for CIV assessment. GAPDH expression served as a loading control. (F–J) Quantification of the expression levels of all mitochondrial proteins shown in (E). The average of $n \geq 2$ technical replicates from 3 biological replicates, normalized to WT condition, is plotted in all cases. Statistical analysis with unpaired t-test, $****p < 0.0001$. Data are plotted as mean ± S.D. (F) NDUFB8 protein (CI). (G) SDHB protein (CII). (H) UQCRC2 protein (CIII). (I) COX IV protein (CIV). (J) ATP5A protein (CV). (K) HEK293 cells were cultured on collagen-coated coverslips for 48 h, incubated with 5 µM CellROX™ for 30 min, washed, and maintained in HBSS for 10 min. A minimum of 4 images were acquired per experiment. Three independent experiments were conducted. Fluorescence intensity per cell after background subtraction. The number of cells evaluated is indicated in parentheses, and the mean of the data represented by the black line. Statistical analysis with unpaired t-test, $****p < 0.0001$. (L) HEK293 (WT and STIM1-KO) cell lysates were assessed for phospho-Ser293 pyruvate dehydrogenase α1 (p-PDH) and total PDH by immunoblotting. As a negative control, cells were treated with 20 mM dichloroacetate (DCA) for 24 h before lysis. (M) Quantification of p-PDH levels from 4 independent experiments ($n = 11$ total replicates) for non-DCA-treated samples, and 2 independent experiments for DCA-treated samples ($n = 4$ total replicates). Statistical analysis with unpaired t-test, $***p = 0.0007$ and $*p = 0.0165$. Data are plotted as mean ± S.E.M. (N) HEK293 cells were labeled with MitoTracker Green FM, MitoTracker Red CMXRos, and DAPI for viability assessment. Fluorescence recordings were normalized to the values of WT cells and plotted accordingly. Data from 5 independent experiments are shown as scatter dot plot. Statistical analysis with unpaired t-test, $**p = 0.0019$. (O) The ratio between the signal from the potential-sensitive MitoTracker Red CMXRos over the signal from potential-insensitive MitoTracker Green is shown as a readout of mitochondrial potential, normalized to mitochondrial mass. Data from 5 independent experiments are shown as scatter dot plot. Statistical analysis with unpaired t-test, $*p = 0.0192$. Source data are available online for this figure.

## STIM1 conformation governs its interaction with GRP75

We hypothesized that STIM1 cooperates with GRP75 to regulate $Ca^{2+}$ transfer from the ER to mitochondria. Supporting evidence includes: (i) STIM1 co-precipitates with GRP75, (ii) STIM1 is an ER-luminal $Ca^{2+}$ sensor, and (iii) $Ca^{2+}$ transfer is a severely affected MAM function in the absence of STIM1. According to this hypothesis, STIM1 acts as a $Ca^{2+}$ sensor at transfer sites, where it interacts with GRP75. Because STIM1 undergoes a conformational change from a closed to an extended (or open) state upon $[Ca^{2+}]_{ER}$ depletion, we propose that this conformational switch regulates its association with GRP75 and modulates $Ca^{2+}$ transfer. To test this, we analyzed the STIM1-GRP75 interaction under two conditions that induce the extended form of STIM1: (1) ER $Ca^{2+}$ release triggered by Tg or ATP+CCh, and (2) with the STIM1(R429C) mutant, which adopts a constitutively open conformation as a constitutively active STIM1 (Maus et al, 2015). First, we assessed co-precipitation of STIM1 and GRP75 following Tg-induced ER $Ca^{2+}$ depletion. After 2-min of 1 µM Tg treatment in $Ca^{2+}$-free HBSS, STIM1-GFP was immunoprecipitated, and GRP75 co-precipitation was evaluated by immunoblotting (Fig. 6A). The results revealed a significant decrease in STIM1-GRP75 interaction following ER $Ca^{2+}$ depletion, suggesting that GRP75 preferentially interacts with the closed form of STIM1 (Fig. 6B). Total GRP75 and STIM1-GFP protein levels from WCL are shown in Fig. EV4A. To rule out the possibility that this decrease was due to SOCE inhibition rather that STIM1 conformation, we reintroduced $Ca^{2+}$ into the assay medium in the presence of Tg, thereby enabling SOCE. However, SOCE did not restore the interaction (see Fig. 6A,B), indicating that the ER $Ca^{2+}$ filling state, rather than SOCE, modulates STIM1-GRP75 binding.

To further explore whether a decreased STIM1-GRP75 interaction following Tg-mediated ER $Ca^{2+}$ depletion correlates with an increase in ER-mitochondria contacts, we performed VAPB-

PTPIP51 PLAs to quantify these contacts upon Tg treatment. As shown in Fig. 6C,D, Tg treatment significantly increased ER-mitochondria contacts, supporting the hypothesis of a compensatory mechanism.

We also examined the STIM1-GRP75 interaction during ATP +CCh-induced ER depletion. Given its rapid kinetics (see Fig. 3E), we performed assays at 15 and 30 s of stimulation. The interaction was reduced by 90% compared to the resting state (Fig. 6E,F). Total GRP75 and STIM1-GFP levels are shown in Fig. EV4B. Stimulation of $Ca^{2+}$ release from the ER with ATP+CCh in a $Ca^{2+}$-free medium not only triggered the dissociation of STIM1-GRP75, but also increased ER-mitochondria contacts, as determined using ddFP (Fig. 6G), a strategy previously described in Fig. 2D,E. The dissociation of STIM1-GRP75 following store depletion occurred in parallel with the release of both STIM1 and GRP75 from ER-mitochondria contact sites, as quantification of their levels in MAMs revealed a significant decrease of both proteins under these conditions (Fig. 6H,I).

These findings indicate that GRP75 primarily interacts with the closed form of STIM1 and reinforce the hypothesis that the $Ca^{2+}$-sensing activity of STIM1 is critical for its cooperation with GRP75 in regulating ER-mitochondria $Ca^{2+}$ transfer.

To isolate the effects of STIM1 conformation from ER $Ca^{2+}$ filling state, we used the STIM1 (R429C) mutant, which remains in a constitutively open conformation. This loss-of-function mutation impairs STIM1 puncta formation and its interaction with ORAI1, resulting in SOCE abrogation, as previously described (Maus et al, 2015). We confirmed that STIM1-KO cells expressing Flag-STIM1(R429C) exhibited abrogated SOCE while maintaining normal $[Ca^{2+}]_{ER}$ levels and that STIM1(R429C) is unable to form puncta in response to store depletion (Fig. EV4C–E). Next, we evaluated the STIM1(R429C)-GRP75 interaction via co-precipitation. The R429C mutation significantly reduced STIM1-GRP75 binding (Fig. 6J,K), suggesting that the $Ca^{2+}$-sensing

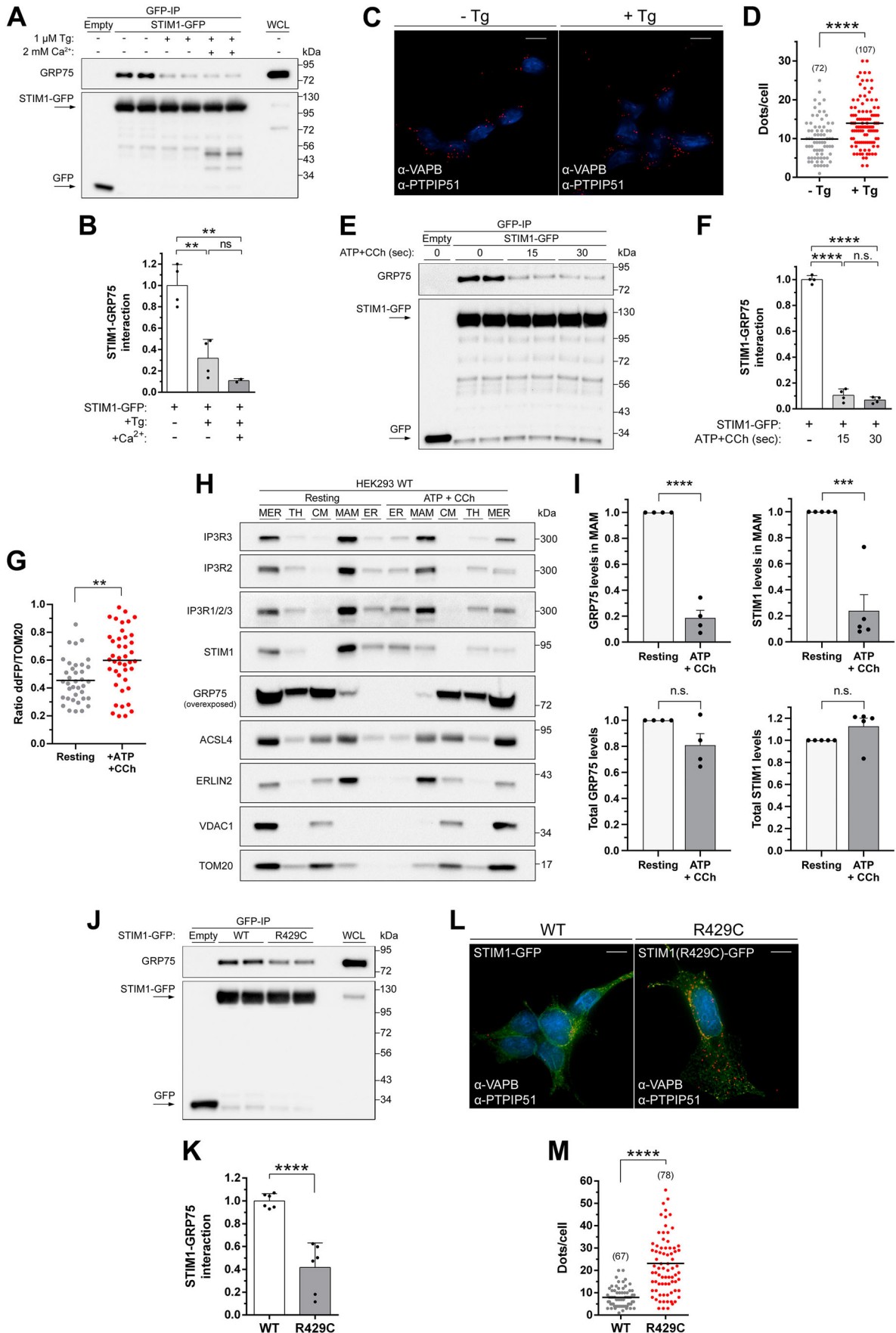

**Figure 6.  STIM1 conformation governs its interaction with GRP75.**

(A) STIM1-KO HEK293 cells inducibly expressing STIM1-GFP, or GFP only (empty GFP-vector) as a control, were washed twice in $Ca^{2+}$-free HBSS and incubated with 1 µM Tg in $Ca^{2+}$-free HBSS for 2 min. Cells from the SOCE-free condition (-$Ca^{2+}$) were then lysed. Cells from the SOCE-allowed condition (+$Ca^{2+}$) were incubated for an additional 2 min with 1 µM Tg + 2 mM $CaCl_2$ before lysis. Immunoprecipitation of GFP-tagged proteins was performed using 1 mg WCL, and the co-precipitation of GRP75 was evaluated by immunoblotting. A fraction of WCL from the untreated condition (3 µg) was loaded as a positive control. Levels of immunoprecipitated GFP were assessed as a loading control. Levels of total GRP75 and total STIM1-GFP were evaluated from WCL by immunoblotting (30 µg protein/lane) and are shown in Fig. EV4A. (B) Quantification of co-precipitated GRP75 (from 4 independent experiments for resting and Tg-treated samples, 2 experiments for SOCE-allowed samples). Statistical analysis with unpaired t-test. Control vs Tg, $p = 0.0021$; Control vs Tg+$Ca^{2+}$, $p = 0.0038$. Data are plotted as mean ± S.D. (C) Evaluation of ER-mitochondria contacts after Tg treatment by PLA. Wild-type HEK293 cells were washed twice with $Ca^{2+}$-free HBSS and incubated with 1 µM Tg in this medium for 2 min, as in (A). Cells were then fixed and incubated with mouse anti-VAPB and rabbit anti-PTPIP51 antibodies, as well as with the DNA probes rabbit-PLUS and mouse-MINUS. The untreated control was fixed with methanol directly from the culture medium, without any washing step. The interaction of proteins (red dots) was analyzed under fluorescence microscopy. Representative images are shown in the figure. Scale bar = 10 µm. Negative controls are shown in Fig. 2. (D) Quantification of the interactions detected by PLA, with the number of cells evaluated in parentheses. The mean of the data is represented by the black line. Statistical analysis with unpaired t-test, ****$p < 0.0001$. (E) Analysis of the STIM1-GRP75 interaction during ER $Ca^{2+}$ release triggered by ATP+CCh. Cells were washed twice with HBSS, then twice with $Ca^{2+}$-free HBSS, and incubated with 100 µM ATP + 100 µM CCh in $Ca^{2+}$-free HBSS for 15–30 s before lysis. Immunoprecipitation of GFP-tagged proteins was performed using 1 mg WCL, and co-precipitated GRP75 was analyzed by immunoblotting. For the untreated control (0-s condition), lysis was conducted immediately after the first two washes with HBSS. Levels of immunoprecipitated GFP were analyzed as a loading control. Total levels of GRP75 in WCL were assessed by immunoblotting (30 µg protein/lane) and are shown in Fig. EV4B. (F) Quantification of co-precipitated GRP75 from 2 independent experiments and 4 technical replicates. Statistical analysis with unpaired t-test, ****$p < 0.0001$ and $p = 0.2266$ (n.s.). Data are plotted as mean ± S.D. (G) Quantification of ER-mitochondria contacts in U2OS cells expressing Mito-GA and ER-B, as described in Fig. 2D. Reconstitution of green fluorescence was normalized to the total mitochondrial mass, revealed with anti-TOM20 + AlexaFluor 594 (red) immunofluorescence. Data show the mean (black line) from two independent experiments. Statistical analysis with unpaired t-test, $p = 0.0014$. (H) HEK293 cells were washed twice with HBSS, then twice with $Ca^{2+}$-free HBSS, and incubated with 100 µM ATP + 100 µM CCh in $Ca^{2+}$-free HBSS for 30 s before isolation of MER, TH, CM, MAM, and ER. For the untreated control (0-s condition), lysis was conducted immediately after the first two washes with HBSS. STIM1 and GRP75 levels were analyzed by immunoblotting. ACSL4 and ERLIN2 were assessed as positive controls for MAMs, while VDAC1 and TOM20 were assessed as positive controls for MER and CM. In all cases, 7 µg protein was loaded in each lane. (I) Top panels: Quantification of GRP75 and STIM1 in MAMs was performed using data from 4 independent subcellular fractionation experiments. For normalization, GRP75 levels in MAMs were normalized to levels in TH, and STIM1 levels in MAMs were normalized to levels in ER. Bottom panels: Quantification of STIM1 and GRP75 levels in TH. Data are normalized to resting conditions. Statistical analysis with unpaired t-test. $p < 0.0001$ for GRP75 in MAM, $p = 0.0003$ for STIM1 in MAM, $p = 0.0757$ for total GRP75, and $p = 0.1220$ for total STIM1. Data are plotted as mean ± S.E.M. (J) Co-immunoprecipitation assays were performed using 1 mg WCL from STIM1-KO HEK293 cells inducibly expressing STIM1-GFP, STIM1(R429C)-GFP, or GFP only (empty GFP vector) as a control. The co-precipitation of GRP75 was evaluated by immunoblotting. A fraction of WCL (3 µg) from cells expressing STIM1-GFP was loaded as a positive control. Blots are representative of 3 biological replicates. Levels of immunoprecipitated GFP were assessed as a loading control. Total levels of GRP75 from WCL were analyzed by immunoblotting (30 µg protein/lane) and are shown in Fig. EV4F. (K) Quantification of the co-precipitation data ($n = 6$ technical replicates from 3 biological replicates) is shown. Data were normalized to immunoprecipitated GFP levels and plotted as mean ± S.D. Statistical analysis with unpaired t-test, $p < 0.0001$. (L) STIM1-KO HEK293 cells inducibly expressing STIM1-GFP or STIM1(R429C)-GFP, labeled as WT or R429C in the graph, were fixed and incubated with mouse anti-VAPB and rabbit anti-PTPIP51 antibodies and the DNA probes rabbit-PLUS and mouse-MINUS. Scale bar = 10 µm. Negative controls are shown in Fig. 2. (M) Quantification of the number of interactions per cell. The number of cells analyzed is indicated in parentheses, and the mean of data is represented by the black line. Statistical analysis with unpaired t-test, $p < 0.0001$. Source data are available online for this figure.

function of STIM1 regulates this interaction through its conformational change, consistent with the results observed after triggering store depletion with Tg or with ATP+CCh. Total GRP75 and STIM1-GFP levels are shown in Fig. EV4F. Additionally, PLA quantification of VAPB-PTPIP51 contacts in cells expressing STIM1(R429C) revealed a greater number of ER-mitochondria contacts compared to WT STIM1-expressing cells (Fig. 6L,M), further supporting the compensatory mechanism observed in Fig. 6C,D and G.

In summary, the decrease in STIM1 and GRP75 levels in MAMs following store depletion (Fig. 6H,I), together with the reduced co-precipitation observed under the same experimental conditions (Fig. 6A,B,E,F), and the reduced interaction with the R429C mutant (Fig. 6J,K), leads us to two conclusions: (i) ER $Ca^{2+}$ depletion governs the localization of STIM1-GRP75, and (ii) GRP75 interaction is dependent on the conformational change in STIM1.

## Evaluation of the interaction domain of STIM1 with GRP75

To further investigate the role of STIM1-GRP75 interaction in $Ca^{2+}$ transfer to mitochondria, we analyzed the interaction between GRP75 and GFP-tagged STIM1 constructs with deletions in the C-terminal domain. Constructs lacking amino acids 235-442, 443-550, and 551-685 are referred to as mutants 1, 2, and 3 in a

schematic representation in Fig. EV5A, and were generated in a previous study by our group (Sanchez-Lopez et al, 2024). The results revealed that the 551-685 region is critical for GRP75 binding, as no co-precipitation of GRP75 was observed for mutant 3 (Fig. EV5B,C). To refine this region, we generated additional constructs with shorter deletions: mutant 4 (Δ551-642), mutant 5 (Δ551-611), mutant 6 (Δ582-642), mutant 7 (Δ643-677), mutant 8 (Δ668-674), and mutant 9 (Δ672-685) (schematic in Fig. 7A). GRP75 co-immunoprecipitation analysis indicated that amino acids 551-642 in STIM1 likely contain an essential sequence or motif for GRP75 binding (mutant 4, Fig. 7A–C). Further analysis of mutants 5 and 6 showed that deletion of 551-611 (mutant 5) abolished GRP75 binding, whereas deletion of residues 582-642 (mutant 6) resulted in only a partial decrease of co-immunoprecipitation (Fig. 7A–C). Total GRP75 and STIM1-GFP levels are shown in Fig. EV5D,E. These findings highlighted the 551-611 sequence as the minimal region containing a putative domain or motif required for GRP75 interaction. We studied this interaction by PLA, which confirmed the loss of interaction between GFP-tagged STIM1(Δ551-611) (mutant 5) and GRP75 (Fig. 7D,E). Notably, deletion of the 551-611 region (mutant 5) did not impair SOCE activation or puncta formation (Fig. EV5F,G). Because deficiencies in the STIM1-GRP75 interaction may increase MAM contacts to compensate for reduced $Ca^{2+}$ transfer, we measured VAPB-PTPIP51 contacts using PLA in STIM1-KO

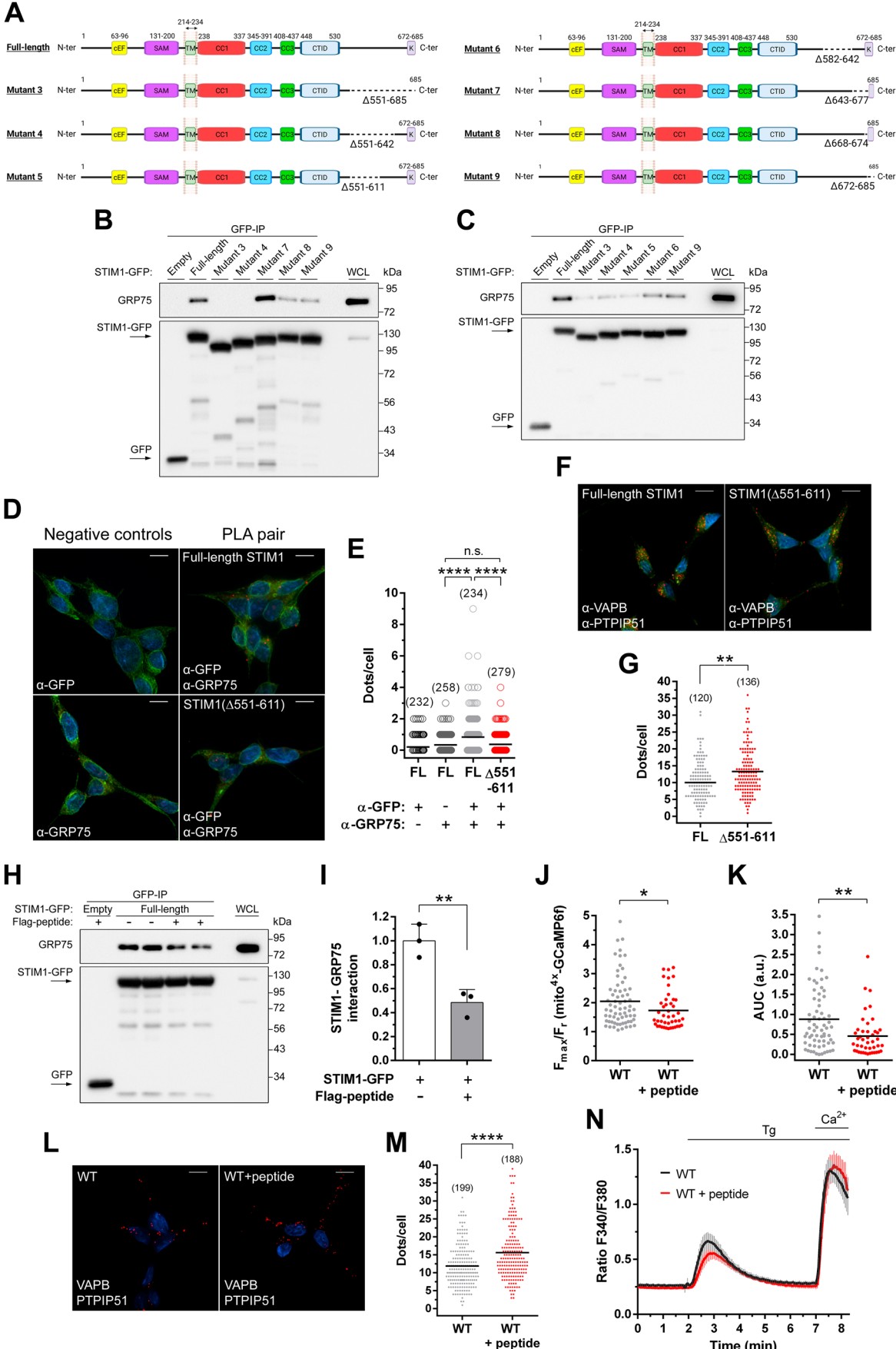

◄  **Figure 7.  Evaluation of the interaction domain of STIM1 with GRP75.**

(A) Schematic representation of STIM1 mutants with deletions in the 551-685 region. The specific sequences deleted between the amino acids 551 and 685 for each of the generated STIM1 mutants are shown. This diagram was created with BioRender.com. (B) Immunoprecipitation of GFP-tagged proteins was performed using 1 mg WCL from STIM1-KO HEK293 inducibly expressing STIM1-GFP, STIM1(Δ551-685)-GFP (mutant 3), STIM1(Δ551-642)-GFP (mutant 4), STIM1(Δ643-677)-GFP (mutant 7), STIM1(Δ668-674)-GFP (mutant 8), STIM1(Δ672-685)-GFP (mutant 9), or GFP (empty vector) as a control. Co-precipitation of GRP75 was assessed by immunoblotting. WCL (3 μg) from cells expressing STIM1-GFP was loaded as a positive control. Total immunoprecipitated GFP was used as a loading control. Total levels of STIM1 and GRP75 in WCL (30 μg protein/lane) were analyzed by immunoblotting (Fig. EV5D). (C) Immunoprecipitation of GFP-tagged proteins was performed using 1 mg WCL from STIM1-KO HEK293 inducibly expressing STIM1-GFP, STIM1(Δ551-685)-GFP (mutant 3), STIM1(Δ551-642)-GFP (mutant 4), STIM1(Δ551-611)-GFP (mutant 5), STIM1(Δ582-642)-GFP (mutant 6), STIM1(Δ672-685)-GFP (mutant 9), or GFP only as a control. Other conditions as stated for (B). Total levels of STIM1 and GRP75 in WCL were analyzed by immunoblotting (30 μg protein/lane) (Fig. EV5E). (D) STIM1-KO HEK293 cells inducibly expressing full-length (FL) STIM1-GFP or STIM1(Δ551-611)-GFP were fixed and incubated with rabbit anti-GRP75 and/or sheep anti-GFP antibodies. Negative controls consisted of cells incubated with single primary antibodies. Scale bar = 10 μm. (E) Quantification of the interactions per cell detected by PLA in (D). The number of cells evaluated is indicated in parentheses, and the mean is represented by the black line. Statistical analysis with unpaired t-test, $p < 0.0001$ and $p = 0.5354$ (n.s.). (F) STIM1-KO HEK293 cells expressing STIM1-GFP (full-length, FL) or STIM1(Δ551-611)-GFP were incubated with a rabbit anti-PTPIP51 and mouse anti-VAPB antibody for a PLA assay. Negative controls are shown in Fig. 2. Scale bar = 10 μm. (G) Quantification of the PLA assay (number of red dots per cell) shown in (F). The number of cells evaluated is indicated in parentheses, and the black line represents the mean of the data. Statistical analysis with unpaired t-test, $p = 0.0016$. (H) HEK293 cells expressing STIM1(full-length)-GFP were transfected for transient expression of a Flag-tagged STIM1 peptide encompassing residues 551-611. After 24 h of transfection, STIM1-GFP was immunoprecipitated and co-precipitated GRP75 was assessed by immunoblotting. Total levels of GRP75 and STIM1-GFP from WCL are shown in Fig. EV5J. (I) Quantification of co-precipitated GRP75 from 2 independent experiments and 3 technical replicates. Statistical analysis with unpaired t-test, $p = 0.0072$. Data are plotted as mean ± S.D. (J) HEK293 cells stably and inducibly expressing mito$^{4x}$-GCaMP6f were transiently transfected for the expression of mCherry-tagged STIM1 peptide corresponding to residues 551-611. The peptide was tagged with mCherry to select transfected cells, and the ratios F/Fr were calculated in mCherry-expressing cells only. The graph shows Fmax/Fr after the addition of ATP+CCh in $Ca^{2+}$-free medium. Statistical analysis with unpaired t-test, $p = 0.0407$. Dots represent data from individual cells. (K) Analysis of the area under the curve (AUC) of the fluorescence profile after the ATP+CCh stimulus. A total of 16 independent experiments were performed in control (WT) cells ($n = 67$ cells) and 17 experiments for cells expressing the peptide (551-611)-mCherry ($n = 44$ cells). The mean of the data is represented by the black line. Statistical analysis with unpaired t-test, $p = 0.0021$. (L) HEK293 cells inducibly expressing the Flag-tagged STIM1(551-611)-peptide or HEK293 control cells (with no peptide expression) were fixed and incubated with anti-VAPB and anti-PTPIP51 antibodies and PLA reagents, as in (F). Scale bar = 10 μm. (M) Quantification of VAPB-PTPIP51 interactions per cell from (L). The number of cells evaluated is indicated in parentheses, and the black line represents the mean of the data. Statistical analysis with unpaired t-test, $p < 0.0001$. (N) Wild-type HEK293 cells inducibly expressing the Flag-STIM1(551-611) peptide (red line) and loaded with fura-2 were assessed for the extension of SOCE and compared to cells with no peptide expression (black line). A total number of 51 (WT) and 49 (WT + peptide) cells were analyzed from 3 independent experiments. Data are plotted as mean ± S.D. Source data are available online for this figure.

HEK293 cells inducibly expressing either full-length STIM1-GFP or STIM1(Δ551-611)-GFP. As predicted, the Δ551-611 mutant exhibited increased MAM contacts compared to controls (Fig. 7F,G).

All previous results strongly suggest that the 551-611 region is critical for the interaction with GRP75, and thereby for preserving $Ca^{2+}$-related MAM function. To further support these findings, we measured mitochondrial $Ca^{2+}$ uptake following disruption of the STIM1-GRP75 interaction. In this experiment, we used a Flag-STIM1(551-611) peptide, which retained the ability to bind GRP75, as demonstrated by a co-immunoprecipitation assay (Fig. EV5H,I). This peptide was expressed in HEK293 cells that inducibly expressed STIM1–GFP. Subsequent immunoprecipitation of STIM1-GFP, followed by immunoblot analysis, revealed that expression of the Flag-STIM1(551-611) peptide reduced the interaction between STIM1 and GRP75 by approximately 50% (Figs. 7H,I and EV5J).

Most importantly, peptide expression also inhibited mitochondrial $Ca^{2+}$ uptake (Fig. 7J,K) following ER $Ca^{2+}$ release triggered by ATP+CCh. This reduction in mitochondrial $Ca^{2+}$ uptake occurred without a significant alteration in ER $Ca^{2+}$ concentration (Fig. EV5K), and without any detectable change in the amount of $Ca^{2+}$ released after stimulation (Fig. EV5L,M). Likewise, expression of the STIM1(551-611) peptide triggers an increase in ER-mitochondria contacts, measured as VAPB-PTPIP51 proximity by PLA (Fig. 7L,M), and without affecting SOCE (Fig. 7N). These findings not only confirm the importance of the STIM1-GRP75 interaction in ER-mitochondria dynamics and function but also underscore the essential role of the 551-611 region.

## Discussion

A previous study suggested the presence of STIM1 at the ER-mitochondria interface through proteomic analyses (Cho et al, 2020). STIM1 has also been detected in MAM fractions by immunoblotting when used as an ER marker, though this observation was not further explored (Filadi et al, 2016). In addition, a set of STIM1 interactors can be accessed on the BioGrid website, which includes mitochondrial proteins such as COX14, FIS1, and SLC25A46, all of which are localized to the outer mitochondrial membrane. Despite these findings, no specific function has been assigned to STIM1 in these regions.

Since MAM are lipid-raft-like domains enriched in cholesterol and sphingolipids (Area-Gomez et al, 2012; Hayashi and Fujimoto, 2010), it is possible that the specific localization of STIM1 involves interaction with cholesterol. Three lipid-binding regions have been characterized in STIM1. The first, the C-terminal polybasic domain (residues 671-685), binds phosphorylated forms of phosphatidyli-nositol in the PM (Bhardwaj et al, 2013; Liou et al, 2007). Recently, Palty's group identified the SOAR region as binding both mono- and poly-phosphorylated inosites via the lysine residues in the 382-386 segment (Cohen et al, 2023). Additionally, a cholesterol-binding domain within the SOAR region has been described, distinct from the inosites-binding site (Pacheco et al, 2016). Given the evidence that STIM1 localizes to MAM, the possibility that cholesterol-mediated targeting plays a role in this localization highlights the importance of investigating whether a cholesterol-binding domain exists within the transmembrane region of STIM1. While this was beyond the scope of the current study, confirming

such a mechanism would enable future research into the impact of disrupting the localization of STIM1 in MAM while preserving its function in other cellular processes, providing a deeper insight into its specific role in MAM.

Our study found that the knockout of STIM1 significantly increases MAM contacts. This contrasts with previous reports where no differences in MAM contacts were observed between STIM1-KO and WT cells (Henke et al, 2012). To explain this discrepancy, it is important to consider the methodologies used, especially the intermembrane distances. The functional relevance of MAM contacts is closely tied to the precise distances between ER and mitochondria membranes, which can vary by tissue, cell type, and whether the ER contains ribosomes. Distances typically range from 10 to 80 nm, with 30 nm often being the upper functional limit (Scorrano et al, 2019). Specific functions are optimized at certain distances—e.g., $Ca^{2+}$ transfer is most efficient at 15–25 nm, while lipid transfer occurs most efficiently around 10 nm (Csordás et al, 2018; Giacomello and Pellegrini, 2016). Larger separations are linked to processes like autophagosome formation, and there are speculation about "functionally dormant" MAM regions that become active when contact distances decrease (Giacomello and Pellegrini, 2016). Thus, observing the mere juxtaposition of organelles is insufficient to define a true contact site. Functionally relevant MAM sites are characterized by binding forces from protein-protein or protein-lipid interactions. Various techniques quantify ER-mitochondria contacts, each with its limitations. Methods such as FRET, super-resolution microscopy, electron microscopy, and PLA differ in resolution and sensitivity (Scorrano et al, 2019). The choice of technique and inclusion threshold significantly impacts the detection of functionally relevant interactions. In our study, we used PLA targeting VAPB and PTPIP51, well-characterized tethering proteins essential for ER-mitochondria interactions (Arjona et al, 2023; Gomez-Suaga et al, 2022; Stoica et al, 2014). PLA yields signal only when target proteins are within 40 nm, a range that aligns with functional MAM distances, reinforcing the physiological relevance of our findings. In contrast, the previous study that reported no differences in MAM contacts in WT and STIM1-KO cells used transmission electron microscopy (TEM) with a broader inclusion criterion of up to 100 nm (Henke et al, 2012). Our PLA-based method allowed for 3-dimensional analysis, identifying MAM dynamics that broader, plane-limited methods, such as TEM, might miss. Notably, we did not observe increased MAM contacts in STIM2-KO cells, suggesting a specific role for STIM1 in MAM activity. In addition to PLA, we used ddFP to monitor fluorescence reconstitution when two monomers (Mito-GA and ER-B) come within a distance of 10–30 nm (Miner et al, 2024). This approach yielded results very similar to those obtained with PLA—namely, an increase in ER-mitochondria contacts in response to either the absence of STIM1 or ER depletion triggered by stimulation of the phosphoinositide pathway.

Our analysis of $Ca^{2+}$ dynamics revealed near-complete inhibition of ER-to-mitochondria $Ca^{2+}$ transfer upon IP3R stimulation in STIM1-KO cells. This transfer is mediated by the IP3R-GRP75-VDAC complex, although its function is modulated by additional proteins. While mitochondrial $Ca^{2+}$ entry depends mainly on MCU, it can also occur through MCU-independent mechanisms (Bround et al, 2024). Our analysis of IP3R, VDAC1, GRP75, and MCU expression in STIM1-KO cells showed no significant changes compared to WT cells. This contrasts with studies in chicken DT40

B lymphocytes, where the knockout of STIM1 or ORAI1 decreased MCU expression via $Ca^{2+}$-dependent regulation of the CREB transcription factor (Shanmughapriya et al, 2015). The discrepancy may reflect cell-type-specific mechanisms, with HEK293 cells possibly using alternative pathways to maintain MCU expression. Additionally, while previous work showed reduced mitochondrial $Ca^{2+}$ uptake in STIM1-KO cells under SOCE-stimulating conditions (Shanmughapriya et al, 2015), our measurement in $Ca^{2+}$-free medium support the existence of a SOCE-independent mechanism for ER-to-mitochondria $Ca^{2+}$ transfer. Despite unchanged expression levels of $Ca^{2+}$ transfer complex components, we observed reduced GRP75 and VDAC1 abundance in MAM fractions in STIM1-KO cells (Fig. 4E–G), though this did not affect the IP3R-VDAC1 interaction (Fig. 4H,I), suggesting that the alteration in $Ca^{2+}$ transfer is not due to insufficient contact between these proteins. Together, these findings lead us to hypothesize that STIM1 directly regulates $Ca^{2+}$ transfer at MAMs, with its absence likely causing compensatory increases in ER-mitochondria contacts.

Consistent with impaired $Ca^{2+}$ transfer in STIM1-KO cells, we observed decreased $[Ca^{2+}]_{mit}$ levels, as previously reported (Pascual-Caro et al, 2020; Wilson et al, 2022). However, other studies using Rhod-2 reported opposite results (Henke et al, 2012). Rhod-2, while commonly used, has limitations: it requires careful control of temperature and concentration for proper mitochondrial localization and can impact mitochondrial morphology and function, complicating accurate measurements (Fonteriz et al, 2010; Kosmach et al, 2021). As a potentiometric dye, Rhod-2 depends on mitochondrial membrane potential $(\Delta\Psi_m)$ for accumulation, and STIM1-KO cells, which have reduced $\Delta\Psi_m$ (Pascual-Caro et al, 2020; Wilson et al, 2022), may make Rhod-2 less reliable for comparing $[Ca^{2+}]_{mit}$ between STIM1-KO and WT cells. In such cases, a mitochondrial-targeted chameleon probe is preferable, as it avoids these issues (Antigny et al, 2009). In our study, we addressed this by using the genetically encoded $Ca^{2+}$ indicator 4mtD3cpv, providing more accurate and precise measurements of $[Ca^{2+}]_{mit}$.

Furthermore, STIM1-KO cells exhibited a significant reduction in ATP production, along with decreased spare respiratory capacity and maximal respiration, indicating compromised mitochondrial efficiency. These bioenergetic deficits are likely linked to reduced $[Ca^{2+}]_{mit}$, as mitochondrial $Ca^{2+}$ uptake is crucial for activating key enzymes involved in ATP production (Denton and McCormack, 1986). In fact, associated with the reduced $Ca^{2+}$ transfer between the ER and mitochondria, we observed that STIM1-KO cells exhibit increased phosphorylation levels of the pyruvate dehydrogenase complex—a complex known to be activated by $Ca^{2+}$-dependent dephosphorylation—which explains the lower mitochondrial membrane potential observed in STIM1-KO cells. The observed decrease in complex I levels in STIM1-KO cells should exacerbate the deficiency in ATP generation, as this complex is a major contributor to mitochondrial ATP production. Under normal conditions, electron transport through complexes I, III, and IV generates a proton gradient across the inner mitochondrial membrane, driving ATP synthesis at complex V. In contrast to glucose oxidation, which favors complex I via higher NADH/$FADH_2$, fatty acid oxidation leads to a lower NADH/$FADH_2$ ratio and increased electron flow through complex II. This can cause over-reduction of ubiquinone, triggering reverse electron transport from reduced (CoQ) back to complex I (Scialò et al, 2017),

producing ROS that degrade complex I (Guarás et al, 2016). Previous studies show STIM1-KO cells prefer fatty acid oxidation, whereas increased STIM1 expression favors glucose oxidation (Liu et al, 2023; Wilson et al, 2022). This metabolic shift may underlie the complex I deficiency and could result from reduced $Ca^{2+}$ transfer to mitochondria, as STIM1-KO cells might compensate for impaired $Ca^{2+}$-dependent pyruvate oxidation. A similar metabolic rewiring has been observed in MCU-KO cells and amyotrophic lateral sclerosis (ALS) models (Gherardi et al, 2020; Larrea et al, 2025).

Supporting the idea that STIM1 regulates $Ca^{2+}$ transfer in MAMs, our study found that GRP75 levels in MAMs decreased when ER $Ca^{2+}$ was depleted with ATP+CCh. Consistent with this observation, we detected a reduction in the STIM1-GRP75 interaction after ATP+CCh or Tg treatment. Moreover, this interaction did not recover after SOCE induction in the presence of Tg, suggesting that the STIM1-GRP75 interaction depends on $[Ca^{2+}]_{ER}$. Derived from this last conclusion, our results also indicate that the STIM1-GRP75 interaction occurs when STIM1 is in a closed conformation, which is the one observed when the ER is filled with $Ca^{2+}$. This conformation transitions to a constitutively open conformation with the R429C mutation. However, this mutant does not exhibit multimerization, as originally described (Maus et al, 2015) and shown in this study, nor does it activate SOCE. Therefore, this mutant is ideal for separating the effects produced by SOCE activation from those dependent on STIM1 multimerization. The fact that STIM1(R429C) shows reduced interaction with GRP75, along with an increase in the number of ER-mitochondria contacts, simulating the phenotypic response to a decrease in $Ca^{2+}$ trafficking between the ER and mitochondria, further supports the idea that the closed conformation of STIM1 is preferential for binding to GRP75 (Fig. 8).

Finally, we found that STIM1 interacts with GRP75 through the 551-611 region, which is not necessary for SOCE activation. This region can be described as an intrinsically disordered region (IDR), meaning it lacks a stable secondary or tertiary structure. Because of this, IDRs can interact with a wide variety of partners, allowing molecules with IDRs to exhibit functional diversity. However, the study of protein-protein interactions mediated by an IDR is challenging. In this work, we have shown that using a competition strategy with a peptide containing the 551-611 sequence from STIM1, the STIM1-GRP75 interaction can be diminished, leading to a reduction in $Ca^{2+}$ transfer. This reduction in $Ca^{2+}$ transfer occurs in the absence of any alteration in SOCE, ER $Ca^{2+}$ levels, or the capacity to release $Ca^{2+}$ in response to IP3R stimulation.

It is important to note that this strategy, as reducing the STIM1-GRP75 interaction could be a valid approach in cases where there is a $Ca^{2+}$ overload in the mitochondria, such as in certain degenerative diseases, including Alzheimer's disease (Jadiya et al, 2019) and amyotrophic lateral sclerosis (reviewed in Kawamata and Manfredi (2010)).

In summary, we not only addressed the presence of STIM1 in MAM but also discovered that it interacts with GRP75, a key protein that facilitates $Ca^{2+}$ transfer from the ER to mitochondria. This novel interaction suggests a mechanistic role for STIM1 that extends beyond its canonical function in SOCE, providing a new framework for understanding the contribution of STIM1 to $Ca^{2+}$ homeostasis. The STIM1-GRP75 interaction was found to be dependent on the $Ca^{2+}$-sensing ability of STIM1, indicating that

this protein may modulate $Ca^{2+}$ transfer to the mitochondria in response to fluctuations in $[Ca^{2+}]_{ER}$. These findings represent the first direct evidence of the contribution of STIM1 to mitochondrial $Ca^{2+}$ regulation and suggest a previously unrecognized regulatory role for STIM1 at the MAM interface.

## Methods

**Reagents and tools table**

| Reagent/Resource | Reference or Source | Identifier or Catalog Number |
|---|---|---|
| **Experimental Models** | | |
| HEK-293 cells (*H. sapiens*) | ThermoFisher Scientific | R78007 |
| U2OS cells (*H. sapiens*) | PMID: 27784791 | Dr. Gopal Sakopta (University of Dundee) |
| **Recombinant DNA** | | |
| pcDNA5/FRT/TO-GFP | MRCPPU Reagents and Services | N/A |
| pcDNA5/FRT/TO-Flag | MRCPPU Reagents and Services | N/A |
| pcDNA5/FRT/TO-Flag-STIM1 | https://doi.org/10.1093/nar/gkae001 | N/A |
| pcDNA5/FRT/TO-STIM1-GFP | https://doi.org/10.1093/nar/gkae001 | N/A |
| pcDNA5/FRT/TO-STIM1(Δ235-442)-GFP | https://doi.org/10.1093/nar/gkae001 | N/A |
| pcDNA5/FRT/TO-STIM1(Δ443-550)-GFP | https://doi.org/10.1093/nar/gkae001 | N/A |
| pcDNA5/FRT/TO-STIM1(Δ551-685)-GFP | https://doi.org/10.1093/nar/gkae001 | N/A |
| pcDNA5/FRT/TO-STIM1(Δ551-642)-GFP | This study | N/A |
| pcDNA5/FRT/TO-STIM1(Δ551-611)-GFP | This study | N/A |
| pcDNA5/FRT/TO-STIM1(Δ582-642)-GFP | This study | N/A |
| pcDNA5/FRT/TO-STIM1(Δ643-677)-GFP | This study | N/A |
| pcDNA5/FRT/TO-STIM1(Δ668-674)-GFP | This study | N/A |
| pcDNA5/FRT/TO-STIM1(Δ672-685)-GFP | This study | N/A |
| pcDNA5/FRT/TO-STIM1-mCherry | This study | N/A |
| pBABED-(Hygro)-STIM1-mCherry | This study | N/A |
| pcDNA5/FRT/TO-STIM1-ddFP-B | This study | N/A |
| pcDNA5/FRT/TO-Flag-STIM1(551-611) | This study | N/A |
| pcDNA5/FRT/TO-STIM1(551-611)-GFP | This study | N/A |
| pcDNA5/FRT/TO-STIM1(551-611)-mCherry | This study | N/A |

| Reagent/Resource | Reference or Source | Identifier or Catalog Number |
|---|---|---|
| pcDNA5/FRT/TO-STIM(R429C)-GFP | This study | N/A |
| pcDNA5/FRT/TO-Flag-STIM1(R429C) | This study | N/A |
| pcDNA5-FRT/TO-Flag-mito$^{4\times}$-GCaMP6f | This study | N/A |
| ER-B | Addgene | 209870 |
| Mito-GA | Addgene | 209865 |
| pCMV ER-GCamP6-210 | Addgene | 86919 |
| pcDNA-4mtD3cpv | Addgene | 36324 |
| pOG-44 Flp-recombinase vector | ThermoFisher Scientific | V600520 |
| **Antibodies** | | |
| Mouse anti-ACSL4 (0.2 µg/ml in TBS-T + 10% non-fat milk for immunoblot, IB) | Santa Cruz Biotechnology | sc-365230 |
| Rabbit anti-COX IV (1:1000 in TBS-T 0.1% + 5% BSA, for IB) | Cell Signaling Technology | 4850 |
| Rabbit anti-ERLIN2 (1:1000 in PBS + 1% non-fat milk, for IB) | Cell Signaling Technology | 2959 |
| Mouse anti-GAPDH (0.03 µg/ml in TBS-T + 10% non-fat milk, for IB) | Santa Cruz Biotechnology | sc-32233 |
| Mouse anti-GFP (0.3 µg/mL in TBS-T + 10% non-fat milk, for IB) | Proteintech | 66002-1-Ig |
| Sheep anti-GFP (1:400 in blocking buffer, for immunofluorescence, IF) | MRCPPU Reagents and Services | S268B |
| Rabbit anti-GRP75 (1:1000 in TBS-T 0.1% + 5% BSA, for IB. 1:100 in blocking buffer, for IF) | Cell Signaling Technology | 3593 |
| Rabbit anti-GRP75 (1:600 in blocking buffer, for IF) | Proteintech | 14887-1-AP |
| Normal Rabbit IgG | Cell Signaling Technology | 2729 |
| Rabbit anti-IP3R1 (1:1000 in 10% non-fat milk, for IB) | Dr. Jan B. Parys (KU Leuven) | RBT-03 |
| Mouse anti-IP3R2 (0.4 µg/ml in TBS-T + 10% non-fat milk, for IB) | Santa Cruz Biotechnology | sc-398434 |
| Mouse anti-IP3R3 (1:1000 in 10% non-fat milk, for IB) | BD Transduction Laboratories | BD-610312 |
| Mouse anti-IP3R1/2/3 (0.4 µg/ml in TBS-T + 10% non-fat milk, for IB. 1:60 in blocking buffer, for IF) | Santa Cruz Biotechnology | sc-377518 |
| Rabbit anti-MCU (1:1000 in TBS-T 0.1% + 5% BSA, for IB) | Cell Signaling Technology | 14997 |
| Goat anti-Mouse IgG HRP-labelled (1:10,000 in TBS-T + 10% non-fat milk) | ThermoFisher Scientific | A16072 |
| Mouse anti-total OXPHOS (1:2000 in TBS-T 0.1% + 5% BSA, for IB) | Abcam | ab110413 |
| Rabbit anti-p38 MAPK (1:1000 in TBS-T 0.1% + 5% BSA, for IB) | Cell Signaling Technology | 9212 |
| Rabbit anti-phospho (Ser293) pyruvate dehydrogenase α1 (1:1000 in TBS-T 0.1% + 5% BSA, for IB) | Cell Signaling Technology | 37115 |

| Reagent/Resource | Reference or Source | Identifier or Catalog Number |
|---|---|---|
| Rabbit anti-PTPIP51 (0.17 µg/ml in TBS-T + 5% non-fat milk, for IB. 1:300 in blocking buffer, for IF) | Proteintech | 20641-1-AP |
| Rabbit anti-pyruvate dehydrogenase α1 (1:1000 in TBS-T 0.1% + 5% BSA, for IB) | Cell Signaling Technology | 3205 |
| Donkey anti-rabbit IgG Alexa Fluor™ 594 (1:1000 in blocking buffer, for IF) | ThermoFisher Scientific | A-21207 |
| Goat anti-rabbit IgG HRP-labelled (1:10,000 in TBS-T + 10% non-fat milk) | ThermoFisher Scientific | A16096 |
| Donkey anti-sheep IgG HRP-labelled (1:10,000 in TBS-T + 10% non-fat milk) | ThermoFisher Scientific | A16041 |
| Rabbit anti-STIM1 (1:1000 in TBS-T 0.1% + 5% BSA, for IB) | Cell Signaling Technology | 5668 |
| Sheep anti-STIM1(614-628) (1 µg/ml in TBS-T + 5% non-fat milk, for IB. 1:20 in blocking buffer, for IF) | University of Dundee, Division of Signal Transduction Therapy | S241D |
| Rabbit anti-TOM20 (1:5000 in TBS-T + 5% non-fat milk, for IB. 1:250 in blocking buffer, for IF) | Proteintech | 11802-1-AP |
| Mouse anti-VAPB (1:2000 in TBS-T + 10% non-fat milk, for IB) | Proteintech | 66191-1-Ig |
| Mouse anti-VDAC1 (1 µg/ml in TBS-T + 10% non-fat milk, for IB) | Santa Cruz Biotechnology | sc-390996 |
| Rabbit anti-VDAC1 (1:300 in blocking buffer, for IF) | Proteintech | 55259-1-AP |
| Rabbit anti-VDAC1/2 (1:1000 in TBS-T + 10% non-fat milk, for IB) | Proteintech | 10866-1-AP |
| Rabbit anti-VDAC3 (0.5 µg/ml in TBS-T + 10% non-fat milk, for IB) | Invitrogen | PA5-51156 |
| Rabbit anti-VDAC3 (0.6 µg/ml in TBS-T + 10% non-fat milk, for IB) | Proteintech | 55260-1-AP |
| **Oligonucleotides and other sequence-based reagents** | | |
| STIM2-KO sense guide oligo: 5'-(G)ATATAACGATTGAGGATTTATGG | Integrated DNA Technologies | N/A |
| STIM2-KO anti sense guide oligo: 5'-(G)GTTTATCTTCTCTGTGCAGATGG | Integrated DNA Technologies | N/A |
| **Chemicals, enzymes and other reagents** | | |
| Acrylamide | National Diagnostics | EC-852 |
| ApaI | New England Biolabs | R0114 |
| ATP | Sigma | A7699-1G |
| BamHI | New England Biolabs | R0136 |
| Benzamidine hydrochloride hydrate | Sigma-Aldrich | B6506-25G |
| Blasticidin | InvivoGen | BLL-42-05 |
| Bovine serum albumin (BSA) | Sigma | A3059-100G |
| Ca$^{2+}$-free HBSS | ThermoFisher Scientific | 14175 |
| Carbachol (CCh) | Tocris | 2810 |
| CellROX™ Orange reagent | Invitrogen | C10443 |

| Reagent/Resource | Reference or Source | Identifier or Catalog Number |
|---|---|---|
| Clarity Max™ Western ECL Substrate | Bio-Rad Laboratories | 1705062 |
| Clarity™ Western ECL Substrate | Bio-Rad Laboratories | 170-5061 |
| Collagen type I, rat tail | Corning | 354236 |
| Coomassie Protein Assay Reagent | ThermoFisher Scientific | 1856210 |
| DAPI | ThermoFisher Scientific | 62248 |
| DH5α *E. coli* | ThermoFisher Scientific | 18265017 |
| DMEM | Gibco™ | 11995065 |
| DMSO | Sigma | D8418-50ML |
| DL-Dithiothreitol (DTT) | Sigma | D9779-5G |
| Doxycycline | Sigma | D9891-1G |
| Duolink In Situ Detection Reagents | Merck Life Science | DUO92008 |
| Dynabeads Protein A magnetic beads | Invitrogen | 10001D |
| DynaMag rack | Invitrogen | 12321D |
| Fetal bovine serum (FBS) | Gibco™ | A56709-01 |
| Fura 2-AM | Sigma-Aldrich | 344905 |
| Gelatin from cold water fish skin | Sigma-Aldrich | G7765-250ML |
| GeneJet Plasmid Miniprep Kit | ThermoFisher Scientific | K0503 |
| GFP-trap agarose beads | Proteintech | gta |
| HBSS | ThermoFisher Scientific | 14025 |
| Hoechst 33258 | Merck Life Science | B-1155 |
| Hydromount | National Diagnostics | HS-106 |
| Hygromycin B Gold | InvivoGen | HGG-45-01 |
| Ionomycin | MedChemExpress | HY-13434 |
| Lipofectamine 2000 | Invitrogen | 11668030 |
| Miniprep Kit (Monarch® Spin Plasmid) | New England Biolabs | T1110L |
| Miniprep Kit (QIAprep®) | Qiagen | 27104 |
| MitoTracker Green FM | Invitrogen | M7514 |
| MitoTracker Red CMXRos | Invitrogen | M7512 |
| Mitotracker Red FM | Invitrogen | M22425 |
| MOPS | Millipore | MPMOPS |
| Mycoplasma Detection kit | BioTools | 4542 |
| Nitrocellulose Blotting Membrane | Amersham™ Protran™ | 10600001 |
| *Not*I | New England Biolabs | R0189 |
| NuPAGE-LDS sample buffer | Invitrogen | NP0007 |
| PBS | Fisher BioReagents | BP3994 |
| Percoll® | GE Healthcare | 17-0891-02 |
| Phenylmethylsulphonyl fluoride (PMSF) | Sigma | P7626-1G |
| PLA probe anti-goat MINUS | Merck Life Science | DUO92006 |

| Reagent/Resource | Reference or Source | Identifier or Catalog Number |
|---|---|---|
| PLA probe anti-mouse MINUS | Merck Life Science | DUO92004 |
| PLA probe anti-rabbit PLUS | Merck Life Science | DUO92002 |
| Polyethylenimine (PEI) | Polysciences, Inc | 23966 |
| Precast gels (4–12% Bis-Tris) | Millipore | MP41G12 |
| Protein A/G Plus agarose beads | ThermoFisher Scientific | 20423 |
| Protein Lobind Tubes | Eppendorf™ | 022431081 |
| Seahorse XFp Cell Mito Stress Test Kit | Agilent Technologies | 103010-100 |
| Sodium dichloroacetate | Sigma-Aldrich | 347795-10 G |
| SSC buffer | Merck Life Science | S6639 |
| Thapsigargin | Tocris | 67526-95-8 |
| Trypsin-EDTA | Gibco™ | 25300-062 |
| XF DMEM Medium | Agilent Technologies | 103575-100 |
| *Xho*I | New England Biolabs | R0146 |
| **Software** | | |
| GraphPad Prism 10 | https://www.graphpad.com/ | N/A |
| NIS-Elements AR software 4.40.00 | https://www.microscope.healthcare.nikon.com/es_EU/ | N/A |
| Image Lab™ 5.1 | Bio-Rad Laboratories | N/A |
| SnapGene 8.2 | https://www.snapgene.com/ | N/A |
| Wave 2.6.1.53 | Agilent Technologies | N/A |
| **Other** | | |
| Falcon Tissue Culture Dish | Corning | 353003 |
| BioLite 35 mm Tissue Culture Dish | Thermo Fisher Scientific | 130180 |
| EM-CCD Hamamatsu C9100-02 digital camera | https://www.hamamatsu.com/jp/en.html | N/A |
| CytoFlex S V4-B2-Y4-R3 | Beckman Coulter | N/A |
| DH-40i micro-incubation platform | Warner Instruments | N/A |
| Inverted Nikon Ti-E microscope | https://www.microscope.healthcare.nikon.com/es_EU/ | N/A |
| Seahorse XFp Extracellular Flux Analyzer | Agilent Technologies | N/A |
| Seahorse XFp Cell Culture Miniplates | Agilent Technologies | 103025-100 |
| ChemiDoc XRS+ | Bio-Rad Laboratories | N/A |
| LightCycler 480 Platform | Roche | N/A |

## Cell lines

Flp-In™ HEK293 T-REx cells were purchased from ThermoFisher Scientific. U2OS cells were a kind gift from Dr. Gopal Sapkota

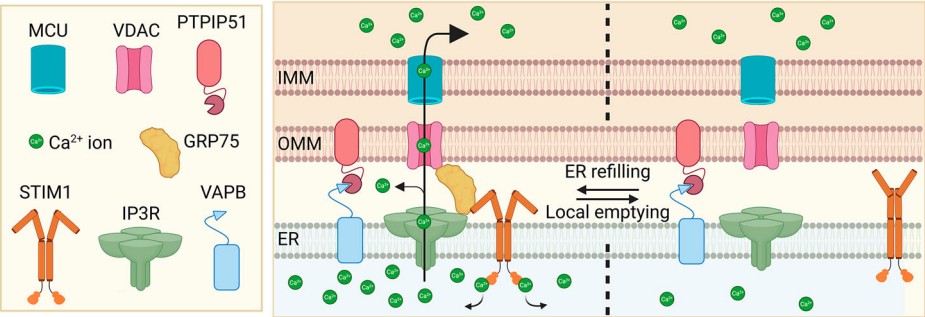

**Figure 8. Proposed mechanism regulating Ca²⁺ trafficking between the ER and mitochondria.**

In this model, GRP75 recruitment requires STIM1 in its closed conformation, reflecting an ER fully replete with intraluminal Ca²⁺. Under these conditions, GRP75 couples to IP3Rs and VDAC to promote Ca²⁺ transfer. This efflux, however, induces a localized and partial ER Ca²⁺ depletion, triggering a conformational change in STIM1 and the dissociation of GRP75 from the IP3R–VDAC complex and MAMs, thereby terminating Ca²⁺ transfer and preventing mitochondrial Ca²⁺ overload.

(University of Dundee), and they were modified to be stably transfected with the Flp-In™ system. HEK293 and U2OS cells edited by CRISPR/Cas9 to knock-out STIM1 or ORAI1 expression (STIM1-KO or ORAI1-KO cells) were generated and characterized elsewhere (Sanchez-Lopez et al, 2024; Lopez-Guerrero et al, 2017b). Cell lines were grown in Dulbecco's modified Eagle's medium (DMEM) supplemented with 10% fetal bovine serum.

Mycoplasma contamination was monitored with the mycoplasma Gel Detection Kit and all cell stocks used in this work were tested and found negative for this contamination.

## DNA constructs

The vector for the inducible expression of STIM1-GFP or Flag-STIM1 was generated by inserting the cDNA of the human STIM1 isoform 2 (RefSeq accession NM_003156.4) into the BamHI-NotI sites of the pcDNA5/FRT/TO-GFP or the pcDNA5/FRT/TO-Flag vector, both described in Sanchez-Lopez et al (2024). The construct for the expression of STIM1-ddFP-B was generated by amplifying the 678-bp fragment corresponding to the ddFP-B monomer from the ER-B construct (Addgene plasmid #209870) by PCR, inserting flanking 5′-NotI and 3′-XhoI restriction sites. This fragment was inserted into the pcDNA5/FRT/TO-STIM1-GFP construct, to replace the GFP coding sequence. The construct for the expression of the STIM1(551-611) peptide labeled with Flag was generated by PCR inserting BamHI + NotI sites into the pcDNA5/FRT/TO-Flag vector (University of Dundee). Similarly, this PCR product was inserted into the pcDNA5/FRT/TO-GFP and pcDNA5/FRT/TO-mCherry vector (University of Dundee) to express the peptide fused to GFP or mCherry.

The vector for the inducible expression of the Ca²⁺ sensor mito⁴ˣ-GCaMP6f was generated by inserting the BamHI-ApaI fragment from the construct CMV-mito⁴ˣ-GCaMP6f (Addgene #127870) into the BamHI-ApaI sites of the pcDNA5-FRT/TO-FLAG vector. The construct for the stable expression of STIM1-mCherry in STIM1-KO U2OS cells was made by inserting the STIM1-mCherry cDNA (STIM1 RefSeq accession NM_003156.4) into BamHI-NotI sites of the pBABED-Hygro vector (#DU37127 from the University of Dundee).

Subsequent site-directed mutagenesis was performed by overlap extension PCR. DNA constructs were verified by DNA sequencing at the Sequencing Facility of the Universidad de Extremadura.

Transfection of cells with DNA constructs was performed with 1.5 μg plasmid DNA per 10-cm dish and polyethyleneimine in serum-containing medium, except for the generation of stable cell lines with pcDNA/FRT/TO vectors, where 0.5 μg construct + 4.5 μg of the pOG44 Flp recombinase expression vector was used.

## Generation of stable cell lines and CRISPR/Cas9 genome editing

Flp-In T-REx HEK293 cells, able to inducibly express tagged STIM1, were generated as described elsewhere (Pozo-Guisado et al, 2010, 2013). Prior to assays, cells were treated with 1 μg/ml doxycycline for 22–24 h to induce expression of tagged STIM1.

Genome editing to knock-out the expression of the human STIM2 gene in Flp-In T-REx HEK293 cells (Ensembl accession ENSG00000109689) was performed following established procedures used for STIM1 locus in other cell lines (Sanchez-Lopez et al, 2024; Lopez-Guerrero et al, 2017b, 2017a; Pascual-Caro et al, 2018). To knock-out STIM2 expression, exon 3 was selected as the CRISPR target site and the guide pair described in the Reagents and Tools were identified using the Sanger Institute CRISPR webtool (https://wge.stemcell.sanger.ac.uk//find_crisprs). Sequencing data revealed a 19-bp + 8-bp deletion (two alleles), leading to the occurrence of new translational stop codons, thus confirming the successful KO of the STIM2 locus. The characterization of this cell line is shown in Fig. EV1F,G.

## Mitochondrial DNA detection

Mitochondrial DNA was extracted from equal amounts of cellular pellets using QIAprep® Miniprep kit (Quispe-Tintaya et al, 2013). Equal volumes of DNA elution were used to determine the levels of the specific mitochondrial gene ND2 using the TaqMan Gene Expression probe Hs02596874_g1 in a LightCycler 480 platform (Roche). Absolute Ct values were calculated from 3 independent experiments.

## Cell fractionation and sample preparation

### Whole cell lysates

Cells were lysed using the following buffer: 50 mM Tris-HCl (pH 7.5), 1 mM EGTA, 1 mM EDTA, 1 mM DTT, 1% (v/v) Nonidet P40 (NP-40), 1 mM sodium orthovanadate, 50 mM NaF, 5 mM sodium pyrophosphate, 0.27 M sucrose, 0.1% (v/v) 2-mercaptoethanol, 1 mM benzamidine, and 0.1 mM phenylmethylsulphonyl fluoride (PMSF) (whole cell lysate buffer). Samples were lysed with 0.5 ml of ice-cold lysis buffer/10-cm dish, sonicated with $4 \times 10$-s pulses with a setting of 40% amplitude using a Branson digital sonifier, and then centrifuged at $20,000 \times g$ for 20 min (4 °C). Protein concentration was determined using the Coomassie Protein Assay Reagent.

### Isolation of mitochondria-associated ER membranes

The isolation of MAMs was performed as described previously (Montesinos and Area-Gomez, 2020). Briefly, cells from 10-cm dishes were scrapped in cold PBS and concentrated by centrifugation at $3000 \times g$ for 15 min. Cells were then washed in the following homogenization buffer (HB): 225 mM mannitol, 25 mM HEPES-KOH (pH = 7.4), 1 mM EGTA, and supplemented with 1 mM DTT, 1 mM benzamidine, and 0.1 mM PMSF. Cells were homogenized with 5 strokes in a Teflon-pestle grinder, followed by 10 additional strokes after diluting with 1 ml HB, yielding the total homogenate fraction. The homogenate was clarified with 2 sequential centrifugations at $900 \times g$ for 10 min. The resulting supernatant was then centrifuged at $9000 \times g$ for 15 min to pellet the mitochondrial fraction. On one hand, the resulting supernatant from this centrifugation contains ER and cytosol which are separated as pellet and supernatant, respectively, by centrifuging at $100,000 \times g$ for 1 h at 4 °C. On the other hand, the initial pellet was resuspended in 0.5 ml of HB and centrifuged at $9000 \times g$ for 10 min, resulting in the pelleted crude mitochondria fraction. This fraction was resuspended in a final volume of 0.5 ml HB, loaded onto a 30% Percoll® gradient and centrifuged at $95,000 \times g$ for 30 min. This way, two phases were obtained: an upper band (phase 1) containing MAM fraction and ER-attached mitochondria (MER); and a lower band (phase 2) containing free mitochondria. In this work, phase 2 was not analyzed due to insufficient yield for immunoblot assays. Phase 1 was diluted with 5 volumes of HB, vortexed for 1 min at maximum speed, and centrifuged at $8000 \times g$ for 10 min at 4 °C. The supernatant contained MAMs, and the pellet was MER. To extract MAM, the supernatant was centrifuged at $100,000 \times g$ for 1 h at 4 °C. The resulting "floating" clear pellet from this centrifugation was the MAM fraction. To extract MER, the pellet obtained after the centrifugation of phase 1 was diluted in 1.5 ml HB and centrifuged at $8000 \times g$. The new pellet obtained was the MER fraction.

## Immunoblot

Lysates (10–40 μg) were subjected to electrophoresis on polyacrylamide gels and subsequent electroblotting to nitrocellulose membranes. Membranes were blocked for 1 h at room temperature (RT) in blocking buffer: TBS-T (Tris-buffered saline buffer, pH 7.5, with 0.2% Tween-20) containing 10% (w/v) non-fat milk. Then the membranes were incubated overnight with the specific antibody (see Reagents and Tools Table), washed with TBS-T, and then incubated with anti-IgG horseradish peroxidase (HRP)-conjugated secondary antibodies (1:10,000 dilution in all cases) for 1 h at RT. Clarity Max™ Western ECL substrate was added to the membranes and the signal recorded with ChemiDoc XRS+ system. The recorded signal was quantified by volumetric integration using the Image Lab software.

## Co-immunoprecipitation analysis

To study the interaction of proteins with STIM1-GFP, tagged STIM1 was pulled-down with Chromotek GFP-Trap beads. Equilibrated GFP-Trap agarose beads (8 μl) were added to 1 mg of the clarified cell lysate. Lysates were incubated with the beads for 1 h at 4 °C with gentle shaking. The agarose beads were washed three times with 1 ml lysis buffer containing 0.15 M NaCl and twice with buffer A (50 mM Tris-HCl (pH 7.5), and 0.1 mM EGTA). Proteins were eluted from the GFP-Trap beads by the addition of 15 μl NuPAGE-LDS sample buffer + 2.5% β-mercaptoethanol to the beads. Eluted proteins were heated at 90–95 °C for 4 min and analyzed by immunoblot.

To immunoprecipitate endogenous STIM1 we followed the protocol described in Sanchez-Lopez et al (2024). Briefly, 2.5 mg HEK293 cell lysate was clarified with $3 \times 15$-min incubations of the lysate with 10 μl protein A/G-agarose beads and gentle rotation at 4 °C. Clarified lysates were incubated with 5 μl anti-STIM1 antibody (Cell Signaling Technology, #5668) in binding buffer (10 mM $NaH_2PO_4$ pH 7.0, 140 mM NaCl, 0.05% Triton X-100) overnight at 4 °C with gentle rotation. Then, 22 μl Dynabeads Protein A were added to the lysates and incubated for 1 h at 4 °C. Dynabeads were washed with binding buffer three times, precipitated with a DynaMag rack, and eluted in 20 μl NuPAGE-LDS sample buffer + 2.5% β-mercaptoethanol. Eluted proteins were analyzed by immunoblot. Negative controls in the absence of specific antibody were carried out with normal rabbit IgG.

## Proximity ligation assay (PLA)

Proximity ligation assay (PLA) was carried out by using the Duolink In Situ Detection Reagents, as previously performed (Sanchez-Lopez et al, 2024). Cells were grown on collagen-coated coverslips. Collagen type I was purchased from Corning (Corning, NY, USA). Cells were fixed in 100% methanol for 15 min at −20 °C and washed $3 \times$ with PBS. Following this, a 2-h blocking step was performed at 37 °C using a blocking solution (3% fish skin gelatin in PBS + 0.2% Tween-20). Subsequently, cells were incubated overnight in a humidified chamber at 4 °C with specific pairs of antibodies. The antibodies and dilutions used for PLA are indicated in the Reagent and Tools Table. Following antibody incubation, the coverslips were washed three times with PBS + 0.2% Tween-20, and incubated with 0.625 μl of each PLA probe (PLUS and MINUS) in a final volume of 50 μl of blocking solution for 1 h at 37 °C. For the PLA probes, we used PLA probe anti-rabbit PLUS, PLA probe anti-mouse MINUS, PLA probe anti-goat MINUS. After three washing steps with PBS + 0.2% Tween-20, coverslips were incubated with 1.25 μl ligase in $1 \times$ ligation buffer (final volume of 50 μl) for 30 min at 37 °C. The ligation mix was removed with three washing steps, and each coverslip was incubated with 0.625 μl polymerase in $1 \times$ amplification red solution (final volume 50 μl) for 100 min at 37 °C. All subsequent steps were performed in the dark to avoid

losing the fluorescent signal. To remove the amplification solution, coverslips were washed twice with $1 \times SSC$ buffer for 10 min each and once with $0.01 \times SSC$ buffer for 1 min. Finally, cells were incubated with 0.5 µg/ml Hoechst 33258 for 5 min, and coverslips were mounted onto glass slides using Hydromount. Imaging was performed on a Nikon Ti-E inverted epifluorescence microscope with a Plan Apochromat $100 \times$ (NA 1.45) oil immersion objective.

## Dimerization-dependent fluorescent proteins

The assay to determine ER–mitochondria contacts using dimerization-dependent fluorescent proteins was based on the constructs described in Miner et al (2024). The constructs for the expression of Mito-GA and ER-B were kindly provided by Sarah Cohen (Addgene plasmids #209865 and #209870, respectively). Plasmids were transfected into cells plated and grown on coverslips. For transfection, 3 µl Lipofectamine 2000 and 1 µg of each plasmid were used per 35 mm culture dish. After 24 h, cells were fixed with 4% paraformaldehyde and subjected to immunolocalization using a rabbit polyclonal anti-TOM20 antibody (Proteintech #11802-1-AP) followed by an AlexaFluor 594-conjugated secondary antibody. Nuclei were stained with Hoechst 33258, and cells were examined under a fluorescence microscope with settings for green fluorescence (ER-mitochondria contacts) and red fluorescence (mitochondrial marker). Images were recorded using an EM-CCD C9100 digital camera mounted on a Nikon Eclipse Ti-E inverted microscope. Wide-field microscopy was used to acquire a z-stack of images at 0.2 µm spacing. Images were subjected to deconvolution, and an equatorial section was taken for the calculation of the green:red fluorescence ratio as an indicator of the percentage of mitochondrial mass in contact with the endoplasmic reticulum.

In addition, to determine STIM1-mitochondria contacts, a new pcDNA5/FRT/TO-STIM1-ddFP-B construct was generated, here referred to as STIM1-ddFP-B, to be used in combination with Mito-GA. Using this construct, stable cell lines were generated with Flp-In T-REx HEK293 STIM1-KO cells, which are able to inducibly express STIM1-ddFP-B upon treatment with 1 µg/ml doxycycline for 22–24 h. At the time of doxycycline addition, i.e., 24 h prior to fluorescence microscopy image acquisition, cells were transfected with the plasmid encoding Mito-GA, as described above. Following fixation, cells were immunostained for TOM20. Quantification of STIM1-mitochondria contacts was performed in the same manner as described for ER-mitochondria contact quantification.

## Free Ca$^{2+}$ concentration measurements

### Cytosolic free Ca$^{2+}$ concentration

Cytosolic free Ca$^{2+}$ concentration ([Ca$^{2+}$]$_i$) was measured in fura-2-AM-loaded cells as described elsewhere (Sanchez-Lopez et al, 2024; Pozo-Guisado et al, 2010; Casas-Rua et al, 2015). Briefly, 170,000 cells were plated on collagen-coated round coverslips and cultured for 36–48 h. Thereafter, cells were loaded with 1 µM fura-2-AM for 45 min in culture medium, washed with Hank's balanced salt solution (HBSS) and placed on the DH-40i micro-incubation platform (Warner Instruments Holliston, MA, USA) of an inverted microscope Nikon Ti-E. Excitation fluorescence wavelengths were selected with 340/26 and 387/11 nm filters (Semrock, Rochester, NY, USA), and emission fluorescence with a 510/10 nm filter. Cells

were treated with 1 µM thapsigargin in Ca$^{2+}$-free HBSS with the following composition: 138 mM NaCl, 5.3 mM KCl, 0.34 mM Na$_2$HPO$_4$, 0.44 mM KH$_2$PO$_4$, 4.17 mM NaHCO$_3$, 2.2 mM Mg$^{2+}$, and EGTA 0.1 mM (pH 7.4). After triggering Ca$^{2+}$ store depletion with thapsigargin, store-operated Ca$^{2+}$ entry (SOCE) was monitored with the addition of 2 mM CaCl$_2$ to the assay medium, as described elsewhere (Pozo-Guisado et al, 2010, 2013).

### ER free Ca$^{2+}$ concentration

Endoplasmic reticulum free calcium concentration ([Ca$^{2+}$]$_{ER}$) was measured using the genetically-encoded Ca$^{2+}$ sensor ER-GCaMP6-210, described in de Juan-Sanz et al (2017). Transfected cells were monitored in Ca$^{2+}$-containing HBSS using the following settings: excitation fluorescence wavelengths were selected with 480/30 nm filters (Semrock), and emission fluorescence with a 535/40 nm filter. The emission of fluorescence was calibrated with the subsequent addition of 5 µM ionomycin + 5 mM EGTA followed by the addition of 5 µM ionomycin + 10 mM Ca$^{2+}$, as reported in (Pascual-Caro et al, 2020).

### Mitochondrial free Ca$^{2+}$ concentration

Steady-state mitochondrial free calcium concentration ([Ca$^{2+}$]$_{mit}$) was measured as described elsewhere (Pascual-Caro et al, 2018; Palmer and Tsien, 2006). Basically, [Ca$^{2+}$]$_{mit}$ was assessed in cells transfected with the genetically encoded Ca$^{2+}$ sensor 4mtD3cpv. CFP, YFP, and FRET efficiency between the two channels were measured with the dual CFP/YFP-2×2M-B filter set (Semrock). All measurements were performed in Ca$^{2+}$-containing HBSS for 4–5 min. Spectral unmixing (i.e., subtracting the bleed-through from one channel into another) was performed by determining the bleed-through coefficients (Pascual-Caro et al, 2018; Palmer and Tsien, 2006). The background corrected ratio, i.e., Ratio = (FRET$_{ROI}$ − FRET$_{background}$)/(CFP$_{ROI}$ − CFP$_{background}$) was converted to [Ca$^{2+}$]$_{mit}$ using a dissociation constant for Ca$^{2+}$ = 0.76 µM (Palmer and Tsien, 2006). The FRET/CFP ratio (R) was evaluated after calibrating the signal with the subsequent addition of 5 µM ionomycin + 5 mM EGTA (Rmin), followed by the addition of 5 µM ionomycin + 10 mM Ca$^{2+}$ (Rmax) (Pascual-Caro et al, 2018).

The study of Ca$^{2+}$ uptake by mitochondria upon ER Ca$^{2+}$ release was performed as described elsewhere (Pascual-Caro et al, 2020). [Ca$^{2+}$]$_{mit}$ dynamics were assessed in cells transfected with the genetically-encoded Ca$^{2+}$ sensor mito$^{4\times}$-GCaMP6f, due to its improved dynamic range and better signal-to-noise ratio when detecting peak responses (Ashrafi et al, 2020), allowing the fast monitoring of the Ca$^{2+}$ uptake by mitochondria. Excitation fluorescence wavelength was selected with a 485/10 filter (Semrock), and emission fluorescence with a 535/20 filter. The ratio Fmax/F0 was calculated, where F0 is the steady fluorescence value after placing the cells in Ca$^{2+}$-free HBSS, and Fmax corresponds to the fluorescence reached after stimulation with 100 µM ATP + 100 µM carbachol (CCh).

In all Ca$^{2+}$ concentration experiments, images were recorded using an EM-CCD C9100 digital camera attached to a Nikon Eclipse Ti-E inverted microscope. Illumination was performed with a xenon arc lamp. All measurements were performed in a DH-40i culture dish incubator, and the temperature was set at 36.5 ± 0.5 °C. In all cases, excitation and emission conditions were controlled by the NIS-Elements AR software.

## XF cell mito stress assay

Cell bioenergetics was evaluated in a Seahorse XFp Extracellular Flux Analyzer, as described elsewhere (Pascual-Caro et al, 2020). Cells were plated on Seahorse XFp plates (20,000 cells/per well). After 24 h, cells were washed with XF DMEM medium followed by a 1 h incubation at 37 °C in a $CO_2$-free incubator. After monitoring basal respiration, cells were sequentially treated with 1 μM oligomycin, 1 μM FCCP, and 1 μM antimycin A + 1 μM rotenone. Oxygen consumption rate (OCR) data were normalized to the total cell amount per well estimated by cell counting. The metabolic parameters of the assay were calculated with the Seahorse Wave software. In all cases, 3 technical replicates were recorded in every experiment, and the experiments were carried out with 3 biological replicates.

## Oxidative stress assay

For measuring oxidative stress, cells were plated on collagen-coated coverslips and after 48 h of cell culture, fluorescence measurements were conducted. Oxidative stress was analyzed after 12, 22, or 72 h of protein expression triggered by the addition of doxycycline to the cell culture medium. For all measurements, the medium was replaced with fresh medium containing 5 μM CellROX™ Orange reagent and cells were incubated for 30 min at 37 °C. Then, cells were washed twice with HBSS on a heated plate to ensure temperature stability at 37 °C and placed in a DH-40i micro-incubation platform of an inverted Nikon Ti-E microscope. After 10 min, 5 images were captured with a CCD Hamamatsu C9100-02 camera coupled to the fluorescence inverted microscope, using the G-2A filter block from Nikon, which includes a 510–560 nm excitation filter, a 565-nm dichroic mirror, and a 590-nm emission filter. Image analysis was performed using the NIS-Elements AR software. The fluorescence intensity of a minimum of 7 cells from each image was assessed, subtracting the background intensity in each case.

## Analysis of mitochondrial fitness

Determination of mitochondrial mass and mitochondria membrane potential was performed by flow cytometry in cells labeled for 15 min with 100 nM MitoTracker Green FM + Mitotracker Red FM for the analysis of mitochondrial mass, and 100 nM MitoTracker Green FM + 50 nM MitoTracker Red CMXRos in PBS + 2% FBS at 37 °C and 5% $CO_2$. After staining, DAPI (1 μg/μl) was added for viability assessment and tubes were kept in ice until analysis. Samples were analyzed using a CytoFlex S flow cytometer and at least 10,000 events per sample were collected.

## Statistical analysis of data

Statistical analyses between pairs of data groups were done with the GraphPad software using parametric unpaired t-test, except for Seahorse experiments, which were analyzed using parametric ratio paired t-test to address the differences between pairs. Immunoblot, and Seahorse data were plotted as mean ± S.D. in bar graphics. PLA, oxidative stress, mitochondrial mass, mitochondrial membrane potential, and $Ca^{2+}$ measurements data were represented in scatter dot plots. Differences between groups of data were taken statistically significant for $p < 0.05$. The $p$-values are represented as follows: (*) $p < 0.05$, (**) $p < 0.01$, (***) $p < 0.001$, (****) $p < 0.0001$.

## Data availability

This study has not generated data deposited in external repositories, but it has used data deposited via the PRIDE partner repository with the dataset identifier PXD041181.

The source data of this paper are collected in the following database record: biostudies:S-SCDT-10_1038-S44318-026-00700-8.

## Peer review information

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

## Acknowledgements

We thank the support of the Bioscience Applied Techniques Facility at the Universidad de Extremadura. This work was supported by Grants PID2020-112997-GBI00 and PID2023-147394NB-I00 (to FJM-R) funded by MCIN/AEI/10.13039/501100011033 and by ERDF/EU. YO-A was recipient of a predoctoral contract funded by the Spanish Ministerio de Universidades [grant FPU2019/00694]. IS-L was recipient of a predoctoral contract funded by the University of Extremadura. PG-S was supported by a Ramón y Cajal contract, reference RYC2022-035146-I, funded by MICIU/AEI/10.13039/501100011033 and the FSE+ and by Centro de Investigación Biomédica en Red en Enfermedades Neurodegenerativas-Instituto de Salud Carlos III (CIBER-CIBERNED-ISCIII) (CB06/05/0041). JM was supported by HORIZON-MSCA-2022-PF-01 101106857 and Talento César Nombela 2024-T1SAL-GL-31373. Funding for open access charge: Ministerio de Ciencia, Innovación y Universidades (MICIU/AEI/10.13039/501100011033) and ERDF/EU, Grant PID2023-147394NB-I00. None of these funding agencies participated in the study's design, execution of the research, or preparation of the manuscript.

## Author contributions

**Yolanda Orantos-Aguilera**: Formal analysis; Investigation; Methodology; Writing—review and editing. **Irene Sanchez-Lopez**: Formal analysis; Investigation; Methodology; Writing—review and editing. **Carlos Pascual-Caro**: Investigation; Methodology; Writing—review and editing. **Patricia Gómez-Suaga**: Resources; Methodology; Writing—review and editing. **Estela Area-Gomez**: Resources; Supervision; Methodology; Writing—review and editing. **Eulalia Pozo-Guisado**: Conceptualization; Formal analysis; Supervision; Writing—review and editing. **Jorge Montesinos**: Resources; Formal analysis; Supervision; Investigation; Writing—review and editing. **Francisco Javier Martin-Romero**: Conceptualization; Resources; Formal analysis; Supervision; Funding acquisition; Investigation; Methodology; Writing—original draft; Project administration; Writing—review and editing.

Source data underlying figure panels in this paper may have individual authorship assigned. Where available, figure panel/source data authorship is listed in the following database record: biostudies:S-SCDT-10_1038-S44318-026-00700-8.

## Disclosure and competing interests statement

The authors declare no competing interests.

# Expanded View Figures

**Figure EV1. Evaluation of the levels of STIM1 and MAM markers, and characterization of STIM2-KO cells.**

(A) Quantification of the level of STIM1 in MAM compared with the level in the ER. Data from 5 different subcellular fractionations. Figure 1E shows the result of a representative fractionation. Data are plotted as mean ± S.E.M. (B, C) The expression of PTPIP51 and VAPB proteins was assessed in WCL from wild-type U2OS, STIM1-KO U2OS, and STIM1-KO U2OS cells stably expressing STIM1-mCherry (KO + rescue) by immunoblot (30 μg protein/lane). STIM2-KO U2OS and ORAI1-KO U2OS cell lines were also analyzed on a separate gel. In all cases, GAPDH was used as a loading control. (D, E) Data from at least 2 biological replicates ($n \geq 5$ technical replicates per cell line) were quantified and plotted as the mean ± S.D. Data were normalized to WT levels. Statistical analysis with unpaired t-test, *$p = 0.0135$ and **$p = 0.0043$. (F) Total levels of STIM2 were evaluated by immunoblotting in wild-type and STIM2-KO U2OS cells (30 μg protein/lane). Levels of GAPDH were assessed as a loading control. (G) Fura-2-loaded cells were treated with 1 μM Tg in $Ca^{2+}$-free HBSS (added at min = 2). After a 7-min incubation, 2 mM $CaCl_2$ was added to assess the extent of SOCE. The graph shows the mean F340/F380 ratio ± S.D. over time from 3 independent experiments for WT U2OS (black line, $n = 38$ cells) and STIM2-KO U2OS cells (red line, $n = 50$ cells). Statistical analysis with unpaired t-test, $p < 0.0001$. Source data are available online for this figure.

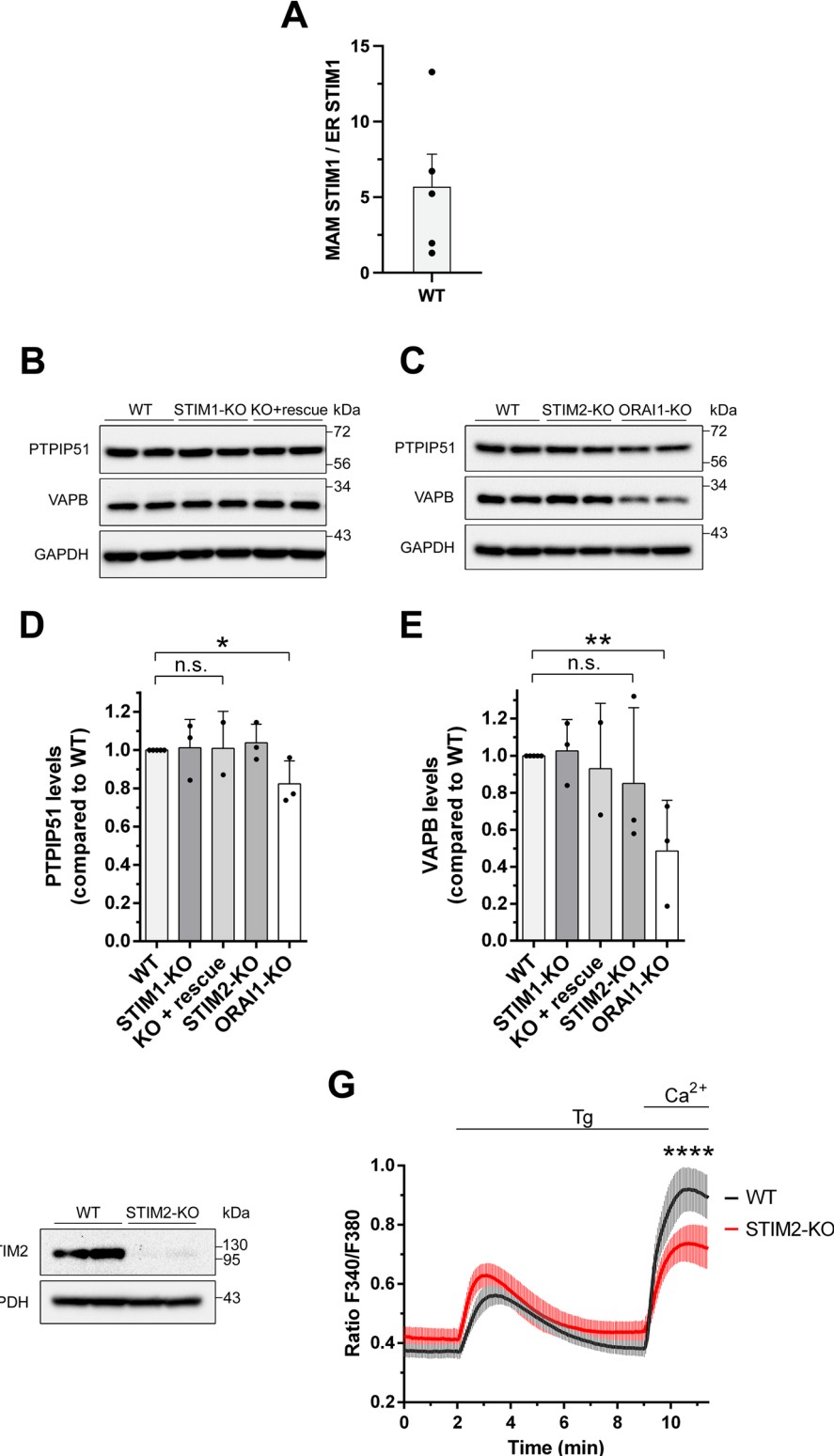

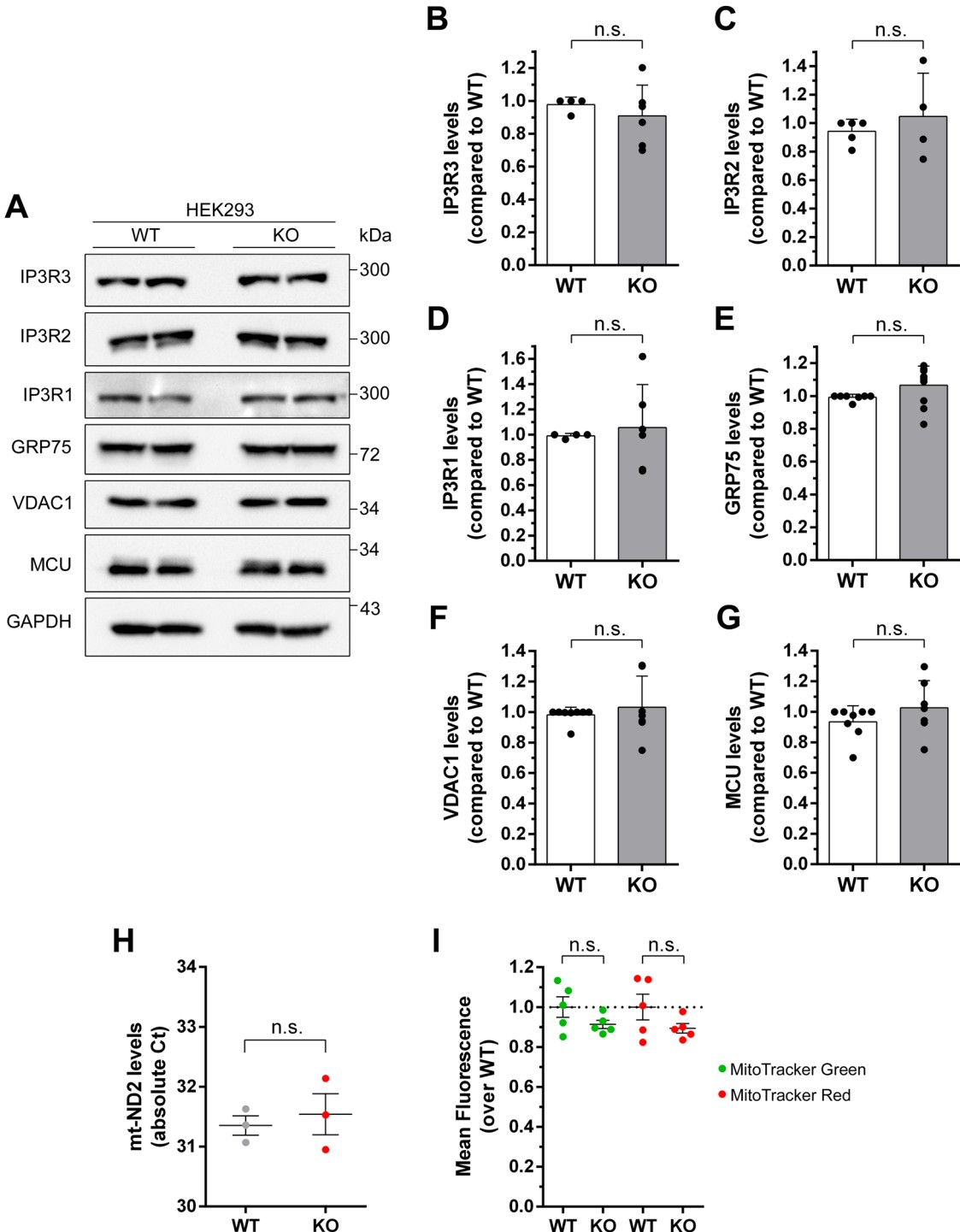

**Figure EV2.  Evaluation of the levels of IP3Rs, GRP75, VDAC1, MCU, and mitochondrial mass.**

(**A**) Expression of IP3R1, IP3R2, IP3R3, GRP75, VDAC1, and MCU was assessed in WCL from WT HEK293 and STIM1-KO HEK293 cells by immunoblot (40 μg/protein/lane). GAPDH was used as a loading control. (**B–G**) Data from blots were quantified, normalized relative to WT levels, and plotted as the mean ± S.D. from a minimum of 4 technical replicates and at least 2 biological replicates. Statistical analysis with unpaired t-test, $p = 0.503$ (**B**), $p = 0.475$ (**C**), $p = 0.7186$ (**D**), $p = 0.1258$ (**E**), $p = 0.5142$ (**F**), $p = 0.2384$ (**G**). (**H**) Mitochondrial DNA from HEK293 cells was purified and the levels of the mitochondrial gene *ND2* were quantified by real-time PCR. Data show the absolute Ct values from 3 independent experiments. Data are plotted as mean ± S.E.M. Statistical analysis with unpaired t-test, $p = 0.6488$. (**I**) Mitochondrial mass was quantified from HEK293 cells labeled with MitoTracker Green FM and Mitotracker Red FM and analyzed by flow cytometry. Fluorescence signal from 5 independent experiments was normalized to WT values. Data are plotted as mean ± S.E.M. Statistical analysis with unpaired t-test, $p = 0.1575$ (MitoTracker Green) and $p = 0.1611$ (MitoTracker Red). Source data are available online for this figure.

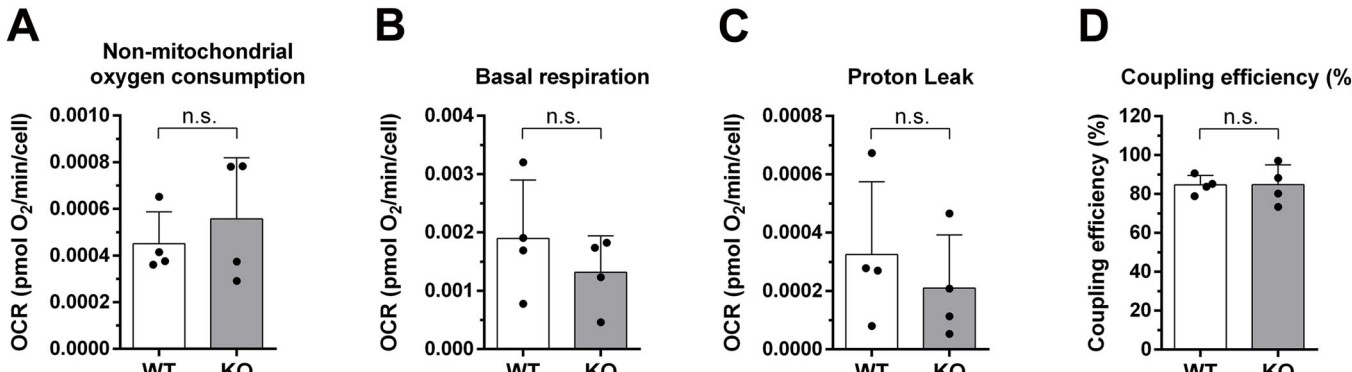

**Figure EV3. Quantification of mitochondrial parameters derived from Seahorse assay analysis.**

The average values from at least 3 technical replicates across 4 independent experiments were evaluated. Statistical analysis was performed using a paired t-test ratio to assess the differences between pairs. (A) Non-mitochondrial oxygen consumption: the minimum rate measurement after rotenone/antimycin A injection. (B) Basal respiration: the last rate measurement before oligomycin injection minus non-mitochondrial respiration rate. (C) Proton leak: the minimum rate measurement after oligomycin injection minus non-mitochondrial respiration. (D) Coupling efficiency: the percentage of ATP production rate divided by the basal respiration rate. In all cases, data are plotted as mean ± S.D. Source data are available online for this figure.

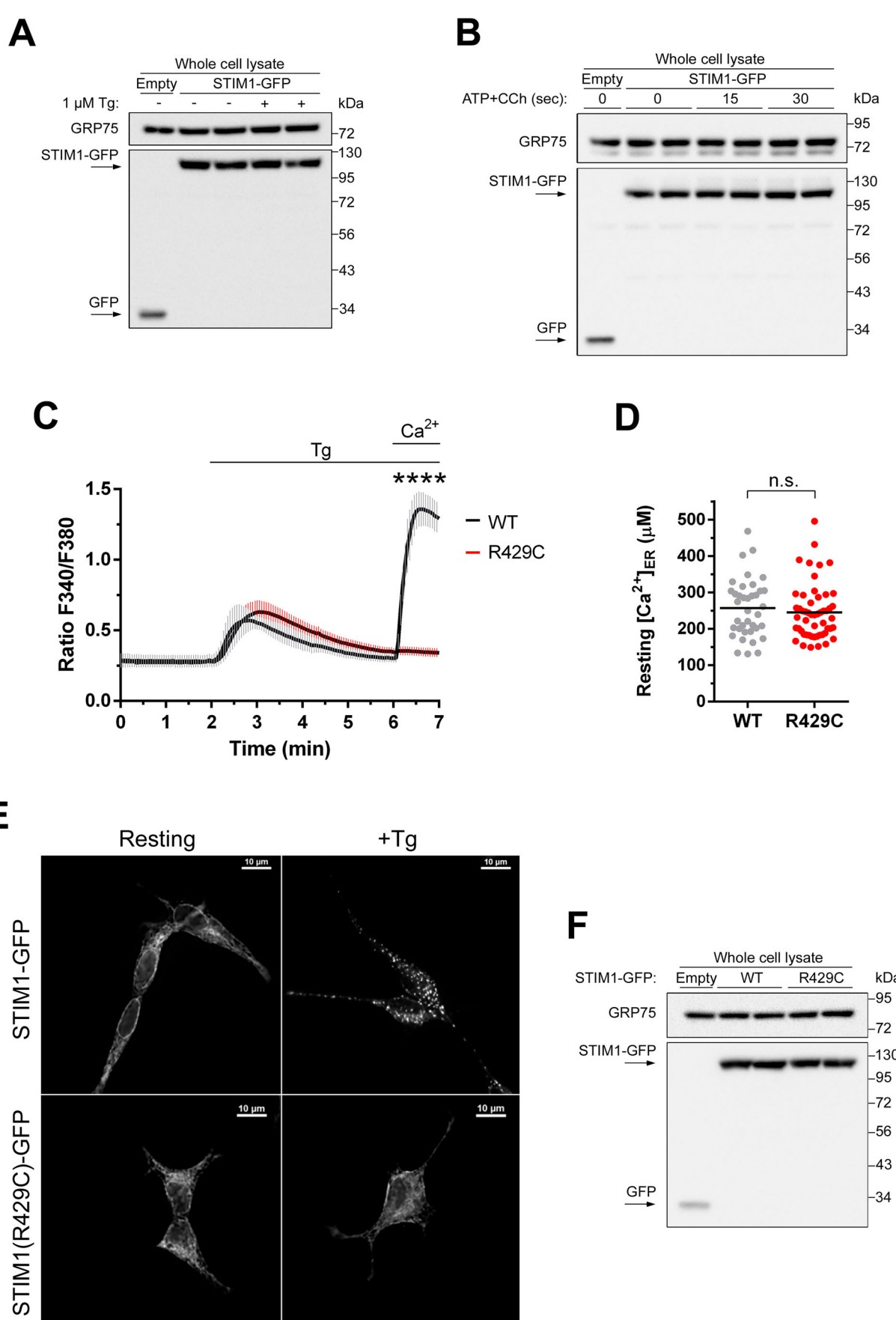

◀ **Figure EV4. Data supporting co-immunoprecipitation assays shown in Fig. 6, and characterization of STIM1(R429C)-expressing cells.**

(A) Data supporting co-immunoprecipitation experiment shown in Fig. 6A: Levels of total GRP75 and total STIM1-GFP were evaluated from WCL by immunoblotting (30 μg protein/lane). (B) Data supporting co-immunoprecipitation experiment shown in Fig. 6E: Levels of total GRP75 and total STIM1-GFP were evaluated from WCL by immunoblotting (30 μg protein/lane). (C) STIM1-KO HEK293 cells inducibly expressing Flag-STIM1 (black line) or Flag-STIM1(R429C) (red line) were incubated with fura-2-AM for 45 min. The F340/F380 ratio was then recorded in $Ca^{2+}$-free HBSS. After 2 min, cells were treated with 1 μM Tg, and 4 min later, 2 mM $CaCl_2$ was added to evaluate SOCE. Data are shown as the mean F340/F380 ratio ± S.D. from 2 independent experiments ($n = 24$ cells) for each cell line. Statistical analysis with unpaired t-test, $p < 0.0001$. (D) Level of $[Ca^{2+}]_{ER}$ in cells expressing STIM1(R429C). STIM1-KO HEK293 cells stably transfected for the expression of Flag-STIM1 (WT) or Flag-STIM1(R429C) were transiently transfected for the expression of the $Ca^{2+}$ sensor ER-GCaMP6-210. The graph shows basal $[Ca^{2+}]_{ER}$ values calculated from 3 independent experiments for cells expressing Flag-STIM1 ($n = 38$ WT ROIs) and from 4 experiments for cells expressing Flag-STIM1(R429C) ($n = 53$ R429C ROIs). The mean of data is represented by the black line. Statistical analysis with unpaired t-test. (E) HEK293 cells stably transfected for the inducible expression of STIM1(WT)-GFP or STIM1(R429C)-GFP were treated with 1 μM Tg in $Ca^{2+}$-free HBSS for 10 min, then fixed and mounted for microscopy analysis of GFP fluorescence. Images are representative of 3 independent experiments. Scale bar = 10 μm. (F) Data supporting co-immunoprecipitation experiment shown in Fig. 6J: Total levels of GRP75 and STIM1-GFP from WCL were analyzed by immunoblotting (30 μg protein/lane). Source data are available online for this figure.

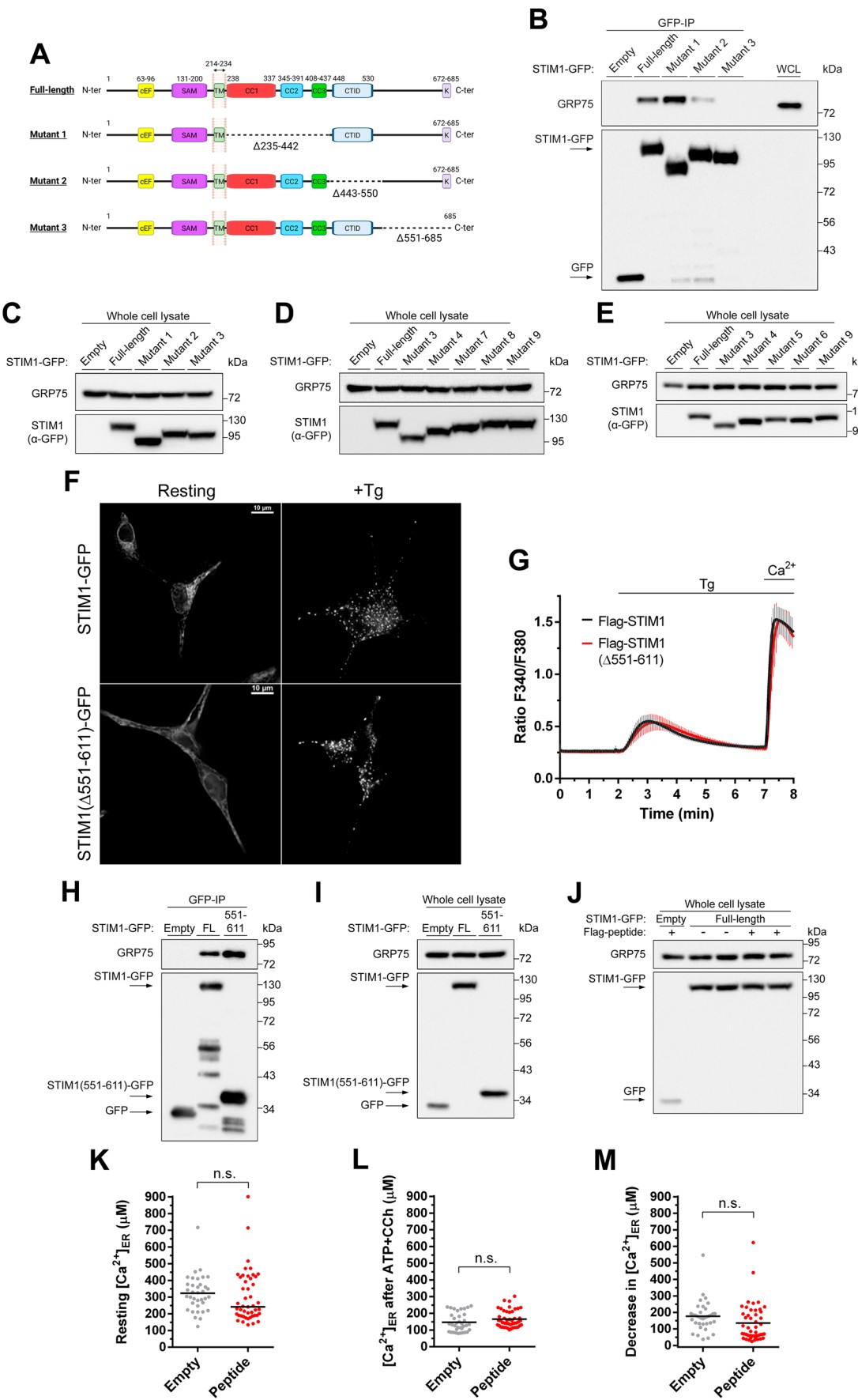

◄ **Figure EV5.**   **Defining the minimal sequence required to interact with GRP75.**

(A) Scheme of full-length STIM1 and STIM1 with C-terminal deletions. The following domains of STIM1 are shown: $Ca^{2+}$-sensitive EF-hand domain (cEF), sterile α motif (SAM), transmembrane domain (TM), coiled-coil domains (CC1-3), C-terminal inhibitory domain (CTID), and lysine-rich domain (K). The figure shows the deleted sequences and C-terminal domains in each mutant (named on the left), as reported elsewhere (Sanchez-Lopez et al, 2024). This scheme was created with BioRender.com. (B) Pull-down assays of GFP-tagged proteins were performed using 1 mg WCL from STIM1-KO HEK293 inducibly expressing full-length STIM1-GFP, STIM1(Δ235-442)-GFP (mutant 1), STIM1(Δ443-550)-GFP (mutant 2), STIM1(Δ551-685)-GFP (mutant 3), or GFP, as a control. Co-precipitation of GRP75 was assessed by immunoblot. A fraction of WCL (3 μg) from cells expressing STIM1-GFP was loaded as a positive control. Blots are representative of 3 biological replicates. Total pulled-down GFP was used as a loading control. (C) Total levels of STIM1 (either mutant or WT) and total levels of GRP75 in WCL were analyzed by immunoblotting (30 μg protein/lane). (D, E) Data supporting co-immunoprecipitation experiment shown in Fig. 7B and 7C, respectively: Total levels of GRP75 and STIM1-GFP from WCL were analyzed by immunoblotting (30 μg protein/lane). (F) HEK293 cells stably transfected for the inducible expression of STIM1(WT)-GFP or STIM1(Δ551-611)-GFP were treated with 1 μM Tg in $Ca^{2+}$-free HBSS for 10 min, then fixed and mounted for microscopy analysis of GFP fluorescence. Images are representative of 3 independent experiments. Scale bar = 10 μm. (G) STIM1-KO HEK293 cells inducibly expressing either Flag-STIM1(full-length) or Flag-STIM1(Δ551-611) were incubated with fura-2-AM. The F340/F380 ratio was then recorded in $Ca^{2+}$-free HBSS. After 2 min, cells were treated with 1 μM Tg, and after a 5-min incubation, 2 mM $CaCl_2$ was added to the assay medium to evaluate SOCE. Data represent the mean ± S.D. of the F340/F380 over time from 3 independent experiments for cells expressing Flag-STIM1 (black line, $n = 33$ cells) and Flag-STIM1(Δ551-611) (red line, $n = 35$ cells). (H) Immunoprecipitation assay (1 mg WCL) from STIM1-KO HEK293 cells inducibly expressing STIM1-GFP (labeled as FL), the STIM1(551-611)-GFP peptide, or GFP as a control. GRP75 co-precipitation was analyzed by immunoblotting. Levels of immunoprecipitated GFP were evaluated as a loading control. (I) Total levels of GRP75 and GFP-tagged proteins in the WCL used for the co-immunoprecipitation shown in (H) were assessed by immunoblotting (30 μg protein/lane). (J) Supporting data for the co-immunoprecipitation experiment shown in Fig. 7H: total levels of GRP75 and STIM1-GFP in WCL were analyzed by immunoblotting (30 μg protein/lane). (K) HEK293 cells stably transfected for the inducible expression of Flag-STIM1(551-611) (referred to as peptide), or Flag-empty vector as a control, were transiently transfected for the expression of the $Ca^{2+}$ sensor ER-GCaMP6-210. The graph shows basal (steady-state) $[Ca^{2+}]_{ER}$ in both cell lines. The mean of data is represented by the black line. Statistical analysis with unpaired t-test. (L) $[Ca^{2+}]_{ER}$ values after $Ca^{2+}$ release stimulated with ATP+CCh. Statistical analysis with unpaired t-test. (M) Decrease in $[Ca^{2+}]_{ER}$ following ATP+CCh stimulus. In (K–M), $[Ca^{2+}]_{ER}$ values were calculated from 3 independent experiments for cells expressing Flag-(empty tag) ($n = 36$ ROIs) and from 4 experiments for cells expressing Flag-peptide 551-611 ($n = 47$ ROIs). Statistical analysis with unpaired t-test. Source data are available online for this figure.

