## [Peer Review File · The EMBO Journal]

STIM1-containing contact sites promote direct calcium flux from the endoplasmic reticulum to mitochondria

Yolanda Orantos-Aguilera, Irene Sanchez-Lopez, Carlos Pascual-Caro, Patricia Gomez-Suaga, Estela Area-Gomez, Eulalia Pozo-Guisado, Jorge Montesinos, and Francisco Javier Martin-Romero

Corresponding author(s): Francisco Javier Martin-Romero (fjmartin@unex.es) , Jorge Montesinos (jorge.montesinos@cib.csic.es)

Review Timeline:

Submission Date:	26th Feb 25
Editorial Decision:	4th Apr 25
Revision Received:	9th Oct 25
Editorial Decision:	4th Dec 25
Revision Received:	8th Dec 25
Accepted:	22nd Dec 25

Editor: William Teale

Transaction Report:

Dear Dr. Martin-Romero,

Thank you again for the submission of your manuscript entitled "STIM1 regulates calcium flux between the endoplasmic reticulum and mitochondria at contact sites" and for your patience during the review process. We have now received the reports from the referees, which I copy below.

As you can see from their comments, while referees expressed enthusiasm for the scope and timeliness of the study, they also expressed concerns over the concrete mechanistic insights gained. At this stage, I am considering this manuscript as a better fit for EMBO Reports; however, it would certainly be a good idea to discuss this on a Zoom call. In the meantime, I am happy to formally invite you to address the comments of all referees in a revised version of the manuscript. That said, I must emphasise that, depending on the revisions that are feasible within a timeframe of three to six months, I may recommend publication in EMBO Reports after re-review.

I should add that it is The EMBO Journal policy to allow only a single major round of revision and that it is therefore important to resolve the main concerns at this stage. Please contact me if you have any questions, need further input on the referee comments or if you anticipate any problems in addressing any of their points. Please, follow the instructions below when preparing your manuscript for resubmission.

I would also like to point out that as a matter of policy, competing manuscripts published during this period will not be taken into consideration in our assessment of the novelty presented by your study ("scooping" protection). We have extended this 'scooping protection policy' beyond the usual 3 month revision timeline to cover the period required for a full revision to address the essential experimental issues. Please contact me if you see a paper with related content published elsewhere to discuss the appropriate course of action.

Again, please contact me at any time during revision if you need any help or have further questions.

Thank you very much again for the opportunity to consider your work for publication. I look forward to your revision.

Best regards,

William Teale

William Teale, PhD
Editor
The EMBO Journal
w.teale@embojournal.org

- a point-by-point response to the referees' comments, with a detailed description of the changes made (as a word file).
- a word file of the manuscript text.

- individual production quality figure files (one file per figure)
- a complete author checklist, which you can download from our author guidelines (<https://www.embopress.org/page/journal/14602075/authorguide>).
- Expanded View files (replacing Supplementary Information)

We realize that it is difficult to revise to a specific deadline. In the interest of protecting the conceptual advance provided by the work, we recommend a revision within 3 months (3rd Jul 2025). Please discuss the revision progress ahead of this time with the editor if you require more time to complete the revisions. Use the link below to submit your revision:

Referee #1:

This study investigates the role of STIM1 in mitochondria-associated ER membranes (MAM) and its involvement in ER-mitochondria Ca^{2+} transfer. The findings indicate that STIM1 (in its inactive state) is localized in MAM, where it interacts with proteins such as PTP51 and GRP75. STIM1 deficiency is associated with reduced mitochondrial Ca^{2+} levels, impaired mitochondrial respiration, and decreased ATP production. The study further suggests that STIM1 interaction with GRP75 is influenced by its Ca^{2+} -sensing ability, with ER Ca^{2+} depletion and the R429C mutation affecting this interaction. Deletion analysis identifies the STIM1(551-611) region as important for GRP75 binding. Overall, the study suggests a potential role for STIM1 in inter-organelle communication.

The subject is interesting, timely, and relevant to the readership of The EMBO Journal.

I have a number of questions, comments and suggestions:

Major

- In its inactive state, i.e., when ER calcium stores are full, STIM1 and STIM2 are homogeneously distributed on the ER membrane. Consequently, it is expected that a portion of STIM1 would be located at MAMs, i.e., at mitochondria-ER contact sites. The authors should analyze and compare STIM1 MAM distribution and its interactome in cells with full and depleted ER calcium stores.
- Previous studies have reported STIM1 interactome analyses as well as mitochondria-ER contact site (MERC) omic datasets. The authors should compare their findings with existing literature to determine whether STIM1 interacts with MERC proteins in a manner similar to their observations. Any discrepancies should be discussed.
- Many figure panels consist of single immunoblots. Presumably, these experiments were performed multiple times. The authors should provide replicate data and include statistical analyses indicating the significance of their findings along with the number of experiments performed.
- A major concern is the data presented in Figures 5D and 5E. If STIM1 knockout does not affect store-operated calcium entry (SOCE), then the validity of the other findings comes into question. Numerous studies have demonstrated that STIM1 downregulation leads to SOCE inhibition. The authors should clarify this apparent inconsistency.
- The calculation and quantification of MAM (MERC) length and number by fluorescence microscopy alone is insufficient. These measurements should be supported by an additional technique, such as electron microscopy.
- The authors suggest that the observed decrease in oxygen consumption rate (OCR) is due to NDUFB8 deficiency. However, what evidence rules out the impact of mitochondrial Ca^{2+} levels on mitochondrial dehydrogenases, which are known to be regulated by matrix Ca^{2+} ?
- A functional evaluation of the proposed mechanism is missing. What is the physiological/pathological relevance?
- The ROS measurements using CellROX (which is not the most precise ROS sensor) indicate elevated ROS levels in STIM1-KO cells despite the reduced OCR. The underlying mechanism of this observation is not sufficiently addressed.

Minor

- Given the small differences observed at only one time point, the conclusions drawn from Figure 4B should be moderated.

- The y-axis scaling in Figures 5A-C should be standardized for better comparison.
- To accurately quantify ER calcium content, the area under the curve should be measured or a similar approach should be employed.
- The data presented in Figures 1-4 could be condensed into a maximum of one or two figures to improve clarity and conciseness.
- What is the rationale to use U2OS cells for some experiments and HEK293 for others?
- The use of HEK cells overexpressing STIM1-GFP does not appear to add substantial value. Since the same STIM1-GRP75 interaction was observed in WT cells, it may be advisable to remove Figure 1A.

Summary:

While the study addresses an interesting and timely topic, the current findings appear to be at a preliminary stage. Additionally, some conclusions are not fully supported by the experimental data. Further validation and additional controls are needed and would strengthen the main findings of this study.

Referee #2:

The present work by Oratos-Aguilera et al., focused in the role of STIM1 in the regulation of calcium influx from ER to mitochondria via contact sites. Here, the authors identified PTP51 and GRP75 as interactors of STIM1 thus modulating the mitochondrial calcium and function.

Using biochemical and cell biology methods the authors studied the interaction of STIM1, Grp75 and VDAC1/2/3 by co-immunoprecipitation endogenous and overexpressing STIM1-GFP. Then, subcellular fractionation was performed showing that STIM1 localize at the membrane contact sites together with TOM20 and ACSL4 and ERLIN2. Then, interaction with another MAM protein was analyzed (PTP51). PLA dots (VAPB-PTP51) were increased in STIM1KO cells which was rescued with the expression of STIM1 but not STIM2. Additionally, phospholipid transfer with radiolabeling of 3H-serine did not show significant differences in the synthesis while the ratio of PE/PS varied in STIM1KO cells. Next, the authors evaluated the calcium concentration at ER. Basal and after adding ATP+CCh showed decreased ER concentration in STIM1 KO cells while the cytosolic calcium levels were not affected. In addition, mitochondria calcium levels were studied using mito4x-GCamp6f. Cells treated with ATP+CCh didn't increase mitochondrial calcium level in the KO cells as well in resting and basal conditions. No changes in protein levels of IP3Rs were observed in the subcellular fractionation assays however the PLA dots (VDAC1 and IP3R31/2/3) were increased in the KO cells. Decreased mitochondria oxygen consumption was observed in STIM1 KO cells (basal and maximal) which was accompanied of a reduced ATP production. Total protein levels of mitochondrial complex I (CI-NDUFB8) were decreased in STIM1 KO cells. Then, cell death was analyzed. STIM1 KO cells were shown to increase the cell death while the rescue partially reduced the death induced with duramycin. Thapsigargin treatment affected the interaction with GRP75 and increased PLA dots (VAPB-PTP51). The addition of ATP-CCh decreased the interaction to GRP75 as well. Disease related mutation of STIM1-R429C was studied. This mutation affected the interaction with GRP75 as well the formation of PLA dots (VAPB-PTP51). Finally, several mutations on STIM1 were evaluated. Here, mutants lacking the residues from 551-685, 551-642 and 551-611 failed to interact. Also, deletion of the residues 668-674, 672-685 and 582-642 were important for the interaction with GRP75 (Mutatns:3,4,5,6,8 and 9). Here, the mutant 5 decreased the PLA dots of GFP-STIM1 and GRP75 while just expressing the residues 551-611 was enough to bind GRP75. Deletions of these residues increased the PLA dots VAPB-PTP51.

The manuscript in the current state is not recommended for revision. The data presented require further analysis and studies to support their conclusions/statements. Several interactions were characterized, however direct link among them has not been properly studied. Further analysis including appropriate methods to provide new evidence will improve the present manuscript.

Major concerns

Although the study of STIM1 as a MAM protein is interesting, the authors did not provide convincing data to support their statements. Unbiased analysis of MAM proteins with current methodologies such as proteomics and localization analysis must to be added to support the present study.

STIM1 has been shown to acts as a calcium sensor regulating the gating from plasma membrane channels like Ora1. How does to modulate the calcium at the mitochondria? Here some evidence was indicating some interaction but direct functional assays and mechanisms of regulation were not provided.

The potential impact of STIM1 deletion over the formation of multiple contacts sites makes difficult the conclusions. Both IP3R3-VDAC and VAPB-PIPT51 membrane contact sites has been directly involved regulating calcium and lipid transfer respectively. How do the authors differentiate of an indirect effect of STIM1 deletion versus direct role regulating channel's function? Also, STIM1 role at plasma membrane can indirectly alter the calcium levels which will affect the calcium uptake capacity of the ER. Here, the data it is not convincing and further studies has to be evaluated. Furthermore, the present work lacs in vivo studies using models to test its physiological relevance either including disease related mutations in primary cells or in a mouse model.

Minor comments

Several microscopy analyses were performed while the conclusions are based on proximity ligation assay. The authors should include other experiments to confirm their findings. Here, electron microscopy analysis and the use of fluorescence reporters (SPLICs for example) coupled to high through put microscopy analysis must be included.

Several co-immunoprecipitation assays were provided; however, no quantifications were added. This makes hard to conclude. In the subcellular fractionation assays, several loading controls are missing and proper MAMs, ER and mitochondria markers are

not used. Cytosolic marker it is not provided, which make difficult to conclude about the GRP75 recruitment or binding since it is an associated cytosolic protein in the later fractionation panels.

Figure 1: Validation of mitochondrial SIMT1 interactors by co-immunoprecipitation.

Here co-IP experiments were presented however, fractionation assays, colocalization assays of STIM1, ER and mitochondria are missing.

The endogenous IP shows the interaction of STIM1 with GRP75. However, the endogenous band of SIMT1 is presented in the IP fraction does not appear in the WCL. The authors should either remove this data or provide a different experiment showing the STIM1 detection in the inputs.

Figure2: Evaluation of the presence of STIM1 in MAMs.

The fractionation assay is not completely clear. The authors produced crude mitochondria which contains MAMs and pure mitochondria it is not provided.

Here, the PLA dots of PTP51 and STIM1 was analyzed, however other MAM proteins can be included such as VDACS or TOMs to better illustrate STIM1 localization as MAM protein.

Figure 3: Analysis of the functional consequences of STIM1 deficiency on MAMs.

It is known that STIM1 oligomerizes and interacts with ORA1. How the authors connect an increased activation of SIMT1 and the amount of PLA dots VAPB and PTP51? The authors should include more dynamics of the contacts upon SIMT1 activation or ER stress, not only chronic deletion.

Figure 4: Analysis of phospholipid synthesis in the crude mitochondrial fraction.

Controls of purity of mitochondrial fraction has to be included. To evaluate it is crude or pure mitochondria.

This assay it seems interesting, why the authors did not directly measure all the lipid profile of mitochondria not only the PE/PS.

Figure 5: Analysis of ER-to mitochondria Ca²⁺ shuttling in the absence of STIM1.

Expression of the calcium sensors should be provided. Representative images are missing. Authors should add these analyses.

Figure 6: Studying the IP3R-GRP75-VDAC axis in the absence of STIM1.

How do the authors explain that contradictory to an increased number of contacts sites in SIMT1 KO cells, the function, meaning the calcium levels at ER and mitochondria are not increased rather decreased? Here inhibitors of the main ER pump can be used (thapsigargin) and measure the calcium changes like in the Fig 5F.

Figure 7: Bioenergetic analysis of WT and STIM1 KO cells.

How do the decreased OCR and ATP production are associated with the membrane potential and the ROS produced in mitochondria? This will be important to measure in the STIM1 ko cells and the mutant to evidence physiological relevance.

Figure 8: Total levels of proteins from the mitochondrial respiratory chain complexes.

Since the authors observe decreased levels of CI-NDUF8 in the STIM1 KO cells, how do they discriminate on changes in the mitochondrial mass or biogenesis? Additional studies to support these observations has to be included.

Figure 9: Analysis of oxidative stress and sensitivity to duramycin.

There are better methods to assess cell death. The authors did evaluate cell proliferation instead? Also, which cell death is increased? Ferroptosis? Apoptosis? Specific analysis has to be included to further conclude that STIM1 KO decreases viability of the cells.

Figure 10: Analysis of the STIM1-GRP75 interaction after ER Ca²⁺ depletion.

Figure11: Analysis of the interaction between STIM1 (R429C) and GRP75 by co-immunoprecipitation.

Figure 12: Analysis of the co-precipitation between STIM1 mutants with C-terminal deletion and GRP75.

Here, several deletions were generated, however a convincing model of interaction has to be included. Molecular simulations and alpha fold methods can be included to between decipher the regions of interactions and support the direct interaction and not misfolding of the protein or aggregates. Do the mutants distribute at the ER and can oligomerizes correctly?

Since the authors showed initial interactions of STIM1 with other MAM proteins, do these mutants affect the interactions with VDACS? Also, did the authors determined that the GRP75 it is actually contributing to the binding to VDACS or PTP51? Or STIM1 can directly binds these proteins?

General

The entire workflow of the manuscript it is confusing. The figure 1 and 2 should be together as the text description is separated but there is not so much data.

In general, there are too many figures some with a lot of data and the majority with very little or no data. The data can be mentioned but only the most relevant information in the main figures.

Fluorescence images are not documented and schemes of work and experimental design should be added to better follow the study.

The intro mentions general aspects of the MAM associated proteins and mechanisms. Here, more directed introduction can be included. Special focus to the proteins studied in the present work and the functionality or what it is known of STIM1.

The figures of the co-immunoprecipitation are named GFP-pulldowns were should be GFP-IP since is not purified GFP-STIM1. The titles of the legends should contain the results rather than the method. It makes hard to connect results to the conclusions and it is not so informative.

Referee #3:

The manuscript entitled, "STIM1 Regulates Calcium Flux Between the Endoplasmic Reticulum and Mitochondria at Contact Sites" is an intriguing study offering a previously unidentified role for STIM1 in control of MAM formation and ER-Mitochondria

calcium transfer. Essentially, what is proposed is that STIM1 binds constitutively to GRP75 within the mitochondria and that these interactions are lost in response to ER calcium depletion. The implications of this study are highly significant, although some modifications needed. Hence, if mitochondrial calcium loading depends on resting STIM1, then its activation would decrease the ability of mitochondria to load. As such, mitochondrial calcium loading would be predicted to initially be highly efficient, but would then decrease as the ER depletes of calcium (which would disrupt MAMs for the purposes of calcium transfer).

Specific Comments:

1. In discussing figure 4B there are statements suggesting that there are significant differences in PE/PS ratio at 10 minutes, but not 20 minutes. However, the true PE/PS ratio at 20 minute group may well follow the same pattern as at 10 minutes; the data is skewed by sampling error and I would encourage the authors to repeat this work and determine what is actually correct. The fact that the data is normalized to the 10 minute WT datapoint only exacerbates this issue of "inconsistent error" amongst groups.

I would also add that PE levels rather than PE/PS ratio seems to be the primary altered group, even if your work didn't pass statistics. More clarity on experimental findings vs. effect of STIM1 on MAM function is needed. Perhaps consult a statistician to help you design a new experiment based on this one.

As an aside, " =min and '=hrs.

2. In figure 5, many of the resting ER calcium concentrations are below 250 μM . At these concentrations, STIM1 should be active. How do you explain this (important to recognize that calibrations are often of questionable reliability)? In addition, it would have been helpful to at least reference the techniques used to measure ER calcium within the results section. Also, was figure 5D performed without extracellular calcium? If so, this should be stated in the results before stating the outcome of the experiment.

3. Complex I produces superoxide; why would loss of complex I lead to increase ROS (the conclusion of figure 8)?

4. The duramycin experiment in figure 9 is difficult to interpret as completed. There are much better approaches to demonstrate changes in the asymmetry of PS and PE. Susceptibility to cell death provides minimal insight into what has actually occurred in the cell.

5. In figure 11, in describing STIM1-R429C, it is described as "constitutively unfolded". That is a confusing and not fully accurate description of this mutation. Actually, in this case, I believe that the construct was used correctly and supports the paper; it needs to be described better. Indeed, the R429C mutation is in CC3 within SOAR. As such, this is a mutant that primarily targets the site of interaction between STIM1 and Orai1. SOAR also has putative roles in STIM1 multimerization. Linking multimerization to STIM1-GRP75 would strengthen the conclusions of this study.

Referee #4:

Mitochondria and ER directly communicate and coordinate functions at mitochondria-associated ER membranes (MCS), regions of close membrane apposition tethered by proteins. To date, MAMs have been demonstrated to enable the bidirectional transport of signaling molecules, coordinate biosynthetic processes. However, the factors that mediate these functions remain little understood. Here, the authors show that the ER Ca^{2+} sensor STIM1-known for its role in activating plasma membrane Ca^{2+} channels-localizes to MAMs. The loss of STIM1 impaired PE synthesis, disrupted ER-to-mitochondria Ca^{2+} transfer, and decreased maximal mitochondrial respiration and ATP production. Although interesting, the authors claims of the role of STIM1 in mitochondria-ER communication are unsubstantiated.

Major Comments:

Role of STIM1 in contact sites:

The authors claim that the loss of STIM1 leads to increased contact sites. The main readout for contact sites is the PLA assay. In most of the figures where the PLA assay is shown, it is unclear whether the puncta correlate with ER/mitochondria. To substantiate their claims of the role of STIM1 and the relevant mutants analyzed in contact sites, the authors should perform electron microscopy. This will allow them to assess not only the length of the contact sites, but also the ER-mitochondria distance which are both important parameters. This is especially important given previous work showing there were no differences in STIM1 KO cells (Henke et al 2012).

Are the effects of STIM1 simply mediated by the loss of SOCE? STIM1 activates plasma membrane Ca^{2+} channels. Thus, all of the consequences of STIM1 deletion may be a result of impaired SOCE. The authors state that "to rule out the possibility that this decrease was due to SOCE inhibition rather than STIM1 conformation, we reintroduced Ca^{2+} into the assay medium." However, this is not sufficient to exclude a role for SOCE, and furthermore was only addressed for the Ca^{2+} depletion, and not for any of the other consequences of the STIM1 loss (i.e. PE synthesis, increase in MAMs) etc. Are there STIM1 mutants that do not interact with Orai that can be used? The authors could check whether the STIM1 mutants they use are still capable of binding Orai, and consider performing experiments in which a MAM defect is demonstrated in Orai1 KO/STIM1 Orai1 DKO

cells.

Minor:

Lack of control in IPs. Whole cell lysate and IPs should be run together (unclear why they are run separately, and only one 'WCL' is run with the pull-down fractions. Furthermore, IPs should be properly controlled. Eg. Fig. 1B/2B: the authors didn't analyze the IP fraction for any other mito or ER proteins to assess whether this is a specific interaction.

STIM1 SOCE, STIM1 MAMs: The authors should discuss how these different subpopulations of Stim1 may be regulated.

Referee #1:

This study investigates the role of STIM1 in mitochondria-associated ER membranes (MAM) and its involvement in ER-mitochondria Ca²⁺ transfer. The findings indicate that STIM1 (in its inactive state) is localized in MAM, where it interacts with proteins such as PTPIP51 and GRP75. STIM1 deficiency is associated with reduced mitochondrial Ca²⁺ levels, impaired mitochondrial respiration, and decreased ATP production. The study further suggests that STIM1 interaction with GRP75 is influenced by its Ca²⁺-sensing ability, with ER Ca²⁺ depletion and the R429C mutation affecting this interaction. Deletion analysis identifies the STIM1(551-611) region as important for GRP75 binding. Overall, the study suggests a potential role for STIM1 in inter-organelle communication.

The subject is interesting, timely, and relevant to the readership of The EMBO Journal.

I have a number of questions, comments and suggestions:

Major

Q1: *In its inactive state, i.e., when ER calcium stores are full, STIM1 and STIM2 are homogeneously distributed on the ER membrane. Consequently, it is expected that a portion of STIM1 would be located at MAMs, i.e., at mitochondria-ER contact sites. The authors should analyze and compare STIM1 MAM distribution and its interactome in cells with full and depleted ER calcium stores.*

A1: To address this question, we prepared samples of total lysates and MAM fractions from HEK293 cells under two conditions: (1) at rest, i.e., with full calcium stores, and (2) after store depletion induced by ATP-carbachol (CCh) stimulation for 30 seconds in Ca²⁺-free medium. Following MAM purification, we analyzed the presence of STIM1 and GRP75 in these MAM fractions, as GRP75 protein is a key interactor of STIM1 within MAMs, and it is involved in the regulation of Ca²⁺ trafficking. The results are presented in the new Figure 6 (panels H-I). These data show that store depletion decreases the levels of STIM1 and GRP75 in MAMs, without affecting their levels in total lysates. This reduction is indeed associated with a decrease in the interaction between STIM1 and GRP75. We attribute this to a conformational change in STIM1 in response to store depletion, as supported by the observation that mutants displaying a more extended (or open) conformational state (such as the R429C mutant) also show reduced interaction with GRP75 (see new Figure 6.J-K).

Since this is one of the most important and valuable points in the manuscript, we sincerely thank the reviewer for the suggestion, which has significantly contributed to shaping the final proposal of the Ca²⁺ transfer control mechanism regulated by STIM1–GRP75 (Figure 8).

Q2: *Previous studies have reported STIM1 interactome analyses as well as mitochondria-ER contact site (MERC) omic datasets. The authors should compare*

their findings with existing literature to determine whether STIM1 interacts MERC proteins in a manner similar to their observations. Any discrepancies should be discussed.

A2: Studies of the STIM1 interactome have been reported in recent literature, and their results are compiled in the BioGRID database. However, no studies have specifically addressed the STIM1 interactome in MAMs. Based on our own data, together with the results available in BioGRID, we were able to confirm a series of interactions with mitochondrial proteins. The interactions derived from BioGRID, are detailed in the table accessible through the following link Biogrid_STIM1(HUMAN) interactome_mitochondrial proteins.xlsx

Since these data originate from different techniques and from tissues/cells of different origins, we do not consider it appropriate to discuss them in the main section of the manuscript, although they are addressed here. Nevertheless, and in response to the reviewer's request, a reference to the BioGRID results has been included, which can be found in the *Discussion* section. This statement in the manuscript refers to the fact that the set of STIM1 interactors we identified differs from the previously reported set, although both datasets essentially converge on the same conclusion: STIM1 interacts with multiple proteins of the outer mitochondrial membrane. For example, in BioGRID we find interactions with COX14, FIS1, and SLC25A46, all of which are localized to the outer mitochondrial membrane.

It should be noted, however, that the discrepancies between different interactomes are primarily attributable to the sample extraction method, since the detergent composition of the extraction buffers determines the resolution of the interactome by setting a threshold of strength for the interactions, particularly when these involve solubilized membranes from different organelles.

Taken together, after identifying interactors in our immunoprecipitation+MS-based interactor identification assay (published under PMID: 38224453, and referenced in the manuscript), complemented by our low-throughput classical co-IP analysis, and by establishing that these represent dynamic interactions responsive to stimuli such as ER store depletion, our findings support the conclusion that the STIM1–GRP75 interaction constitutes a *bona fide* interaction.

Q3: *Many figure panels consist of single immunoblots. Presumably, these experiments were performed multiple times. The authors should provide replicate data and include statistical analyses indicating the significance of their findings along with the number of experiments performed.*

A3: The figures submitted in the initial version of the manuscript were not accompanied by the raw data, as this is not required for a first submission. However, in this revised version, we have included the original data as well as all possible statistical analyses.

Q4: A major concern is the data presented in Figures 5D and 5E. If STIM1 knockout does not affect store-operated calcium entry (SOCE), then the validity of the other findings comes into question. Numerous studies have demonstrated that STIM1 downregulation leads to SOCE inhibition. The authors should clarify this apparent inconsistency.

A4: We regret that the original manuscript was not sufficiently clear and may have led to this misinterpretation. STIM1-KO cells, both U2OS and HEK293, are deficient in SOCE, as we have previously reported in PMID:28341841 (for U2OS cells) and in PMID:38224453 (for HEK293 cells).

The misunderstanding regarding Figures 5D and 5E (as numbered in the first version) arises from the fact that it was not made sufficiently explicit that **these two figures do not describe SOCE**, but rather cytosolic Ca^{2+} levels in the absence of extracellular Ca^{2+} , that is, **Ca^{2+} release from intracellular stores** (mainly the ER). To ensure that this information is clear and precise, we have added this clarification to the figure legend and included a banner within the figure indicating that this corresponds to an assay in Ca^{2+} -free medium (now in Fig 3.E).

Q5: The calculation and quantification of MAM (MERC) length and number by fluorescence microscopy alone is insufficient. These measurements should be supported by an additional technique, such as electron microscopy.

A5: After careful discussion of this point, we provide an alternative approach to assess ER-mitochondria contacts. While electron microscopy offers high 2D spatial resolution, it lacks the 3D sensitivity (along the Z axis) that can be achieved with fluorometric techniques. In this regard, and in response both to Reviewer 1, 4, and to the Reviewer 2, who explicitly suggested the use of SPLICS, we have performed new experiments using dimerization-dependent fluorescent proteins (ddFPs) targeted to mitochondria and the ER, as recently reported in PMID:38327561.

Cells were transfected for the expression of Mito-Green(A) and ER(B), such that the reconstitution of A-B products indicates ER-mitochondria contact sites. For quantification, individual cells were analyzed by microscopy, measuring green fluorescence intensity, which was normalized to anti-TOM20 immunolabeling (in red) as a mitochondrial marker. The results, presented in the new Figures 2 and 6, further support our previous conclusions, namely: (1) STIM1-KO cells display increased levels of ER-mitochondria contacts (Figure 2.D-E), and (2) depletion of ER Ca^{2+} stores induced by ATP+CCh in Ca^{2+} -free medium also increases ER-mitochondria contacts compared to resting conditions (Fig 6.G).

In addition, although this was not explicitly requested, we cloned the coding sequence of the B-gene (from the ER(B) construct) into a plasmid encoding STIM1, thereby generating the pair Mito-Green(A) and STIM1(B), which allowed us to

assess STIM1–mitochondria contacts. These new results, presented in Figure 1 (panel I-K), demonstrate the localization of STIM1 at MAMs.

Q6: *The authors suggest that the observed decrease in oxygen consumption rate (OCR) is due to NDUFB8 deficiency. However, what evidence rules out the impact of mitochondrial Ca^{2+} levels on mitochondrial dehydrogenases, which are known to be regulated by matrix Ca^{2+} ?*

A6: The reviewer is correct in highlighting the role of mitochondrial dehydrogenases, and we are grateful for this insightful comment, which we have carefully considered. Indeed, one of the regulatory mechanisms of the Krebs cycle is linked to Ca^{2+} concentration within the mitochondrial matrix, since pyruvate dehydrogenase (PDH) complex phosphatase activity is stimulated by Ca^{2+} . Dephosphorylation of the complex leads to its active state; thus, monitoring the phosphorylation level of PDH is a classical approach to assessing its activity. This is precisely what we have performed, and the results are shown in the new Figure 5 (panels L-M). In summary, STIM1-KO cells display higher levels of phospho-PDH, indicating reduced activity. Given that PDH is one of the main control points of the Krebs cycle, this implies a decreased supply of reducing equivalents to the electron transport chain (ETC) and, consequently, lower ATP production.

As a control, and to ensure that we were accurately detecting changes in phosphorylation status, lysates of cells treated with dichloroacetate, an inhibitor of the PDH kinase, were analyzed. As expected, this treatment resulted in reduced PDH phosphorylation.

Q7: *A functional evaluation of the proposed mechanism is missing. What is the physiological/pathological relevance?*

A7: At the reviewer's request, which we consider highly appropriate, we have added a final figure presenting the proposed molecular mechanism underlying the control of Ca^{2+} transfer between the ER and mitochondria mediated by STIM1–GRP75. This model integrates all our results and is described in detail in the *Discussion* section. Furthermore, the physiological relevance is also addressed in this section: the regulation of Ca^{2+} transfer constitutes a mechanism for controlling mitochondrial dehydrogenase activity and, consequently, the electron transport chain and the capacity for ATP generation.

Since the pathological relevance can be inferred from these data, we also note in the *Discussion* the potential of targeting the STIM1–GRP75 interaction, as demonstrated in this study with a competitor peptide, to mitigate Ca^{2+} overload observed in neuronal cells under certain neurodegenerative conditions. However, this has not been included in the final figure, because it is a comment intended for consideration by our group and others for future experimentation, as this hypothesis still requires experimental validation in samples from patients.

Q8: *The ROS measurements using CellROX (which is not the most precise ROS sensor) indicate elevated ROS levels in STIM1-KO cells despite the reduced OCR. The underlying mechanism of this observation is not sufficiently addressed.*

A8: At this point, we would like to emphasize that determining the specific nature of the ROS (superoxide anion, H₂O₂, peroxynitrite, hydroxyl radical, etc.) is not the focus of this work, since this information is intended to support the observation of mitochondrial dysfunction. It should also be noted that the mechanism underlying this ROS overproduction may be multiple, and by no means contradicts a reduced electron transport.

When electron transport slows down, complexes I and III can become highly reduced, i.e., electrons accumulate in these complexes. The highly reduced state of these complexes increases the likelihood that electrons will react with oxygen, generating superoxide anion, instead of being transferred to other complexes of the ETC (complex IV). In particular, the phenomenon known as reverse electron transport (RET) at complex I can occur when substrates such as succinate are oxidized, and the electron acceptor pool (CoQ) is highly reduced. Under these conditions, RET constitutes a particularly potent source of ROS (this thought is already discussed in the Discussion section).

However, we would like to reiterate that mitochondrial defects beyond those directly arising from impaired Ca²⁺ transport from the ER—such as their direct consequences on mitochondrial dehydrogenases of the Krebs cycle—lie outside the scope of a study focused on describing a regulatory mechanism of Ca²⁺ flux between the ER and mitochondria.

Minor

Q9: *Given the small differences observed at only one time point, the conclusions drawn from Figure 4B should be moderated.*

A9: We have conducted additional lipid exchange assays to increase the sample size. Consistently, we observed that this lipid conversion is reduced in the absence of STIM1 in HEK293 cells, while a slight increase was monitored in U2OS cells. Given that this result leads us to suspect a potential cell-specific behaviour and considering the important relationship between lipid exchange and Ca²⁺ transport, we have decided to reserve these results for further study, where they will be examined in greater depth due to their biological significance.

Q10: *The y-axis scaling in Figures 5A-C should be standardized for better comparison.*

A10: We initially adjusted the scales to facilitate the visualization of individual data points. Nevertheless, we fully agree with the reviewer that presenting the three figures on comparable scales provides a clearer overall perspective. Accordingly, the scales have been modified in the new Figure 3.

Q11: *To accurately quantify ER calcium content, the area under the curve should be measured or a similar approach should be employed.*

A11: On this specific point, we respectfully disagree with the reviewer. The area under the curve is not a direct measure of Ca²⁺ content in the ER, since the relationship between fluorescence probe and Ca²⁺ concentration is not linear but logarithmic. The fluorescence of the ER-GCaMP6-210 probe, used in the previous Figure 5A–C, can be calibrated to convert fluorescence data into Ca²⁺ concentration, as indicated by the original authors of the probe (de Juan-Sanz J, Holt GT, Schreiter ER, de Juan F, Kim DS & Ryan TA (2017) *Axonal Endoplasmic Reticulum Ca²⁺ Content Controls Release Probability in CNS Nerve Terminals. Neuron* 93: 867–881.e6). We have followed this approach in the present study, citing the calibration procedure in the *Methods* section, as well as we did in our previous work (Pascual-Caro C, Orantos-Aguilera Y, Sanchez-Lopez I, de Juan-Sanz J, Parys JB, Area-Gomez E, Pozo-Guisado E & Martin-Romero FJ (2020) *STIM1 Deficiency Leads to Specific Down-Regulation of ITPR3 in SH-SY5Y Cells. Int J Mol Sci* 21: 6598), which was carried out in collaboration with Jaime de Juan-Sanz, lead author of the aforementioned article.

Q12: *The data presented in Figures 1-4 could be condensed into a maximum of one or two figures to improve clarity and conciseness.*

A12: In accordance with the reviewer's suggestion, figures have been consolidated as much as possible.

Q13: *What is the rationale to use U2OS cells for some experiments and HEK293 for others?*

A13: The purpose of employing two cell lines was to determine that the observed phenotypes are not cell specific. We would also like to add that we made an effort to perform all experiments with both cell lines; however, in many cases this was not feasible due to the inherent limitations of the cultures. Nevertheless, we ensured that the most critical experiments, and those that could potentially raise greater controversy, were conducted in both cell types.

Q14: *The use of HEK cells overexpressing STIM1-GFP does not appear to add substantial value. Since the same STIM1-GRP75 interaction was observed in WT cells, it may be advisable to remove Figure 1A.*

A14: On this point, we respectfully only partially agree with the reviewer. We believe it is important to validate the mitochondrial interactors identified by mass spectrometry in our previous study (PMID: 38224453). In that work, the tagged “bait” protein was STIM1-GFP; therefore, in the present study, the experiments shown in the Figure 1A were also performed using the same “bait,” STIM1-GFP, although we additionally included the interaction with endogenous STIM1. Furthermore, many of the experiments carried out to determine the interaction

region between STIM1 and GRP75 were performed with STIM1-GFP. For these reasons, we consider it appropriate to retain this figure.

Summary:

While the study addresses an interesting and timely topic, the current findings appear to be at a preliminary stage. Additionally, some conclusions are not fully supported by the experimental data. Further validation and additional controls are needed and would strengthen the main findings of this study.

We sincerely thank the reviewer for the time dedicated to evaluating this manuscript and for considering it both interesting and timely. While the initial version may have given the impression of being preliminary, the revised version is certainly not, as it confirms the previous conclusions through the incorporation of extensive new experimentation:

1. A new approach to assess ER–mitochondria contacts using ddFP has been included.
2. A new method to determine the localization of STIM1 in MAMs using ddFP has been added.
3. The dynamics of GRP75 responses to the conformational change of STIM1 and to the filling state of stores have been characterized.
4. The impact of the absence of STIM1 on the main regulatory checkpoint of the TCA cycle has been studied.
5. The effect of disrupting the STIM1-GRP75 interaction on mitochondrial steady-state Ca^{2+} levels has been determined.
6. Taken together, these results have allowed us to propose a mechanistic model explaining the role of STIM1 in regulating Ca^{2+} transfer between the ER and mitochondria.
7. All experiments include both technical and biological replicates, ensuring statistical significance.

For all these reasons, we believe that the final format of this manuscript—further strengthened by the constructive discussion with the reviewers—represents a valuable contribution for researchers working in this field.

Referee #2:

The present work by Oratos-Aguilera et al., focused in the role of STIM1 in the regulation of calcium influx from ER to mitochondria via contact sites. Here, the authors identified PTP51 and GRP75 as interactors of STIM1 thus modulating the mitochondrial calcium and function.

Using biochemical and cell biology methods the authors studied the interaction of STIM1, Grp75 and VDAC1/2/3 by co-immunoprecipitation endogenous and overexpressing STIM1-GFP. Then, subcellular fractionation was performed showing that STIM1 localize at the membrane contact sites together with TOM20 and ACSL4 and ERLIN2. Then, interaction with another MAM protein was analyzed (PTP51). PLA dots (VAPB-PTP51) were increased in STIM1KO cells which was rescued with the expression of STIM1 but not STIM2. Additionally, phospholipid transfer with radiolabeling of 3H-serine did not show significant differences in the synthesis while the ratio of PE/PS varied in STIM1KO cells. Next, the authors evaluated the calcium concentration at ER. Basal and after adding ATP+CCh showed decreased ER concentration in STIM1 KO cells while the cytosolic calcium levels were not affected. In addition, mitochondria calcium levels were studied using mito4x-GCamP6f. Cells treated with ATP+CCh didn't increase mitochondrial calcium level in the KO cells as well in resting and basal conditions. No changes in protein levels of IP3Rs were observed in the subcellular fractionation assays however the PLA dots (VDAC1 and IP3R31/2/3) were increased in the KO cells. Decreased mitochondria oxygen consumption was observed in STIM1 KO cells (basal and maximal) which was accompanied of a reduced ATP production. Total protein levels of mitochondrial complex I (CI-NDUFB8) were decreased in STIM1 KO cells. Then, cell death was analyzed. STIM1 KO cells were shown to increase the cell death while the rescue partially reduced the death induced with duramycin. Thapsigargin treatment affected the interaction with GRP75 and increased PLA dots (VAPB-PTP51). The addition of ATP-CCh decreased the interaction to GRP75 as well. Disease related mutation of STIM1-R429C was studied. This mutation affected the interaction with GRP75 as well the formation of PLA dots (VAPB-PTP51). Finally, several mutations on STIM1 were evaluated. Here, mutants lacking the residues from 551-685, 551-642 and 551-611 failed to interact. Also, deletion of the residues 668-674, 672-685 and 582-642 were important for the interaction with GRP75 (Mutatns:3,4,5,6,8 and 9). Here, the mutant 5 decreased the PLA dots of GFP-STIM1 and GRP75 while just expressing the residues 551-611 was enough to bind GRP75. Deletions of these residues increased the PLA dots VAPB-PTP51.

The manuscript in the current state is not recommended for revision. The data presented require further analysis and studies to support their conclusions/statements. Several interactions were characterized, however direct link among them has not been properly studied. Further analysis including appropriate methods to provide new evidence will improve the present manuscript.

Major concerns

Q1: Although the study of STIM1 as a MAM protein is interesting, the authors did

not provide convincing data to support their statements. Unbiased analysis of MAM proteins with current methodologies such as proteomics and localization analysis must to be added to support the present study.

A1: We appreciate the reviewer's constructive criticism regarding the request for proteomics data to determine the set of proteins at MAMs. This information has already been published (see PMID: 32424107), and the supplementary information of that article, which includes proteomic data, already identifies STIM1 as a resident MAM protein.

The novelty of our work does not lie in detecting STIM1 through omics approaches, but rather in identifying it using low-throughput experiments, which—by their nature—are higher-quality when it comes to localizing one or a few specific proteins, provided the appropriate controls are used.

We believe that using specific antibodies and proper negative controls (such as STIM1-deficient cells for immunoblot negative controls) provides a more reliable means to determine the presence of a particular protein (such as those reported in this study: STIM1, GRP75, ACSL4, etc.). This approach is not inferior to performing a proteomic analysis of MAMs, especially considering that MS-based experiments entail greater experimental error margins due to the inherent limitations of trypsinized peptide identification (mass fingerprinting).

To further confirm the presence of STIM1 in MAMs, we not only performed subcellular fractionation, but also PLA assays between STIM1 and PTPIP51, a mitochondrial marker (Figure 1), which indeed forms a tethering complex between the ER and mitochondria along with VAPB.

In addition, although this was not explicitly requested, we cloned the “B-monomer” coding sequence (from the ER-B construct – see PMID: 38327561) into a plasmid encoding STIM1, thereby generating the pair Mito-Green(A) and STIM1(B), which allowed us to assess STIM1–mitochondria contacts. These new results, presented in Figure 1 (panels J-K), demonstrate the localization of STIM1 at MAMs.

Q2: *STIM1 has been shown to acts as a calcium sensor regulating the gating from plasma membrane channels like Ora1. How does to modulate the calcium at the mitochondria? Here some evidence was indicating some interaction but direct functional assays and mechanisms of regulation were not provided.*

A2: At this point, we respectfully disagree with the reviewer's statement that “direct functional assays and mechanisms of regulation were not provided,” for several reasons:

1. **Functional assays were indeed provided**, including transport assays between the two organelles. These assess not only Ca^{2+} uptake by mitochondria but also Ca^{2+} release from the ER.

2. We also provide **functional assays measuring mitochondrial activity**, using Seahorse experiments.
3. Regarding regulatory mechanisms, we present here what is the **first proposed mechanism regulating Ca²⁺ transfer between the ER and mitochondria**, based on the interaction between STIM1 and GRP75 (Fig 1). This interaction is sensitive to Ca²⁺ levels in the ER, and thus to STIM1 folding (Fig 6), and is mediated specifically by the STIM1 551–611 region binding GRP75 (as demonstrated by competition assays with the 551–611 peptide – Fig 7 – and deletion of this region in STIM1 – Fig 7). Finally, this mechanism implies a dissociation of GRP75, leading to its release from MAMs (Fig 6.H.I). A summary of this mechanism is included in the final figure of the revised manuscript (Fig 8).

Although another reviewer (referee 3) stated: “Mitochondrial calcium loading depends on resting STIM1, and its activation by local ER Ca²⁺ decrease would disrupt STIM1-GRP75 interaction, leading to decreased ability of mitochondria to load,”—which suggests that the regulatory mechanism might have already been sufficiently clear in the original version—we have added a new summary figure that we believe will help a broader readership (Figure 8).

The reviewer also refers to “some interaction.” However, we provide clear evidence of interaction via co-IP (with GRP75), PLA assays, or both, as in the case of STIM1–PTPIP51. Moreover, we show that these interactions are dynamic and respond to physiological stimuli (e.g., ER Ca²⁺ depletion by thapsigargin, or stimulation with ATP + carbachol), and that they depend on a defined STIM1 domain, as described above. For all these reasons, we believe it is not accurate to describe this as merely “some interaction,” and it could in fact be considered the first functional evidence of STIM1–GRP75 interaction with consequences for mitochondrial function.

Q3: *The potential impact of STIM1 deletion over the formation of multiple contacts sites makes difficult the conclusions. Both IP3R3-VDAC and VAPB-PIPT51 membrane contact sites has been directly involved regulating calcium and lipid transfer respectively. How do the authors differentiate of an indirect effect of STIM1 deletion versus direct role regulating channel's function?*

A3: The most direct evidence for the role of STIM1 is that deletion of the 551–611 segment reduces its interaction with GRP75, and results in higher levels of ER-mitochondria contacts. This segment (STIM1 551-611) shows affinity for GRP75, as demonstrated by the immunoprecipitation of GRP75 following transfection of cells with the 551–611 peptide (Fig EV5.H). Unlike STIM1-KO cells, STIM1(Δ551–611) cells retain SOCE (see Fig EV5.G). However, these cells still exhibit an increased number of ER-mitochondria contacts (Fig 7.G), similar to KO cells. This suggests that the absence of SOCE in KO cells does not account for the increased ER-mitochondria contacts.

Furthermore, in the revised version of the manuscript we have included an additional experiment, not present in the original submission, in which induction of 551–611 peptide expression reduces Ca^{2+} transfer by disrupting the STIM1–GRP75 interaction. Altogether, these findings support the conclusion that the interaction STIM1–GRP75 is mediated by this domain (551–611) and that it is required to initiate Ca^{2+} transfer.

Q4: *Also, STIM1 role at plasma membrane can indirectly alter the calcium levels which will affect the calcium uptake capacity of the ER. Here, the data it is not convincing and further studies has to be evaluated.*

A4: In our view, the reviewer’s comment is somewhat imprecise, as it suggests that the function of STIM1 at the plasma membrane might affect the ER’s ability to refill Ca^{2+} stores, without providing further criticism beyond this statement. In response, we would like to clarify that we actually share this opinion, namely that STIM1–KO cells are indeed less efficient than WT cells in refilling ER Ca^{2+} . For this reason, we performed functional measurements to specifically address this situation:

1. In Fig 3, ER Ca^{2+} levels in WT and KO cells are shown. As expected, ER Ca^{2+} levels are slightly lower in KO cells due to the absence of SOCE.
2. The same figure also demonstrates that, despite statistical significance, the total amount of Ca^{2+} released from the ER is very similar in both conditions (Fig 3, panel D).
3. This ER Ca^{2+} release is sufficient to generate a nearly identical cytosolic signal (Fig 3, panel F), indicating that ER Ca^{2+} release is not impaired in KO cells.
4. Consistently, WT and KO cells display comparable levels of IP3R31/2/3 (Fig EV2A–D).

Taken together, these observations do not support the statement that “further studies have to be evaluated,” which we consider not justified in this context, as we have demonstrated the alteration of the ER Ca^{2+} concentration and how it is mobilized in response to IP3 generation.

Moreover, to further clarify any remaining doubts, the effects we observe cannot be attributed to the absence of SOCE, for several reasons:

1. The STIM1(Δ 551–611) mutant exhibits normal SOCE (Fig EV5.G) yet displays defective GRP75 interaction and increased ER–mitochondria contacts (PTPIP51–VAPB) (Fig 7.F–G).
2. SOCE is present when extracellular Ca^{2+} is added after ER store depletion. However, this SOCE does not restore the STIM1–GRP75 interaction (Fig 6.A–B).

Q5: Furthermore, the present work lacks in vivo studies using models to test its physiological relevance either including disease related mutations in primary cells or in a mouse model.

A5. At this specific point, we must clarify that the present work uses cellular models because it aims to define a regulation of Ca²⁺ trafficking at the molecular level, that is, revealing the participants in this trafficking and their mode of action. The immediate cellular impact of a defect in this Ca²⁺ trafficking being on the mitochondria and its energy generation is sufficiently evidence of its physiological impact.

Our viewpoint is that the use of a mouse model is not justified at this level, and should only be considered in a subsequent evaluation aimed at assessing the impact on various factors such as aging, cognitive capacity, physiology of exercise, etc. In other words, this work is entirely outside the scope of such models, as it is not required to define either the involvement of STIM1-GRP75-ITPR-VDAC on Ca²⁺ transfer for the first time, nor its regulation.

Would the option of having cellular lines with mutations in this region be desirable? It is clear that it would be advantageous to work with these lines, provided they are related to a pathological phenotype (derived from patients). However, it must also be considered that it is not possible to use just any cellular line or mutation, since STIM1 is involved in at least four different functions: (1) control of extracellular Ca²⁺ (PMID: 16005298) (2) activation of STING (PMID: 30643259), (3) DNA repair, PMID: 38224453) (4) in this work, we demonstrate that it is directly involved in Ca²⁺ trafficking between the ER and mitochondria. Each of these actions has a specific domain responsible for these functions (except for the case of activities 1 and 2, where the same domains are shared). Therefore, it is necessary to use a cellular line with mutations in the region defined here as key (551-611) and not in any other, so that we can study the exclusive effects of this region. This limitation, which is significant, explains why we do not yet have such specific cellular lines (unlike what happens with the multiple lines associated, for example, with Stormorken syndrome).

Minor comments

Q6: Several microscopy analyses were performed while the conclusions are based on proximity ligation assay. The authors should include other experiments to confirm their findings. Here, electron microscopy analysis and the use of fluorescence reporters (SPLICs for example) coupled to high throughput microscopy analysis must be included.

A6: We would like to clarify that the PLAs designed to detect ER-mitochondria contacts are based on the colocalization of the two proteins: VAPB and PTPIP51 (localized at the ER and mitochondria, respectively), following a published protocol (PMID: 28132811), a study led by Patricia Gomez-Suaga, co-author of the

present manuscript under review. This pair of proteins has been extensively used to evaluate such contacts, and therefore its validity cannot be questioned.

Nevertheless, in accordance with the reviewer's request, we designed alternative approaches to evaluate our previous conclusions. While electron microscopy offers high 2D spatial resolution, it lacks the 3D sensitivity (along the Z axis) that can be achieved with fluorometric techniques. In this regard, and in response both to Reviewer 1, 4, and to the Reviewer 2, who is suggesting the use of SPLICS, we have performed new experiments using dimerization-dependent fluorescent proteins (ddFPs) targeted to mitochondria and the ER, as recently reported in PMID:38327561.

Cells were transfected for the expression of Mito-Green(A) and ER(B), such that the reconstitution of A-B products, indicates ER-mitochondria contact sites. For quantification, individual cells were analyzed by microscopy, measuring green fluorescence intensity, which was normalized to anti-TOM20 immunolabeling (in red) as a mitochondrial marker. The results, presented in the new Figure 2, further support our previous conclusions, namely: (1) STIM1-KO cells display increased levels of ER-mitochondria contacts (Figure 2.D-E), and (2) depletion of ER Ca²⁺ stores induced by ATP+CCh in Ca²⁺-free medium also increases ER-mitochondria contacts compared to resting conditions (Fig 6.G).

We respectfully disagree with the suggestion of using high-throughput experiments at this point, as these approaches are generally more suitable for preliminary screenings (such as siRNA screenings), which subsequently require validation through more detailed and accurate low-throughput techniques. In this context, we believe that the use of dedicated ddFP provides additional value to the manuscript and strengthens our conclusions.

In addition, although this was not explicitly requested, we cloned the "B-monomer" coding sequence (from the ER-B construct) into a plasmid encoding STIM1, thereby generating the pair Mito-Green(A) and STIM1(B), which allowed us to assess STIM1-mitochondria contacts. These new results, presented in Figure 1 (panels I-K), demonstrate the localization of STIM1 at MAMs.

Q7: *Several co-immunoprecipitation assays were provided; however, no quantifications were added. This makes hard to conclude.*

A7: We thank the reviewer for this valuable comment. In this revised version, we have included quantifications for the coIP assays presented.

Q8: *In the subcellular fractionation assays, several loading controls are missing and proper MAMs, ER and mitochondria markers are not used. Cytosolic marker it is not provided, which make difficult to conclude about the GRP75 recruitment or binding since it is an associated cytosolic protein in the later fractionation panels.*

A8. We thank the reviewer for the suggestion to include additional markers to improve the interpretation of the subcellular fractionation. In this regard, Figure 1.E now shows these markers along with a new purification procedure (which has been performed up to five independent times). The newly included markers are p38MAPK as a cytosolic marker, IP3R1/2/3 as an ER marker, ACSL4 as a MAM marker, and TOM20 as a mitochondrial marker.

Q9: *Figure 1: Validation of mitochondrial SIMT1 interactors by co-immunoprecipitation. Here co-IP experiments were presented however, fractionation assays, colocalization assays of STIM1, ER and mitochondria are missing.*

A9: We partially agree with the reviewer. The subcellular fractionation experiments were already included in the manuscript (previously Figure 2, now Figure 1). Regarding the suggested colocalization assays of STIM1, ER, and mitochondria—although the request is not entirely clear—we understand it could refer to a triple-labelling approach to assess colocalization. However, fluorescence microscopy colocalization is limited by a spatial resolution of approximately 80–100 nm, which makes it unsuitable to visualize organelle contacts. For this purpose, techniques such as SPLICS, as mentioned by the reviewer, are more appropriate. Since SPLICS is not reversible, we opted instead, as also explained in response A6, to use dimerization-dependent fluorescent proteins (ddFP, see PMID: 38327561). This reversible contact-FP probe system allows monomers targeted to distinct cellular membranes to dimerize when in close proximity (~10–30 nm), resulting in increased fluorescence. The results obtained with ddFP support the idea of an increase in ER–mitochondria contacts in STIM1-KO (Fig 2.D-E). Furthermore, using ddFP, we were also able to reconfirm the presence of STIM1 in MAMs by labelling STIM1 and mitochondria (Fig 1.I-K), which resulted in fluorescence reconstitution with this pair.

Q10: *The endogenous IP shows the interaction of STIM1 with GRP75. However, the endogenous band of SIMT1 is presented in the IP fraction does not appear in the WCL. The authors should either remove this data or provide a different experiment showing the STIM1 detection in the inputs.*

A10: We would like to clarify that in the IP there is an enrichment of STIM1, since the assay was performed using an anti-STIM1 antibody. Consequently, it is expected to observe the STIM1 band in the IP but also expected *not* to detect the STIM1 band in the total lysate when using an anti-GRP75 antibody. Therefore, we believe there is no error and that the blot should not be modified, as the absence of the STIM1 band in the input (or WCL) is correct, and only the specific GRP75 band should be detected. In the case of the IP, this situation is comparable to the frequent detection of heavy- and light-chain antibody bands in IPs where antibodies are not covalently bound to the beads: the high concentration of the immunoprecipitated protein (STIM1 in this case) can be revealed when probed with an antibody directed against another protein (GRP75).

Q11: *Figure 2: Evaluation of the presence of STIM1 in MAMs. The fractionation assay is not completely clear. The authors produced crude mitochondria which contains MAMs and pure mitochondria it is not provided. Here, the PLA dots of PTP51 and STIM1 was analyzed, however other MAM proteins can be included such as VDACS or TOMs to better illustrate STIM1 localization as MAM protein.*

A11: The fractionation procedure followed is the same as that published by Montesinos and Area-Gomez (PMID: 32183966), who are co-authors of the current manuscript. The reviewer is asking for the fraction containing pure mitochondria; however, in this work we did not need this fraction for any of the analyses performed, and therefore it was not included.

Regarding the mitochondrial markers in PLA assays, in addition to the PLA between STIM1 (ER) and PTP51 (mitochondria), which is shown in Fig 1.H, the reviewer requested a PLA between STIM1 and TOM20 or VDAC. This request cannot be fulfilled, as PLA must be carried out with proteins that interact with STIM1, and STIM1 does not interact with VDAC1/2/3, as we show in Fig 1.A, and there is no evidence for an interaction with TOM20. Nevertheless, we have implemented an alternative approach, the reversible contact-FP probe system, which allows us to tag STIM1 with gene B and mitochondria with gene A, in order to determine the increase in fluorescence derived from the A–B interaction. This method enables us to confirm the presence of STIM1 at ER–mitochondria contact sites (see Fig 1.I-K).

Q12: *Figure 3: Analysis of the functional consequences of STIM1 deficiency on MAMs. It is known that STIM1 oligomerizes and interacts with ORA1. How the authors connect an increased activation of SIMT1 and the amount of PLA dots VAPB and PTP51? The authors should include more dynamics of the contacts upon SIMT1 activation or ER stress, not only chronic deletion.*

A12: To explain the reason for an increase in ER–mitochondria contacts, as determined by a PLA between VAPB and PTP51, we mentioned in the manuscript that a reduced capacity for Ca²⁺ transfer—whether due to a deficiency in STIM1 (in STIM1-KO cells with inactive SOCE), in cells expressing STIM1-R429C (inactive SOCE), or cells expressing STIM1-Δ551-611 (active SOCE)—is likely the result of a compensatory effect. In other words, more contacts are established to enhance Ca²⁺ transfer, which is inefficient under these experimental conditions, since mitochondria require a certain level of Ca²⁺ uptake to maintain TCA cycle activity. Otherwise, we would cease to observe these mitochondria, which would eventually undergo mitophagy.

In response to the reviewer's request for dynamic assays, in this revised version of the manuscript we have included a new experiment aimed at determining how store depletion modifies GRP75 levels associated with MAMs. We believe this experiment is more informative than simply monitoring contacts, since such contacts would only represent the cellular response to Ca²⁺ deficiency. In

contrast, the direct observation of GRP75 behaviour has allowed us to propose a model explaining the molecular mechanism controlling Ca^{2+} transfer between the ER and mitochondria. The evaluation of GRP75 levels in MAMs in response to store depletion triggered by ATP + carbachol is presented in the new Fig 6 (panels H-I), while the model explaining the control mechanism of Ca^{2+} transfer is shown in the final figure (Fig 8).

Q13: *Figure 4: Analysis of phospholipid synthesis in the crude mitochondrial fraction. Controls of purity of mitochondrial fraction has to be included. To evaluate it is crude or pure mitochondria. This assay it seems interesting, why the authors did not directly measure all the lipid profile of mitochondria not only the PE/PS.*

A13: At this point, the reviewer requests a control to assess the purity of the mitochondrial fraction, in order to determine whether it is a pure fraction or it contains other membranes—what we refer to here as crude mitochondria. In response to this suggestion, we must clarify that such a control cannot be performed, since the lipid-exchange analysis between the ER and mitochondria cannot be carried out with pure mitochondria, as there would then be no interaction with the ER. This is the reason why this assay is performed with crude mitochondria. Moreover, the purity of this fraction, as well as of the others, is already shown in the blot presented in Fig 1.E, Fig 4.A or in Fig 6.H. Nevertheless, and as we have indicated in our responses to other reviewers, we have conducted new biological replicates with the aim of determining the differences between data groups with greater precision. The experiments were carried out in both cell lines to exclude cell-specific processes. We have observed that this lipid conversion is reduced in the absence of STIM1 in HEK293 cells. However, a slight increase in U2OS cells was observed. Given that this result leads us to suspect the occurrence of cell-specific behaviour and considering the important relationship between lipid exchange and Ca^{2+} transport, we have decided to reserve these results for further study, as they will be examined in greater depth due to their biological significance.

The lipid-exchange assay is a dynamic pulse-chase methodology that enables monitoring of the incorporation of radioactive serine into phosphatidylserine within the ER, which is subsequently transferred to mitochondria and converted into phosphatidylethanolamine. However, while mitochondrial lipidomic profiling under these conditions would be scientifically informative, it does not allow for the specific attribution of lipid changes to ER-mitochondria connectivity and is, therefore, out of the scope of our study. The lipid-exchange assay is indeed a valuable approach to monitor ER-mitochondria contacts, but a global lipidomic analysis constitutes a completely different objective from what is addressed in this work, even though it would certainly be of great interest to pursue.

Q14: *Figure 5: Analysis of ER-to mitochondria Ca^{2+} shuttling in the absence of STIM1. Expression of the calcium sensors should be provided. Representative images are missing. Authors should add these analyses.*

A14: In response to the reviewer's request, we have incorporated a representative image for the fluorophores used in this study. These images are now included in the manuscript at the point where each fluorophore is first mentioned, to ensure clarity and to fully address the reviewer's suggestion. The mitochondrial localization of other fluorophores, whose images have not been included here, such as 4mtD3cpv, was described in our previous work (PMID: 30088035).

Q15: *Figure 6: Studying the IP3R-GRP75-VDAC axis in the absence of STIM1. How do the authors explain that contradictory to an increased number of contacts sites in SIMT1 KO cells, the function, meaning the calcium levels at ER and mitochondria are not increased rather decreased? Here inhibitors of the main ER pump can be used (thapsigargin) and measure the calcium changes like in the Fig 5F.*

A15. The answer to this question is already provided in the manuscript: we explain the data (without there being any contradiction, in our view) by clarifying that, although there is an increase in ER-mitochondria contacts, levels of IP3Rs do not change between STIM1-KO and WT cells, and even though a PLA assay between IP3R1/2/3 and VDAC shows a slight increase in contacts in STIM1-KO cells, these complexes are not functional as they lack interaction with GRP75, in accordance with the critical role of GRP75 in ensuring proper ITPR3-VDAC coupling (PMID: 29367884). This conclusion is inferred from the lower levels of GRP75 found in the MAMs of KO cells, or from the decrease in GRP75 levels observed in WT cells treated with ATP+CCh (in extracellular medium without Ca²⁺). These data demonstrate that there is a dissociation of GRP75 from MAMs when (1) STIM1 is absent, or (2) when STIM1 is present but in an open conformation. For this response, the use of additional inhibitors is not necessary, since store depletion with ATP+CCh in Ca²⁺-free medium already maintains STIM1 in this open conformation, given that intracellular stores cannot be refilled (there is no SOCE in the absence of extracellular Ca²⁺).

Q16. *Figure 7: Bioenergetic analysis of WT and STIM1 KO cells. How do the decreased OCR and ATP production are associated with the membrane potential and the ROS produced in mitochondria? This will be important to measure in the STIM1 ko cells and the mutant to evidence physiological relevance.*

A16. In response to the reviewer's request, we have measured the mitochondrial membrane potential in STIM1-KO cells. The new data are presented in the revised Figure 5.N–O, where we assessed the potential using MitoTracker Red CMXRos (a potentiometric probe) and, in parallel, MitoTracker Green, which is not sensitive to membrane potential and serves as a control for mitochondrial mass. The results reveal a slight but significant loss of mitochondrial membrane potential in the absence of STIM1, a finding consistent with our previous report in SH-SY5Y STIM1-KO cells (PMID: 30088035).

Q17. *Figure 8: Total levels of proteins from the mitochondrial respiratory chain complexes. Since the authors observe decreased levels of Cl-NDUF8 in the STIM1*

KO cells, how do they discriminate on changes in the mitochondrial mass or biogenesis? Additional studies to support these observations has to be included.

A17. As suggested by the reviewer, in the revised version of the manuscript we assessed mitochondrial mass in WT and STIM1-KO cells in two ways: (1) by measuring mitochondrial DNA levels using real-time PCR of the *mt-ND2* gene, and (2) by MitoTracker staining followed by flow cytometry analysis. The results are shown in Fig EV2.H-I.

Q18. Figure 9: Analysis of oxidative stress and sensitivity to duramycin. There are better methods to assess cell death. The authors did evaluate cell proliferation instead? Also, which cell death is increased? Ferroptosis? Apoptosis? Specific analysis has to be included to further conclude that STIM1 KO decreases viability of the cells.

A18. Regarding the determination of sensitivity to duramycin, we have decided to remove this experiment, as we agree with another reviewer that it represents an overly indirect approach to establish an alteration in global lipid transport in the absence of STIM1. While it is true that alternative methods exist to assess such alterations in lipid transport, we believe that, conceptually, this evaluation falls outside the central message of the article. For this reason, we have focused on establishing the mechanism by which STIM1 regulates Ca^{2+} transfer between the ER and mitochondria, with new experiments and new techniques/approaches, and leaving aside the potential impact on lipid transport between the plasma membrane and the ER. Consequently, determining the type of cell death in response to duramycin does not add significant value to the present work, although it would certainly be of interest in studies specifically focused on lipid transport.

Q19. Figure 10: Analysis of the STIM1-GRP75 interaction after ER Ca^{2+} depletion.

A19: The reviewer did not request any changes regarding this figure.

Q20. Figure 11: Analysis of the interaction between STIM1 (R429C) and GRP75 by co-immunoprecipitation.

A20: The reviewer did not request any changes regarding this figure.

Q21a. Figure 12: Analysis of the co-precipitation between STIM1 mutants with C-terminal deletion and GRP75. Here, several deletions were generated, however a convincing model of interaction has to be included. Molecular simulations and alpha fold methods can be included to between decipher the regions of interactions and support the direct interaction and not misfolding of the protein or aggregates.

A21a. In response to this question, we performed a prediction using AlphaFold3, employing STIM1 and GRP75 as interacting partners. This simulation, which we

include as part of this response (see below), identified several possible interaction sites between GRP75 and STIM1, consistently located within the intrinsically disordered region (IDR) of STIM1. This region does not show a defined structure and therefore has low predictive accuracy in AlphaFold. Nevertheless, AlphaFold3 predicted a possible contact site between residues 676–680 of STIM1 and GRP75. Although other regions are also possible, and this is not the only one, the motif comprising residues 676–LKIFK–680 yielded the highest ipTM (interface predicted Template Modelling) score.

AlphaFold prediction:

Chord diagram showing the predicted interaction between GRP75 sites and the motif LKIFK:

Based on this prediction, we generated STIM1-KO HEK293 cells with inducible expression of STIM1(LKIFK>AAAA), in which residues 676–680 were mutated to Ala without altering protein length. CoIP analysis of STIM1 with GRP75 showed that this mutation did not affect coprecipitation between STIM1 and GRP75.

This result allows for at least two possible interpretations:

1. The 676–680 region of STIM1 does not play a critical role in the interaction with GRP75, and the AlphaFold3 prediction lacks experimental support.
2. The 676–680 region of STIM1 may indeed represent one of the interaction sites with GRP75, but within a multi-motif interaction context. It is important to note that the STIM1 sequence downstream of residue 485 belongs to an IDR (intrinsically disordered region), a region where motif spacing may be more relevant than sequence conservation. IDRs often harbour multiple interaction motifs that enable simultaneous or sequential binding to different regions of target proteins or complexes (see PMID: 25531225). In this context, disruption of a single motif might not significantly impair binding. While this remains speculative, mapping the interaction through systematic deletions would be the most appropriate approach.

In summary, the interaction between STIM1 and GRP75 may be flexible, and only removal of the most critical motifs would allow us to clearly define the interaction. Importantly, our data identify a region (residues 551–611) that is sufficient for GRP75 binding. Indeed, a synthetic peptide corresponding to residues 551–611 can compete for this interaction and even reduce Ca^{2+} transfer between the ER and mitochondria, providing substantial experimental support for the hypothesis we propose here.

Q21b. Do the mutants distribute at the ER and can oligomerizes correctly? Since the authors showed initial interactions of STIM1 with other MAM proteins, do these mutants affect the interactions with VDACs?

A21b. Since the deletion of residues 551–611 lies outside both the SAM and SOAR regions, this mutant does not affect SOCE (as we have already shown in Figure EV5.G), nor does it impact oligomerization (as presented in the new Fig EV5.F), which is consistent with its lack of effect on SOCE. In contrast, we have shown that the R429C mutant results in a loss of SOCE (Figure EV4.C) due to impaired multimerization (Fig EV4.E), as previously described in the original reference (PMID: 25918394).

The other mutants used in this study are partial constructs that were generated to perform the coIP with GRP75. These were only intermediate truncations used to ultimately define the 551–611 region. For this reason, further analysis of these mutants is not necessary, as they did not provide additional crucial information beyond confirming the importance of the 551–611 region.

Regarding the reviewer's comment on the interaction with VDAC, we believe this to be a misunderstanding. As shown in Figure 1, VDAC is not an interactor of STIM1. Therefore, testing the interaction of the mutants with VDAC is not applicable.

Q21c. *Also, did the authors determined that the GRP75 it is actually contributing to the binding to VDACS or PTPIP51? Or STIM1 can directly binds these proteins?*

A21c. We would like to reiterate that STIM1 does not directly bind to VDAC, as the coIP experiments presented in Figure 1 clearly rule out such an interaction. Regarding PTPIP51, the reported association was suggested by a PLA, which does not necessarily imply a direct interaction. Whether this is direct or indirect is not relevant for the present study, and although it may be of general interest, it falls outside the scope of this work.

In addition, we do not address here the contribution of GRP75 to the STIM1–PTPIP51 interaction for two reasons: (1) it lies completely outside the objectives of this study, as we have already stated; and more importantly, (2) addressing this question would require mutating GRP75 (and not STIM1) to determine whether GRP75 mediates this interaction. As we explained in our response concerning the AlphaFold3 prediction, the GRP75–STIM1 interaction occurs within a highly structured globular region of GRP75. Mutating this region would disrupt the structural integrity of GRP75, rendering such an experimental strategy unfeasible.

General

Q22. *The entire workflow of the manuscript it is confusing. The figure 1 and 2 should be together as the text description is separated but there is not so much data. In general, there are too many figures some with a lot of data and the majority with very little or no data. The data can be mentioned but only the most relevant information in the main figures.*

A22. We believe that the comment suggesting some figures do not contain data is not entirely fair, as all figures provide relevant information. That being said, we acknowledge that some figures could indeed be merged. Our figures are organized conceptually by topic, and extensive merging would make them more difficult to interpret due to the high density of data presented. Nevertheless, in response to the reviewer's request, the figures have been merged as much as possible (from 12 to 7 figures + final scheme).

Q23. *Fluorescence images are not documented and schemes of work and experimental design should be added to better follow the study.*

A23. For each of the fluorophores used, we have included an image documenting the recorded fluorescence signal, whether corresponding to Ca²⁺ sensors or ddFP assays. The basic function of ddFP has also been described with a diagram, as well as the purified fractions in the isolation of MAMs, to clarify the sequence of events for the reader.

Q24. *The intro mentions general aspects of the MAM associated proteins and mechanisms. Here, more directed introduction can be included. Special focus to the proteins studied in the present work and the functionality or what it is known of STIM1.*

A24. At the reviewer's request, we have expanded the Introduction to provide additional details on STIM1 and its function as a regulator of Ca²⁺ channels.

Q25. *The figures of the co-immunoprecipitation are named GFP-pulldowns were should be GFP-IP since is not purified GFP-STIM1.*

A25. We believe there is no error here, as the term *pull-down* is generic and can be used for both antibody-driven precipitations and affinity resins. Nevertheless, at the reviewer's request, we have replaced the label "GFP pull-down" with "GFP-IP."

Q26. *The titles of the legends should contain the results rather than the method. It makes hard to connect results to the conclusions and it is not so informative.*

A26. We thank the reviewer for this suggestion. We have modified the titles and figure legends accordingly, which we believe improves the clarity and overall readability of the manuscript.

Referee #3:

The manuscript entitled, "STIM1 Regulates Calcium Flux Between the Endoplasmic Reticulum and Mitochondria at Contact Sites" is an intriguing study offering a previously unidentified role for STIM1 in control of MAM formation and ER-Mitochondria calcium transfer. Essentially, what is proposed is that STIM1 binds constitutively to GRP75 within the mitochondria and that these interactions are lost in response to ER calcium depletion. The implications of this study are highly significant, although some modifications needed. Hence, if mitochondrial calcium loading depends on resting STIM1, then its activation would decrease the ability of mitochondria to load. As such, mitochondrial calcium loading would be predicted to initially be highly efficient, but would then decrease as the ER depletes of calcium (which would disrupt MAMs for the purposes of calcium transfer).

Q0: We are very grateful to the reviewer for capturing the essence of our study with accuracy and insight. Indeed, this summary could well serve as the legend for the new final figure, which illustrates the proposed molecular mechanism for the regulation of Ca²⁺ trafficking between the ER and mitochondria.

Specific Comments:

Q1. *In discussing figure 4B there are statements suggesting that there are significant differences in PE/PS ratio at 10 minutes, but not 20 minutes. However, the true PE/PS ratio at 20 minute group may well follow the same pattern as at 10 minutes; the data is skewed by sampling error and I would encourage the authors to repeat this work and determine what is actually correct. The fact that the data is normalized to the 10 minute WT datapoint only exacerbates this issue of "inconsistent error" amongst groups.*

I would also add that PE levels rather than PE/PS ratio seems to be the primary altered group, even if your work didn't pass statistics. More clarity on experimental findings vs. effect of STIM1 on MAM function is needed. Perhaps consult a statistician to help you design a new experiment based on this one.

A1: For this specific experiment, we have conducted new biological replicates with the aim of determining the differences between data groups with greater precision. The experiments were carried out in both cell lines to exclude cell-specific processes. We have observed that this lipid conversion is reduced in the absence of STIM1 in HEK293 cells. However, a slight increase in U2OS cells is observed. Given that this result leads us to suspect the occurrence of cell-specific behaviour and considering the important relationship between lipid exchange and Ca²⁺ transport, we have decided to reserve these results for further study, as they will be examined in greater depth due to their biological significance.

As an aside, "=min and '=hrs.

A1: We appreciate the reviewer's observation and would like to clarify that the symbols (') and (") are often used for minutes and seconds (as we noted, ' = min and " = sec). In any case, we have removed this figure, so no changes are applicable. We thank the reviewer for helping us improve the clarity of our figures.

Q2. *In figure 5, many of the resting ER calcium concentrations are below 250 uM. At these concentrations, STIM1 should be active. How do you explain this (important to recognize that calibrations are often of questionable reliability)? In addition, it would have been helpful to at least reference the techniques used to measure ER calcium within the results section. Also, was figure 5D performed without extracellular calcium? If so, this should be stated in the results before stating the outcome of the experiment.*

A2: The reviewer is correct in noting that the calibration may not be entirely precise, since—as we detailed in the *Methods* section—it depends on reaching a minimal fluorescence value after EGTA treatment, which can be rather aggressive for cultured cells. As a result and based on our extensive experience calibrating different fluorometric probes across various cell types, it is possible that in some cell types with low adherence to the substrate, the true F_{min} value is not fully achieved, but rather values very close to F_{min}. Consequently, fluorescence readings may be slightly closer to F_{min} than expected. Although the resulting error should be minimal, values could indeed be slightly underestimated. Importantly, the differences observed between WT and KO cells would still remain and would not differ substantially from those shown in the former Figure 5. For this reason, we have now included in the *Result* section a reminder of the calibration approach used in this experiment.

Regarding the (former) Figure 5D, the reviewer is right that it was not explicitly stated that the measurement of ER Ca²⁺ efflux was performed in the absence of extracellular Ca²⁺. This information has now been added to the *Results* text and to the panel in Figure 3.

Q3. *Complex I produces superoxide; why would loss of complex I lead to increase ROS (the conclusion of figure 8)?*

A3: We understand that this is a point that may raise discussion; however, we also believe that it falls outside the scope of the present study. The main reason is that the primary consequence of impaired mitochondrial Ca²⁺ uptake is the reduction in the activity of TCA cycle dehydrogenases, as we have shown in the revised version of the manuscript by assessing phospho-pyruvate dehydrogenase complex levels (Fig 5.L-M).

The explanation for increased ROS production (particularly superoxide anion) under conditions of reduced electron flow through the ETC can be highly diverse. For this reason, we believe this topic lies entirely beyond the scope of the current article, as it would involve evaluating several hypotheses. One such hypothesis is that the deactivation of complex I is involved in the superoxide burst (PMID:

28511347). In acute hypoxia, this phenomenon has been linked to the Na⁺/H⁺ antiporter activity of complex I.

In general, reduced electron transport through the ETC can increase superoxide levels due to electron leakage and the incomplete reduction of oxygen during impaired mitochondrial respiration (PMID: 20430626). Superoxide production can also occur at complex I under specific conditions such as reverse electron transport (RET), particularly when the CoQ pool is highly reduced. To explore this latter possibility, we would need to focus on the role of succinate oxidation as the most plausible source of CoQH₂ accumulation. However, as we have mentioned, this would deviate from the scope of the present study, as the aforementioned scenarios remain possibilities, and the observed phenotype could result from a combination of these mechanisms. This goes beyond the central message of the manuscript, which is specifically focused on the detailed regulation of Ca²⁺ transport at MAMs. However, some of these considerations are already present in the *Discussion* section with the aim of encouraging further investigation by other research groups working in this field.

Q4. *The duramycin experiment in figure 9 is difficult to interpret as completed. There are much better approaches to demonstrate changes in the asymmetry of PS and PE. Susceptibility to cell death provides minimal insight into what has actually occurred in the cell.*

A4: We fully agree with the reviewer. This experiment only provides a phenotypic characterization data that does not significantly contribute to understanding the molecular mechanisms underlying the defect in Ca²⁺ transport between the ER and mitochondria. Therefore, we have decided to remove this figure along with the accompanying text. Importantly, this removal does not weaken any conclusions nor alter any of the previously presented messages but rather helps to further focus the manuscript on its central message.

Q5. *In figure 11, in describing STIM1-R429C, it is described as "constitutively unfolded". That is a confusing and not fully accurate description of this mutation. Actually, in this case, I believe that the construct was used correctly and supports the paper; it needs to be described better. Indeed, the R429C mutation is in CC3 within SOAR. As such, this is a mutant that primarily targets the site of interaction between STIM1 and Orai1. SOAR also has putative roles in STIM1 multimerization. Linking multimerization to STIM1-GRP75 would strengthen the conclusions of this study.*

A5: We agree that a more detailed description of the STIM1-R429C mutant is necessary. This mutant was initially characterized in the laboratory of Stephan Feske (PMID: 25918394). That group summarized the effect of this mutation as follows: *The C-terminal domain of STIM1 exists in a closed conformation that is maintained by R429 and the coiled-coil domain 3 (CC3). After store depletion, the C-terminal domain undergoes a CC3-dependent conformational change, resulting in exposure of the polybasic domain, STIM1 translocation to ER-plasma*

membrane junctions, oligomerization, and binding to the plasma membrane channel ORAI1. However, the STIM1-R429C mutant has an extended conformation with an exposed polybasic domain and constitutive localization at ER-PM junctions but fails to oligomerize and bind to ORAI1 due to destabilization of the CC3 structure (PMID: 25918394).

Consequently, this mutant exhibits several alterations that are directly relevant to our study: (1) an extended C-terminal structure, (2) loss of multimerization, and (3) absence of SOCE activation.

As the reviewer correctly pointed out, these features strongly support our message: the interaction between GRP75 and STIM1 is favoured when STIM1 adopts a folded structure in its C-terminal domain. By contrast, destabilization and extension of the C-terminal domain, even in the absence of multimerization, significantly reduces the interaction in the absence of its partner. Therefore, in line with the reviewer's suggestion, we have incorporated a more detailed description of this mutant in the text accompanying Figure 6 as well as in the last part of the *Discussion*.

Finally, we are grateful to the reviewer for their thoughtful comments, constructive criticism, and valuable suggestions. Taken together, these recommendations have substantially improved the manuscript.

Referee #4:

Mitochondria and ER directly communicate and coordinate functions at mitochondria-associated ER membranes (MAMs), regions of close membrane apposition tethered by proteins. To date, MAMs have been demonstrated to enable the bidirectional transport of signaling molecules, coordinate biosynthetic processes. However, the factors that mediate these functions remain little understood. Here, the authors show that the ER Ca²⁺ sensor STIM1-known for its role in activating plasma membrane Ca²⁺ channels-localizes to MAMs. The loss of STIM1 impaired PE synthesis, disrupted ER-to-mitochondria Ca²⁺ transfer, and decreased maximal mitochondrial respiration and ATP production. Although interesting, the authors claims of the role of STIM1 in mitochondria-ER communication are unsubstantiated.

Major Comments:

Q1: *Role of STIM1 in contact sites: The authors claim that the loss of STIM1 leads to increased contact sites. The main readout for contact sites is the PLA assay. In most of the figures where the PLA assay is shown, it is unclear whether the puncta correlate with ER/mitochondria. To substantiate their claims of the role of STIM1 and the relevant mutants analyzed in contact sites, the authors should perform electron microscopy. This will allow them to assess not only the length of the contact sites, but also the ER-mitochondria distance which are both important parameters. This is especially important given previous work showing there were no differences in STIM1 KO cells (Henke et al 2012).*

A1: At this point, we would like to clarify that the PLAs designed to detect ER-mitochondria contacts are based on the colocalization of the two proteins: VAPB and PTPIP51 (localized at the ER and mitochondria, respectively), following a published protocol (PMID: 28132811), a study led by Patricia Gomez-Suaga, co-author of the present manuscript under review. This pair of proteins has been extensively used to evaluate such contacts, and therefore its validity cannot be questioned.

Nevertheless, in accordance with the reviewer's request, we designed alternative approaches to evaluate our previous conclusions. While electron microscopy offers high 2D spatial resolution, it lacks the 3D sensitivity (along the Z axis) that can be achieved with fluorometric techniques. In this regard, and in response both to Reviewer 1, 4, and to the Reviewer 2, who explicitly suggested the use of SPLICS, we have performed new experiments using dimerization-dependent fluorescent proteins (ddFPs) targeted to mitochondria and the ER, as recently reported in PMID:38327561.

Cells were transfected for the expression of Mito-Green(A) and ER(B), such that the reconstitution of A-B products indicates ER-mitochondria contact sites. For quantification, individual cells were analyzed by microscopy, measuring green fluorescence intensity, which was normalized to anti-TOM20 immunolabeling (in

red) as a mitochondrial marker. The results, presented in the new Figures 2 and 6, further support our previous conclusions, namely: (1) STIM1-KO cells display increased levels of ER-mitochondria contacts (Figure 2.D-E), and (2) depletion of ER Ca^{2+} stores induced by ATP+CCh in Ca^{2+} -free medium also increases ER-mitochondria contacts compared to resting conditions (Fig 6.G).

In addition, although this was not explicitly requested, we cloned the “B-gene” coding sequence (from the ER(B) construct) into a plasmid encoding STIM1, thereby generating the pair Mito-Green(A) and STIM1(B), which allowed us to assess STIM1-mitochondria contacts. These new results, presented in Figure 1, demonstrate the localization of STIM1 at MAMs.

Q2: *Are the effects of STIM1 simply mediated by the loss of SOCE? STIM1 activates plasma membrane Ca^{2+} channels. Thus, all of the consequences of STIM1 deletion may be a result of impaired SOCE.*

A2: This is indeed an interesting question, as it could represent the primary hypothesis to be tested in the case of STIM1. However, the data presented in the manuscript already demonstrate clearly that the phenotype observed in STIM1-deficient cells, with respect to Ca^{2+} transfer between the ER and mitochondria, is **not due to the absence of SOCE**. The reasons are as follows:

1. The STIM1(Δ 551-611) mutant displays normal SOCE (see Fig EV5.G), although it shows impaired interaction with GRP75 and increased VAPB-PTPIP51 contacts, similar to what is observed in STIM1-KO cells.
2. The STIM1-GRP75 interaction is not dependent on SOCE, but rather on ER Ca^{2+} levels and the conformational state of STIM1. This is demonstrated in the experiment shown in Figure 6: after depletion of ER stores with thapsigargin, STIM1-GRP75 interaction decreases. When extracellular Ca^{2+} is subsequently added, cytosolic Ca^{2+} levels rise via SOCE, which is fully functional in WT cells. However, this Ca^{2+} influx does not restore STIM1-GRP75 interaction. The increase in cytosolic Ca^{2+} levels upon Ca^{2+} addition is confirmed in Fig EV4.C (WT – black line), an experiment that was carried out with the same sequence of treatments (store depletion with thapsigargin in the absence of extracellular Ca^{2+} , followed by Ca^{2+} addition to assess SOCE).

At this point, it is important to recall that due to the presence of thapsigargin in the assay, even if SOCE is active, ER refilling does not occur, as SERCA remains inhibited. As a result, STIM1 remains in its open (or active) conformation, in which it does not interact with GRP75 (Figure 6).

3. The STIM1-R429C mutant does not undergo multimerization (Fig EV4.E) but instead adopts an extended conformation (PMID: 25918394), mimicking the conformation of STIM1 under ER depletion. This occurs despite the absence of changes in ER Ca^{2+} levels between WT cells and cells

expressing STIM1(R429C), as clearly shown in Fig EV4.D. Because we do have lower levels of coprecipitation between STIM1-R429C and GRP75, this strongly suggests that the conformation of STIM1 is the main determinant regulating its interaction with GRP75.

4. Regarding Ca^{2+} transfer between the ER and mitochondria, the absence of SOCE cannot account for the differential behaviour observed between WT cells and STIM1-KO cells, since these measurements are performed in the absence of extracellular Ca^{2+} . Therefore, in this assay, SOCE is not an active pathway.

In summary, the effects observed depend on the conformational state of STIM1, but not on the activation or the absence of SOCE.

Q3: *The authors state that 'to rule out the possibility that this decrease was due to SOCE inhibition rather than STIM1 conformation, we reintroduced Ca^{2+} into the assay medium.' However, this is not sufficient to exclude a role for SOCE, and furthermore was only addressed for the Ca^{2+} depletion, and not for any of the other consequences of the STIM1 loss (i.e. PE synthesis, increase in MAMs) etc.*

A3: As we have already stated in the previous response, adding Ca^{2+} to the medium after thapsigargin-induced store depletion is indeed a method to activate SOCE, since store depletion induces the opening of ORAI1 channels. In fact, this reintroduction of Ca^{2+} into the assay medium follows the same protocol used to measure SOCE in cells loaded with the Ca^{2+} indicator Fura-2 (as shown in Fig EV4.C). Therefore, the addition of extracellular Ca^{2+} elevates cytosolic Ca^{2+} while maintaining ER depletion.

The STIM1(Δ 551–611) mutant, which exhibits normal SOCE activation comparable to WT cells, displays a clear defect in its interaction with GRP75 together with an increase in ER–mitochondria contacts. This demonstrates that the interaction defect cannot be attributed to the absence of SOCE, since, as mentioned, this mutant shows SOCE activation identical to that of WT cells (see Fig EV5.G).

In addition, in a new experiment included in this revised version of the manuscript, we stimulated the expression of the 551–611 peptide (a Flag-tagged peptide) in order to compete with the STIM1–GRP75 interaction. With this strategy, we achieved several outcomes: (1) a reduction in STIM1–GRP75 interaction (Fig 7.H-I), (2) a reduction in Ca^{2+} transfer between ER and mitochondria (Fig 7.J-K), and (3) an increase in ER–mitochondria contacts, detectable as a PLA signal between VAPB and PTPIP51 (see Fig 7.L-M). Importantly, these effects occurred without any impairment of SOCE, as the expression of the peptide does not alter this Ca^{2+} entry pathway (see Fig 7.N).

Taken together, these results confirm that SOCE does not influence either STIM1–GRP75 interaction or the number of ER–mitochondria contacts, which led us to exclude SOCE as a modulatory pathway in this context.

Q4: Are there *STIM1* mutants that do not interact with *Orai* that can be used? The authors could check whether the *STIM1* mutants they use are still capable of binding *Orai*, and consider performing experiments in which a MAMS defect is demonstrated in *Orai1* KO/*STIM1* *Orai1* DKO cells.

A4. The answer is **YES**. The *STIM1*(R429C) mutant does not interact with *ORAI1*, and this lack of interaction underlies the absence of SOCE. We have addressed a similar question raised by Reviewer 3, where we referred to the original publication (PMID: 25918394) that first described this mutation. For clarity, we reproduce here the relevant statement from that study:

The C-terminal domain of STIM1 exists in a closed conformation that is maintained by R429 and the coiled-coil domain 3 (CC3). After store depletion, the C-terminal domain undergoes a CC3-dependent conformational change, resulting in exposure of the polybasic domain, STIM1 translocation to ER–plasma membrane junctions, oligomerization, and binding to the plasma membrane channel ORAI1. However, the STIM1-R429C mutant has an extended conformation with an exposed polybasic domain and constitutive localization at ER–PM junctions but fails to oligomerize and bind to ORAI1 due to destabilization of the CC3 structure (PMID: 25918394).

Therefore, in our study we have demonstrated that defective SOCE, caused by the *STIM1*(R429C) mutant, exhibits reduced interaction with GRP75. In contrast, the *STIM1*(Δ 551–611) mutant, which loses interaction with GRP75 but retains intact SOCE, further supports our conclusions. Taken together, these complementary findings allowed us to propose the molecular mechanism summarized in the final figure of the revised manuscript.

Minor:

Q5: Lack of control in IPs. Whole cell lysate and IPs should be run together (unclear why they are run separately, and only one 'WCL' is run with the pull-down fractions).

A5. We believe there is no flaw in the design of the co-IP experiments. Whole-cell lysates (WCL), one per co-IP membrane, are included to indicate the specificity of the antibody on that membrane. The WCLs for all experimental conditions corresponding to each IP can be run in parallel on a separate blot, since the IP lanes should be compared with each other, whereas the WCL lanes are compared among themselves as loading controls. The WCL blots confirm that all lanes contain the same amount of the epitope targeted for immunoprecipitation. Because the quantification of WCL lanes—provided that they are equally loaded—is never directly compared with the quantification of the IP lanes, we do not consider it necessary to include all conditions within the same blot. This has been our standard approach since 2010, when we published our first *STIM1* co-IP (PMID: 20736304).

Q6: Furthermore, IPs should be properly controlled. Eg. Fig. 1B/2B: the authors didn't analyze the IP fraction for any other mito or ER proteins to assess whether this is a specific interaction.

A6. While we understand that the question is aimed at assessing the specificity of the IPs, our response is that the experiments indeed include all the appropriate controls. The specificity of the antibody and the IPs is determined by the negative controls (such as the use of a non-specific antiserum, or an empty GFP vector in the case of GFP-tagged proteins). Specificity is not defined by the presence or absence of interaction with other proteins, since this depends on the nature of the protein interactome (in this case, STIM1).

In fact, in the present study we show that STIM1 interacts with PTPIP51 (Fig 1.F-G), a mitochondrial protein, but not with VDAC1/2/3 (Fig 1.A), which are also mitochondrial. By contrast, interactions with ER-resident proteins should not be used as a specificity criterion, since such interactions are highly likely to occur; in all such cases, the IP would still be specific if those proteins are part of STIM1's interactome.

Q7: STIM1 SOCE, STIM1 MAMs: The authors should discuss how these different subpopulations of Stim1 may be regulated.

We thank the reviewer for this comment, as it is indeed reasonable that readers may question the potential sorting of the protein into different subcellular compartments. At present, there is no definitive answer to this question. However, we do know that STIM1 displays a differential distribution: our group has shown that a subpopulation of STIM1 localizes to the nuclear envelope and, in fact, translocates to the inner nuclear membrane, where it interacts with chromatin to provide a protein scaffold for DNA damage repair (PMID: 38224453).

Since MAMs are cholesterol-rich regions and STIM1 contains cholesterol-interacting domains (PMID: 27459950), one possibility is that STIM1 simply diffuses into these regions due to its affinity for cholesterol. The lipid interaction domains, as well as those with cholesterol, are already included in the first part of the *Discussion*. Nevertheless, this cannot fully explain the distribution, as MAMs are not the exclusive localization site of STIM1. Another possibility is that post-translational modifications regulate its localization, given that STIM1 is subject to multiple modifications, particularly phosphorylation. However, this remains speculative, as no previous study has directly addressed this question.

Dear Dr. Martin-Romero,

We have now received re-review reports from three referees, which I have included below. As you will see, you have addressed their concerns satisfactorily; however, I would like you to discuss the points raised by Reviewer #2 regarding the mechanism through which STIM1 operates in this context. Before I can finally accept the manuscript, there are some remaining editorial points which need to be addressed. In this regard would you please:

- include funding information in the 'Acknowledgements' section,
- include five keywords immediately below the abstract,
- keep the "Disclosure and competing interests statement", but remove the other conflict of interest statement,
- remove the AC/Credit section from the text,
- you are invited to upload any protocols associated with the study that might be of use to the community,
- ensure all figure callouts are listed sequentially, and refer to all figure panels for main and EV figures, and provide the exact p-values for the same in the legend of figure 1H, 2B, C, E; 3B, C, D, I, J; 4F, G, I; 5B-D, F, K, M, N, O; 6B, D, F, G, I, K, M; 7E, G, I, J, K, M; EV1 D, E, G; EV2 B-I; EV4 C as appropriate,
- indicate the statistical test used for data analysis in the legends of figures 1H, 2B, C, E; 3B, C, D, F, I, J; 4B, C, D, F, G, I; 5B-D, F-K, M-O; 6B, D, F, G, I, K, M, N, O; 7E, G, I, J, K, M; EV1 D, E, G ; EV2 B-I; EV3 A-D; EV4 C, D; EV5 K-M,
- define the nature of n in the legends of figures 3B, C, F, 4I, 7J, EV1 A, EV5 K, L,
- define error bars in the legends of figures 5B-D, F-J, M; 6B, F, I; 7I, N; EV1 A, EV2 H, I; EV3 A-D,
- change the name of the "Materials and methods" section to "Methods", and
- correct the section order as follows: Title page - Abstract - Keywords - Introduction - Results - Discussion - Methods - Data Availability - Acknowledgements - Disclosure and Competing Interests Statement - References - Figure Legends - Table(s) - Expanded View Figure Legends.

We include a synopsis of the paper (see <http://emboj.embojournal.org/>). Please provide me with a general summary image, a two sentence statement and 3-5 bullet points that capture the key findings of the paper.

I am looking forward to receiving your revised manuscript.

EMBO Press is an editorially independent publishing platform for the development of EMBO scientific publications.

Best wishes,

William

William Teale, PhD
Editor
The EMBO Journal
w.teale@embojournal.org

Read our guidance for manuscript revisions and related editorial policies: <https://link.springer.com/journal/44318/submission-guidelines#cms-Revised-submissions>

<https://media.springernature.com/original/springer-cms/rest/v1/content/27825798/data/v1>

- a point-by-point response to the referees' comments, with a detailed description of the changes made (as a word file).
- a word file of the manuscript text.
- individual production quality figure files (one file per figure)
- a complete author checklist
- Expanded View files (replacing Supplementary Information)
- a Reagents and Tools Table as part of the Methods section

Please remember: Digital image enhancement is acceptable practice, as long as it accurately represents the original data and conforms to community standards. If a figure has been subjected to significant electronic manipulation, this must be noted in the figure legend or in the 'Methods' section. The editors reserve the right to request original versions of figures and the original images that were used to assemble the figure.

We realize that it is difficult to revise to a specific deadline. In the interest of protecting the conceptual advance provided by the work, we recommend a revision within 3 months (4th Mar 2026). Please discuss the revision progress ahead of this time with the editor if you require more time to complete the revisions. Use the link below to submit your revision:

Referee #2:

The manuscript has improved and now includes additional data and revisions based on the previous comments. however, there are still few points that require clarification or correction. The impact of the study could be further increased and may require additional experiments, as detailed below.

The data support that STIM1 influences calcium transfer through its interaction with GRP75; however, it is not clearly defined whether STIM1 functions as a tethering factor regulating calcium transfer, or whether additional mechanisms are involved whereby STIM1 more directly modulates calcium transfer-for example, through interactions with IP3R3, SERCA, or via PTPIP51 and GRP75. This should be discussed and clearly stated throughout the manuscript.

In addition, the potential role of STIM1 as a tethering factor modulating the VAPB-PTPIP51 interaction (as shown) or forming a complex with PTPIP51-as observed in the Co-IP analyses-should be further clarified regarding how these interactions influence calcium transfer capacity. Discussion and investigation of this point would strengthen the manuscript. Have the authors tested any mutants that specifically impair interaction with PTPIP51? Including such data would be important.

The authors evaluated MAMs and measured mitochondrial function; however, no mitochondrial morphology analysis was included. It would be important to add mitochondrial imaging to assess changes in mitochondrial network structure under the experimental conditions. This would help define the functional impact of MAMs on mitochondrial morphology and the function as they showed. Co-staining of the ER in parallel would also be valuable. Although the EM analysis shown is based on 2D images, EM remains the gold-standard method for studying contact sites, particularly when combined with PLA and fluorescence reporters; therefore, including this dataset-ideally alongside endogenous ER and mitochondrial staining-would strengthen the conclusions. Images corresponding to the MitoSOX or MitoGreen quantifications could also be included. These additions would enhance the overall impact of the work.

Regarding the calcium imaging experiments, it would be helpful to test endogenous modulation of STIM1 and contact sites (PLA assay, STIM1/VAPB-PTPIP51 and VDAC1-IP3R3) under varying intracellular calcium conditions-for example, using conditions without extracellular calcium followed by ATP + CCh stimulation, or using an ionophore to induce a global increase in intracellular calcium. Under these conditions, the authors could assess whether increased cytosolic calcium promotes STIM1 recruitment to contact sites and interaction with GRP75 or other MAM proteins. Additionally, examining the effects of calcium chelation (e.g., with BAPTA) would be informative. These experiments would provide stronger evidence for STIM1 association with MAMs under changes in intracellular calcium.

Referee #3:

The revised manuscript has been carefully revised in line with my comments. However, I have a couple of minor comments that may help improve the credibility/readability of the work.

Specific Comments:

1. I appreciate the changes to the results section made by the authors regarding the quantitation of ER Ca²⁺ content. I would also recommend the following comment:

"While this approach provides an internally consistent basis for comparison, we acknowledge that the absolute numbers may be over- or under-estimated due to lack of certainty regarding F_{min} and F_{max} ."

2. Regarding the increase in ROS after loss of Complex I, please cite prior papers establishing this point as consistent with your findings.

3. Regarding the R429C mutant, my comment was, essentially, a "tip". The use of the term "unfolding" is confusing to someone working on STIM. Please describe as a "constitutively active STIM1 mutant."

Referee #4:

The reviewers have addressed all concerns.

Dear Dr. William Teale,

Thank you again for the time and attention you have devoted to the editorial evaluation of our work. I would like to sincerely express my appreciation for the careful and professional handling of this manuscript throughout the process. Below, we present the revisions to the manuscript as requested, together with our responses to reviewers 2,3, and 4. We hope that these revisions and our responses will be helpful toward the final acceptance of our manuscript.

Sincerely,

Francisco Javier Martin-Romero
Professor of Biochemistry and Molecular Biology
University of Extremadura

Dear Dr. Martin-Romero,

We have now received re-review reports from three referees, which I have included below. As you will see, you have addressed their concerns satisfactorily; however, I would like you to discuss the points raised by Reviewer #2 regarding the mechanism through which STIM1 operates in this context.

Before I can finally accept the manuscript, there are some remaining editorial points which need to be addressed. In this regard would you please:

- include funding information in the 'Acknowledgements' section,

The funding is now included in the "Acknowledgements" section

- include five keywords immediately below the abstract,

We have added five keywords below the abstract

- keep the "Disclosure and competing interests statement", but remove the other conflict of interest statement,

We have reformatted the section, as indicated.

- remove the AC/CrediT section from the text,

To the best of my knowledge, the AC/CrediT information was not in the main text file.

- you are invited to upload any protocols associated with the study that might be of use to the community,

In this case, there are no additional protocols to be uploaded.

- ensure all figure callouts are listed sequentially, and refer to all figure panels for main and EV figures,

Minor errors in callouts have been corrected, as follows:

Fig EV1.F-G. is mentioned for the first time in page 7, after Fig EV1.A-E, and not in page 19, as in the previous version of the manuscript.

Fig. EV5.J is now mentioned in page 13.

and provide the exact p-values for the same in the legend of figure 1H, 2B, C, E; 3B, C, D, I, J; 4F, G, I; 5B-D, F, K, M, N, O; 6B, D, F, G, I, K, M; 7E, G, I, J, K, M; EV1 D, E, G; EV2 B-I; EV4 C as appropriate,

- indicate the statistical test used for data analysis in the legends of figures 1H, 2B, C, E; 3B, C, D, F, I, J; 4B, C, D, F, G, I; 5B-D, F-K, M-O; 6B, D, F, G, I, K, M, N, O; 7E, G, I, J, K, M; EV1 D, E, G ; EV2 B-I; EV3 A-D; EV4 C, D; EV5 K-M,

- define the nature of n in the legends of figures 3B, C, F, 4I, 7J, EV1 A, EV5 K, L,

- define error bars in the legends of figures 5B-D, F-J, M; 6B, F, I; 7I, N; EV1 A, EV2 H, I; EV3 A-D,

All this information is now included in the legends of the figures. In addition, we would like to state that:

- P values lower than 0.0001 were provided as $p < 0.0001$ by the GraphPad software so we do not have the exact number in this specific case. Other values have been added to the legends.
- The nature of N in Fig 3B, 3C, and 3D are regions of interests (ROIs) and this is stated already in the legend.
- The nature of N in Fig 3F are cells, and this is already stated in the legend.
- The nature of N in Fig 4I are cells, and this is stated in the legend.
- The nature of N in Fig EV1.A are biological replicates, and this is already stated in the legend.
- The nature of N in Fig EV5.K, L, and M are ROIs, and this is already stated in the legend.
- The statistical analysis in Fig EV3.A was done with paired t-test, and this information is already provided in the legend.

- change the name of the "Materials and methods" section to "Methods", and
- correct the section order as follows: Title page - Abstract - Keywords -
Introduction - Results - Discussion - Methods - Data Availability -
Acknowledgements - Disclosure and Competing Interests Statement -
References - Figure Legends - Table(s) - Expanded View Figure Legends.

Materials and Methods section is now "Methods". The order of the manuscript has been corrected. The only modification was the relocation of the "References" section before "Figure legends".

We include a synopsis of the paper (see <http://emboj.embopress.org/>). Please provide me with a general summary image, a two sentence statement and 3-5 bullet points that capture the key findings of the paper.

The general summary image could be the current Figure 8, as it summarizes the main conclusions.

Two-sentence statement:

This study shows that STIM1 localizes at ER-mitochondria contact sites and directly interacts with the mitochondrial protein GRP75 to regulate Ca²⁺ transfer between the two organelles. Loss of STIM1 disrupts this calcium signalling, leading to impaired mitochondrial function and triggering compensatory increases in ER-mitochondria contacts.

Bullet points:

- STIM1 is present at mitochondria-associated ER membranes (MAMs) and directly interacts with mitochondrial-associated proteins such as GRP75 and PTPIP51.
- STIM1 deficiency increases the number of ER-mitochondria contacts sites but reduces functional Ca²⁺ transfer to mitochondria.
- The interaction between STIM1 and GRP75 depends on STIM1 conformation, with the closed form promoting binding.
- A specific STIM1 region (amino acids 551-611) is essential for GRP75 binding and proper ER-mitochondria Ca²⁺ transfer, independent of SOCE.

I am looking forward to receiving your revised manuscript.

EMBO Press is an editorially independent publishing platform for the development of EMBO scientific publications.

Referee #2:

The manuscript has improved and now includes additional data and revisions based on the previous comments. However, there are still few points that require clarification or correction. The impact of the study could be further increased and may require additional experiments, as detailed below.

The data support that STIM1 influences calcium transfer through its interaction with GRP75; however, it is not clearly defined whether STIM1 functions as a tethering factor regulating calcium transfer, or whether additional mechanisms are involved whereby STIM1 more directly modulates calcium transfer—for example, through interactions with IP3R3, SERCA, or via PTPIP51 and GRP75. This should be discussed and clearly stated throughout the manuscript.

We provide evidence that STIM1 interacts with GRP75 and PTPIP51. Furthermore, our data demonstrate that these interactions are dynamic, essential for calcium regulation, responsive to physiological stimuli, and dependent on a specific domain of STIM1. While additional mechanisms may also contribute to these processes, exploring them is beyond the scope of the present study.

In addition, the potential role of STIM1 as a tethering factor modulating the VAPB-PTPIP51 interaction (as shown) or forming a complex with PTPIP51—as observed in the Co-IP analyses—should be further clarified regarding how these interactions influence calcium transfer capacity. Discussion and investigation of this point would strengthen the manuscript. Have the authors tested any mutants that specifically impair interaction with PTPIP51? Including such data would be important.

We agree with the reviewer that these are really interesting questions for future research. At present, we do not have data that confirm a direct interaction with PTPIP51, as the PLA indicates proximity and the co-immunoprecipitation (co-IP) indicates complex formation, but there is no evidence of a direct interaction. Therefore, this line of investigation deserves dedicated and in-depth study, which is clearly beyond the scope of a first article in this field, particularly when introducing a novel concept (the role of STIM1 in ER–mitochondria communication) and a mechanistic insight (interaction with GRP75 and identification of the interaction domain).

The authors evaluated MAMs and measured mitochondrial function; however, no mitochondrial morphology analysis was included. It would be important to add mitochondrial imaging to assess changes in mitochondrial network structure under the experimental conditions. This would help define the

functional impact of MAMs on mitochondrial morphology and the function as they showed. Co-staining of the ER in parallel would also be valuable

We agree with the reviewer that many authors use alterations in mitochondrial morphology to sustain the impact of alterations in ER-mitochondria connections or other insults. However, we note that morphology analysis by imaging is not a reliable measurement of mitochondrial functionality. For that reason, we have included Seahorse analysis.

Although the EM analysis shown is based on 2D images, EM remains the gold-standard method for studying contact sites, particularly when combined with PLA and fluorescence reporters; therefore, including this dataset-ideally alongside endogenous ER and mitochondrial staining-would strengthen the conclusions. Images corresponding to the MitoSOX or MitoGreen quantifications could also be included. These additions would enhance the overall impact of the work.

We thank the reviewer for these comments. As mentioned above, we provide clear evidence of interaction of STIM1-GRP75, including PLA assays as suggested.

While we appreciate the reviewer's insight, we would like to offer clarification regarding the absence of further ER-mitochondria structural data in our study. Rather than the physical interaction between ER and mitochondria, our work is focused on functional studies without heavily relying on the connectivity between ER and mitochondria markers as read outs of MAM regulation. The reason why being that in our experience imaging data may reveal defects in the physical apposition between organelles without necessarily signifying a functional interaction.

While the reviewer also proposes to perform EM, it is worth noting that, at the moment, there are no data that consistently and rigorously link the degree of physical apposition and functional crosstalk. Therefore, the physical distance between ER-mitochondria considered to be functional in imaging data is set by the observer, and thus arbitrary.

Regarding the calcium imaging experiments, it would be helpful to test endogenous modulation of STIM1 and contact sites (PLA assay, STIM1/VAPB-PTPTI51 and VDAC1-IP3R3) under varying intracellular calcium conditions-for example, using conditions without extracellular calcium followed by ATP + CCh stimulation, or using an ionophore to induce a global increase in intracellular calcium. Under these conditions, the authors could assess whether increased cytosolic calcium promotes STIM1 recruitment to contact sites and interaction with GRP75 or other MAM proteins. Additionally, examining the effects of

calcium chelation (e.g., with BAPTA) would be informative. These experiments would provide stronger evidence for STIM1 association with MAMs under changes in intracellular calcium.

At this point, we would like to note to the reviewer that the experiments suggested have already been performed. The STIM1-GRP75 interaction in response to depletion of intracellular Ca^{2+} stores (in the absence of extracellular Ca^{2+} followed by stimulation with ATP + carbachol, as suggested by the reviewer) is precisely what is shown in Fig 6, panels E–F. Likewise, the response to an increase in intracellular calcium (which the reviewer suggests inducing with an ionophore) is what we have shown using thapsigargin at short time points in Fig 6, panels A–B and C–D.

While it is of course possible to perform additional experimental combinations, we believe these would not add substantial value beyond what was already requested in the first round of review and what is now presented in the revised manuscript.

With respect to additional interactions, such as those involving VDAC and IP3R3, as mentioned above, we agree with the reviewer that these represent highly interesting questions for future research efforts.

Referee #3:

The revised manuscript has been carefully revised in line with my comments. However, I have a couple of minor comments that may help improve the credibility/readability of the work.

Specific Comments:

1. I appreciate the changes to the results section made by the authors regarding the quantitation of ER Ca²⁺ content. I would also recommend the following comment:

"While this approach provides an internally consistent basis for comparison, we acknowledge that the absolute numbers may be over- or under-estimated due to lack of certainty regarding F_{min} and F_{max}."

We thank the reviewer for the careful evaluation of our work and for the constructive suggestions to improve it. As we indicated in our response during the first round of review, we fully agree with the reviewer's comment regarding the calibration of Ca²⁺ measurements, and this paragraph has now been incorporated into the manuscript in the Results section, where we believe it is more appropriately placed, specifically within the subsection presenting the ER Ca²⁺ measurements. We have slightly modified the wording to improve clarity, as follows:

"...While this approach provides a consistent internal basis for comparison, we acknowledge that the absolute values may be either overestimated or underestimated due to the lack of high accuracy in the determination of F_{min} and F_{max}."

2. Regarding the increase in ROS after loss of Complex I, please cite prior papers establishing this point as consistent with your findings.

The previous version of the manuscript already cited the reference Guarás *et al* (2016) The CoQH₂/CoQ ratio serves as a sensor of respiratory chain efficiency. *Cell Rep* 15: 197–209. We have now included an additional reference: Scialò F, Fernández-Ayala DJ & Sanz A (2017) Role of Mitochondrial Reverse Electron Transport in ROS Signaling: Potential Roles in Health and Disease. *Front Physiol* 8: 42.

3. Regarding the R429C mutant, my comment was, essentially, a "tip". The use of the term "unfolding" is confusing to someone working on STIM. Please describe as a "constitutively active STIM1 mutant."

As requested by the reviewer, the R429C mutant has been described in page 11, under the section "STIM1 conformation governs its interaction with GRP75",

as follows: ...the STIM1(R429C) mutant, which adopts a constitutively open conformation as a constitutively active STIM1.

We would like to thank the reviewer for the time and dedication devoted to the review of our manuscript, as the suggestions have significantly improved the final version of the manuscript.

Referee #4:

The reviewers have addressed all concerns.

We would like to thank the reviewer for the time and dedication devoted to the review of our manuscript,

Dear Dr. Martin-Romero,

I am pleased to inform you that your manuscript has been accepted for publication in the EMBO Journal.

Congratulations to you and your team!

You may qualify for financial assistance for your publication charges - either via a Springer Nature fully open access agreement or an EMBO initiative. Check your eligibility: <https://link.springer.com/journal/44318/how-to-publish-with-us>

Yours sincerely,

William Teale

William Teale, PhD
Editor
The EMBO Journal
w.teale@embojournal.org

Please note that it is The EMBO Journal policy for the transcript of the editorial process (containing referee reports and your response letters) to be published as an online supplement to each paper. If you should prefer removal of any referee-only figures included in the point-by-point response(s), e.g. because they may still be used for future publication or because they have been reproduced from published work by others, please do let us know immediately via response email.

More information is available here: <https://link.springer.com/partners/embo-press/editorial-policies#Peer%20review>